# A Framework for Bilevel Optimization on Riemannian Manifolds

**Andi Han**[1]   **Bamdev Mishra**[2]   **Pratik Jawanpuria**[2]   **Akiko Takeda**[1]
[1]RIKEN AIP   [2]Microsoft, India   [3]University of Tokyo
andi.han@riken.jp
{bamdevm, pratik.jawanpuria}@microsoft.com.
takeda@mist.i.u-tokyo.ac.jp

## Abstract

Bilevel optimization has gained prominence in various applications. In this study, we introduce a framework for solving bilevel optimization problems, where the variables in both the lower and upper levels are constrained on Riemannian manifolds. We present several hypergradient estimation strategies on manifolds and analyze their estimation errors. Furthermore, we provide comprehensive convergence and complexity analyses for the proposed hypergradient descent algorithm on manifolds. We also extend our framework to encompass stochastic bilevel optimization and incorporate the use of general retraction. The efficacy of the proposed framework is demonstrated through several applications.

## 1 Introduction

Bilevel optimization is a hierarchical optimization problem where the upper-level problem depends on the solution of the lower-level, i.e.,

$$\min_{x\in\mathbb{R}^{d_x}} F(x) = f(x, y^*(x)), \qquad \text{s.t. } y^*(x) = \arg\min_{y\in\mathbb{R}^{d_y}} g(x, y).$$

Applications involving bilevel optimization include meta learning [16], hyperparameter optimization [18], and neural architecture search (NAS) [53], to name a few. The lower-level problem is usually assumed to be strongly convex.

Common strategies for solving such problem can be classified into two categories: single-level reformulation [29, 60] and approximate hypergradient descent [19, 40]. The former aims to reformulate the bilevel optimization problem into a single-level one using the optimality conditions of the lower-level problem as constraints. However, this may impose a large number of constraints for machine learning applications. The latter scheme directly solves the bilevel problem through iteratively updating the lower and upper-level parameters and, hence, is usually more efficient. Nevertheless, existing works have mostly focused on unconstrained bilevel optimization [19, 32, 40, 11, 52, 45, 14].

In this work, we study bilevel optimization problems where $x$ and $y$ are on Riemannian manifolds $\mathcal{M}_x$ and $\mathcal{M}_y$, respectively. We focus on the setup where the lower-level function $g(x, y)$ is geodesic strongly convex (a generalized notion of convexity on manifolds, defined in Section 2) in $y$. This ensures the lower-level problem has a unique solution $y^*(x)$ given $x$. The upper-level function $f$ can be nonconvex on $\mathcal{M}_x \times \mathcal{M}_y$. Because the unconstrained bilevel optimization is a special case of our formulation on manifolds, such a formulation includes a wider class of applications. Examples of Riemannian bilevel optimization include Riemannian meta learning [64] and NAS over SPD networks [62]. Moreover, there has been a surge of interest of min-max optimization over Riemannian manifolds [37, 41, 73, 27, 25, 67, 35], which also gets subsumed in the framework of bilevel optimization with $g = -f$.

38th Conference on Neural Information Processing Systems (NeurIPS 2024).

**Contributions. (i)** We derive intrinsic Riemannian hypergradient via the implicit function theorem and propose four strategies for estimating the hypergradient, i.e., through Hessian inverse, conjugate gradient, truncated Neumann series, and automatic differentiation. We then provide hypergradient estimation error bounds for all the proposed strategies. **(ii)** We introduce the Riemannian hypergradient descent algorithm to solve bilevel optimization problems on manifolds and provide convergence guarantees. We also generalize the framework to the stochastic setting and to allow the use of retraction. **(iii)** The efficacy of the proposed modeling is shown on several problem instances including hyper-representation over SPD matrices, Riemannian meta learning, and unsupervised domain adaptation. The proofs, extensions, and experimental details are deferred to the appendix sections.

**Related works in unconstrained setting.** Unconstrained bilevel optimization where the lower-level problem is strongly convex has been widely studied [19, 32, 40, 11, 52, 45, 14]. A crucial ingredient is the notion of hypergradient in bilevel optimization problems and its computation. There exist strategies for approximating the hypergradient, e.g., using conjugate gradient [40], Neumann series [19], iterative differentiation [21], and Nyström method [31]. While bilevel optimization with constraints is relatively unexplored, a few works exists that impose constraints only for the upper level problem [32, 10]. Recently, linearly lower-level constrained bilevel optimization has been explored in [65, 68], where a projected gradient method is employed for the lower-level problem.

**Related works on manifolds.** There has been limited work on bilevel optimization problems on manifolds. [7] studies semivectorial bilevel optimization on Riemannian manifolds where the upper-level is a scalar optimization problem while the lower-level is a multiobjective problem under greatest coalition. [50, 49] reformulate bilevel problems on manifolds into a single-level problem based on the KKT conditions on manifolds. However, for all those works, it is unclear whether there exists an algorithm that efficiently solves the problem in large-scale settings. In contrast, we aim to provide a general framework for solving bilevel optimization on Riemannian manifolds. [47] is a contemporary work that also proposes gradient-based algorithms for bilevel optimization on Riemannian manifolds. The main differences of our work with respect to [47] are as follows: (1) We provide an analysis for various hypergradient estimators while [47] focuses on conjugate gradient for deterministic setting and Neumann series for stochastic setting; (2) We provide an analysis for retraction which is more computationally efficient than exponential map and parallel transport employed in [47]; and (3) We explore the utility of Riemannian bilevel optimization in various machine learning applications, which is not the case with [47].

## 2 Preliminaries and notations

A Riemannian manifold $\mathcal{M}$ is a smooth manifold equipped with a smooth inner product structure (a Riemannian metric) $\langle \cdot, \cdot \rangle_p : T_z\mathcal{M} \times T_z\mathcal{M} \to \mathbb{R}$ for any $z \in \mathcal{M}$ and its tangent space $T_z\mathcal{M}$. The induced norm is thus $\|u\|_z = \sqrt{\langle u, u \rangle_z}$ for any $u \in T_z\mathcal{M}$. A geodesic $c : [0, 1] \to \mathcal{M}$ generalizes the line segment in the Euclidean space as the locally shortest path on manifolds. The exponential map on a manifold is defined as $\mathrm{Exp}_z(u) = c(1)$ for a geodesic $c$ that satisfies $c(0) = z, c'(0) = u$. In a totally normal neighbourhood $\mathcal{U}$ where exponential map has a smooth inverse, the Riemannian distance $d(x, y) = \|\mathrm{Exp}_x^{-1}(y)\|_x = \|\mathrm{Exp}_y^{-1}(x)\|_y$. The parallel transport operation $\Gamma_{z_1}^{z_2} : T_{z_1}\mathcal{M} \to T_{z_2}\mathcal{M}$ is a linear map which preserves the inner product, i.e., $\langle u, v \rangle_{z_1} = \langle \Gamma_{z_1}^{z_2}u, \Gamma_{z_1}^{z_2}v \rangle_{z_2}, \forall u, v \in T_{z_1}\mathcal{M}$. The (Cartesian) product of Riemannian manifolds $\mathcal{M}_x \times \mathcal{M}_y$ is also a Riemannian manifold.

For a differentiable function $f : \mathcal{M} \to \mathbb{R}$, the Riemannian gradient $\mathcal{G}f(z) \in T_z\mathcal{M}$ is the tangent vector that satisfies $\langle \mathcal{G}f(z), u \rangle_z = \mathrm{D}f(z)[u]$ for all $u \in T_z\mathcal{M}$. Here D is the differential operator and $\mathrm{D}f(z)[u]$ represents the directional derivative of $f$ at $z$ along $u$. For a twice differentiable function $f$, Riemannian Hessian $\mathcal{H}f(z)$ is defined as the covariant derivative of Riemannian gradient.

Geodesic convexity extends the convexity notion in the Euclidean space to Riemannian manifolds. A geodesic convex set $\mathcal{Z} \subseteq \mathcal{M}$ is where any two points can be joined by a geodesic. A function $f : \mathcal{M} \to \mathbb{R}$ is said to be geodesic (strongly) convex if for all geodesics $c : [0, 1] \to \mathcal{Z}, f(c(t))$ is (strongly) convex in $t \in [0, 1]$. If the function is smooth, then $f$ is called $\mu$-geodesic strongly convex if and only if $f(\mathrm{Exp}_z(tu)) \geq f(z) + t\langle \mathcal{G}f(z), u \rangle_z + t^2\frac{\mu}{2}\|u\|_z^2, \forall t \in [0, 1]$. An equivalent second-order characterization is $\mathcal{H}(z) \succeq \mu\mathrm{id}$, where we denote id as the identity operator.

For a bifunction $\phi : \mathcal{M}_x \times \mathcal{M}_y \to \mathbb{R}$, we denote $\mathcal{G}_x\phi(x, y), \mathcal{G}_y\phi(x, y)$ as the Riemannian (partial) gradient and $\mathcal{H}_x\phi(x, y), \mathcal{H}_y\phi(x, y)$ as the Riemannian Hessian. The Riemannian cross-derivatives

are linear operators $\mathcal{G}^2_{xy}\phi(x,y) : T_y\mathcal{M}_y \to T_x\mathcal{M}_x, \mathcal{G}^2_{yx}\phi(x,y) : T_x\mathcal{M}_x \to T_y\mathcal{M}_y$ defined as $\mathcal{G}^2_{xy}\phi(x,y)[v] = \mathrm{D}_y\mathcal{G}_x\phi(x,y)[v]$ for any $v \in T_y\mathcal{M}_y$ (with D representing the differential operator) and similarly for $\mathcal{G}^2_{yx}\phi(x,y)$. For a linear operator $T : T_x\mathcal{M}_x \to T_y\mathcal{M}_y$, the adjoint operator, denoted as $T^\dagger$ is defined with respect to the Riemannian metric, i.e., $\langle T[u], v\rangle_y = \langle T^\dagger[v], u\rangle_x$ for any $u \in T_x\mathcal{M}_x, v \in T_y\mathcal{M}_y$. The operator norm of $T$ is defined as $\|T\|_y := \sup_{u \in T_x\mathcal{M}_x : \|u\|_x=1} \|T[u]\|_y$.

## 3 Proposed Riemannian hypergradient algorithm

In this work, we consider the constrained bilevel optimization problem

$$\min_{x\in\mathcal{M}_x} F(x) := f(x, y^*(x)), \qquad \text{s.t. } y^*(x) = \underset{y\in\mathcal{M}_y}{\arg\min}\, g(x, y), \tag{1}$$

where $\mathcal{M}_x, \mathcal{M}_y$ are two Riemannian manifolds and $f, g : \mathcal{M}_x \times \mathcal{M}_y \to \mathbb{R}$ are real-valued jointly smooth functions. We focus on the setting where the lower-level function $g(x, y)$ is geodesic strongly convex. This ensures the lower-level problem has a unique solution $y^*(x)$ for a given $x$. The upper-level function $f$ can be nonconvex on $\mathcal{M}_x \times \mathcal{M}_y$.

We propose to minimize $F(x)$ directly within the Riemannian optimization framework. To this end, we need the notion of the Riemannian gradient of $F(x) := f(x, y^*(x))$, which we call the Riemannian hypergradient.

**Proposition 1.** *The differential of $y^*(x)$ and the Riemannian hypergradient of $F(x)$ are given by*

$$\begin{aligned}
\mathrm{D}y^*(x) &= -\mathcal{H}_y^{-1}g(x, y^*(x)) \circ \mathcal{G}^2_{yx}g(x, y^*(x)) \\
\mathcal{G}F(x) &= \mathcal{G}_x f(x, y^*(x)) - \mathcal{G}^2_{xy}g(x, y^*(x))[\mathcal{H}_y^{-1}g(x, y^*(x))[\mathcal{G}_y f(x, y^*(x))]].
\end{aligned} \tag{2}$$

The above proposition crucially relies on the implicit function theorem on manifolds [25] and requires the invertibility of the Hessian of the lower level function $f$ with respect to $y$. This is guaranteed in our setup as $f$ is geodesic strongly convex in $y$. Hence, there exists a unique differentiable function $y^*(x)$ that maps $x$ to the lower-level solution. We show the Riemannian hypergradient descent (RHGD) algorithm for (1) in Algorithm 1.

---

**Algorithm 1** Riemannian hypergradient descent (RHGD)

1: Initialize $x_0 \in \mathcal{M}_x, y_0 \in \mathcal{M}_y$.
2: **for** $k = 0, ..., K-1$ **do**
3: $\quad y_k^0 = y_k$.
4: $\quad$ **for** $s = 0, ..., S-1$ **do**
5: $\quad\quad y_k^{s+1} = \mathrm{Exp}_{y_k^s}(-\eta_y\, \mathcal{G}_y g(x_k, y_k^s))$.
6: $\quad$ **end for**
7: $\quad$ Set $y_{k+1} = y_k^S$.
8: $\quad$ Compute approximated hypergradient $\widehat{\mathcal{G}}F(x_k)$.
9: $\quad$ Update $x_{k+1} = \mathrm{Exp}_{x_k}(-\eta_x\widehat{\mathcal{G}}F(x_k))$.
10: **end for**

- Steps 3 to 7 solve the lower-level problem using the Riemannian gradient descent method. Since computing the optimal solution $y^*(x)$ is computationally challenging, we obtain an approximate solution $y_{k+1}$.
- Step 8 involves computing the Riemannian hypergradient $\mathcal{G}F(x)$ of $F(x)$. For computational efficiency, we compute an approximation $\widehat{\mathcal{G}}F(x_k)$.
- Step 9 is the usual exponential map to find the updated point $x_{k+1}$.

---

We highlight that Step 8 of Algorithm 1 approximates the Riemannian hypergradient. In the rest of the section, we discuss various computationally efficient ways to estimate the Riemannian hypergradient and discuss the corresponding theoretical guarantees for RHGD. The error of hypergradient approximation comes from the inaccuracies of $y_{k+1}$ to $y^*(x_k)$ and also from the Hessian inverse.

### 3.1 Hypergradient estimation

When the inverse Hessian of the lower-level problem can be computed efficiently, we can estimate the hypergradient directly by evaluating the Hessian inverse (**HINV**) at $y_{k+1}$, i.e., $\widehat{\mathcal{G}}_{\mathrm{hinv}}F(x_k) = \mathcal{G}_x f(x_k, y_{k+1}) - -\mathcal{G}^2_{xy}g(x_k, y_{k+1})[\mathcal{H}_y^{-1}g(x_k, y_{k+1})[\mathcal{G}_y f(x_k, y_{k+1})]]$. However, computing the inverse Hessian is computationally expensive in many scenarios. We now discuss three practical strategies for estimating the Riemannian hypergradient when $y_{k+1}$ is given.

**Conjugate gradient approach (CG).** When evaluating the Hessian inverse is difficult, we can solve the linear system $\mathcal{H}_y g(x_k, y_{k+1})[u] = \mathcal{G}_y f(x_k, y_{k+1})$ for some $u \in T_{y_{k+1}}\mathcal{M}_y$. To this

end, we employ the tangent space conjugate gradient algorithm (Appendix F, Algorithm 3) that solves the linear system on the tangent space $T_{y_{k+1}}\mathcal{M}_y$ with only access to Hessian-vector products, i.e., $\widehat{\mathcal{G}}_{\text{cg}}F(x_k) = \mathcal{G}_x f(x_k, y_{k+1}) - \mathcal{G}_{xy}^2 g(x_k, y_{k+1})[\hat{v}_k^T]$, where $\hat{v}_k^T$ is computed as a solution to $\mathcal{H}_y g(x_k, y_{k+1})[\hat{v}_k^T] = \mathcal{G}_y f(x_k, y_{k+1})$, where $T$ is the number of iterations of the tangent space conjugate gradient algorithm.

**Truncated Neumann series approach (NC).** The Neumann series states for an invertible operator $H$ such that $\|H\| \leq 1$, its inverse $H^{-1} = \sum_{i=0}^{\infty}(\text{id} - H)^i$, where id is the identity operator. An alternative approach to estimate the Hessian inverse is to use a truncated Neumann series, which leads to the following approximated hypergradient, $\widehat{\mathcal{G}}_{\text{ns}}F(x_k) = \mathcal{G}_x f(x_k, y_{k+1}) - \mathcal{G}_{xy}^2 g(x_k, y_{k+1})[\gamma \sum_{i=0}^{T-1}(\text{id} - \gamma\mathcal{H}_y g(x_k, y_{k+1}))^i[\mathcal{G}_y f(x_k, y_{k+1})]]$, where $\gamma$ is chosen such that $(\text{id} - \gamma\mathcal{H}_y g(x_k, y_{k+1})) \succ 0$. $\gamma$ can be set as $\gamma = \frac{1}{L}$, where the gradient operator is $L$-Lipschitz (discussed later in Definition 1). Empirically, we observe that this approach is faster than the conjugate gradient approach. However, it requires estimating $T$ and $L$ beforehand.

**Automatic differentiation approach (AD).** Another hypergradient estimation strategy follows the idea of iterative differentiation by backpropagation. After running several iterations of gradient update to obtain $y_{k+1}$ (which is a function of $x_k$), we can use automatic differentiation to compute directly the Riemannian gradient of $f(x_k, y_{k+1}(x_k))$ with respect to $x_k$. We can compute the Riemannian hypergradient from the differential in the direction of arbitrary $u \in T_{x_k}\mathcal{M}_x$ using basic chain rules.

## 3.2 Theoretical analysis

This section provides theoretical analysis for the proposed hypergradient estimators as well as the Riemannian hypergradient descent. First, we require the notion of Lipschitzness of functions and operators defined on Riemannian manifolds. Below, we introduce the definition in terms of bi-functions and bi-operators and state the assumptions that are required for the analysis.

**Definition 1** (Lipschitzness). (1) For a bifunction $f : \mathcal{M}_x \times \mathcal{M}_y \to \mathbb{R}$, we say $f$ has $L$ Lipschitz Riemannian gradient in $\mathcal{U}_x \times \mathcal{U}_y \subseteq \mathcal{M}_x \times \mathcal{M}_y$ if it satisfies for any $x, x_1, x_2 \in \mathcal{U}_x, y, y_1, y_2 \in \mathcal{U}_y$, $\|\Gamma_{y_1}^{y_2}\mathcal{G}_y f(x, y_1) - \mathcal{G}_y f(x, y_2)\|_{y_2} \leq Ld(y_1, y_2)$, $\|\mathcal{G}_x f(x, y_1) - \mathcal{G}_x f(x, y_2)\|_x \leq Ld(y_1, y_2)$, $\|\Gamma_{x_1}^{x_2}\mathcal{G}_x f(x_1, y) - \mathcal{G}_x f(x_2, y)\|_{x_2} \leq Ld(x_1, x_2)$ and $\|\mathcal{G}_y f(x_1, y) - \mathcal{G}_y f(x_2, y)\|_y \leq Ld(x_1, x_2)$.

(2) For an operator $\mathcal{G}(x, y) : T_y\mathcal{M}_y \to T_x\mathcal{M}_x$, we say $\mathcal{G}(x, y)$ is $\rho$-Lipschitz if it satisfies, $\|\Gamma_{x_1}^{x_2}\mathcal{G}(x_1, y) - \mathcal{G}(x_2, y)\|_{x_2} \leq \rho\, d(x_1, x_2)$ and $\|\mathcal{G}(x, y_1) - \mathcal{G}(x, y_2)\Gamma_{y_1}^{y_2}\|_x \leq \rho\, d(y_1, y_2)$.

(3) For an operator $\mathcal{H}(x, y) : T_y\mathcal{M}_y \to T_y\mathcal{M}_y$, we say $\mathcal{H}(x, y)$ is $\rho$-Lipschitz if it satisfies, $\|\Gamma_{y_1}^{y_2}\mathcal{H}(x, y_1)\Gamma_{y_2}^{y_1} - \mathcal{H}(x, y_2)\|_{y_2} \leq \rho\, d(y_1, y_2)$ and $\|\mathcal{H}(x_1, y) - \mathcal{H}(x_2, y)\|_y \leq \rho\, d(x_1, x_2)$.

It is worth mentioning that Definition 1 implies the joint Lipschitzness over the product manifold $\mathcal{M}_x \times \mathcal{M}_y$, which is verified in Appendix C.2. Due to the possible nonconvexity for the upper level problem, the optimality is measured in terms of the Riemannian gradient norm of $F(x)$.

**Definition 2** ($\epsilon$-stationary point). We call $x \in \mathcal{M}_x$ an $\epsilon$-stationary point of bilevel optimization (1) if it satisfies $\|\mathcal{G}F(x)\|_x^2 \leq \epsilon$.

**Assumption 1.** All the iterates in the lower level problem are bounded in a compact subset that contains the optimal solution, i.e., there exists a constants $D_k > 0$, for all $k$ such that $d(y_k^s, y^*(x_k)) \leq D_k$ for all $s$. Such a neighbourhood has unique geodesic. We take $\bar{D} := \max_k\{D_1, ..., D_k\}$.

**Assumption 2.** Function $f(x, y)$ has bounded Riemannian gradients, i.e., $\|\mathcal{G}_y f(x, y)\|_y \leq M$, $\|\mathcal{G}_x f(x, y)\|_x \leq M$ for all $(x, y) \in \mathcal{U}$ and the Riemannian gradients are $L$-Lipschitz in $\mathcal{U}$.

**Assumption 3.** Function $g(x, y)$ is $\mu$-geodesic strongly convex in $y \in \mathcal{U}_y$ for any $x \in \mathcal{U}_x$ and has $L$ Lipschitz Riemannian gradient $\mathcal{G}_x g(x, y), \mathcal{G}_y g(x, y)$ in $\mathcal{U}$. Further, the Riemannian Hessian $\mathcal{H}_y g(x, y)$, cross derivatives $\mathcal{G}_{xy}^2 g(x, y), \mathcal{G}_{yx}^2 g(x, y)$ are $\rho$-Lipschitz in $\mathcal{U}$.

Assumption 1 is standard in Riemannian optimization literature by properly bounding the domain of variables, which allows to express Riemannian distance in terms of (inverse) Exponential map. Also, the boundedness of the domain implies the bound on curvature, as is required for analyzing convergence for geodesic strongly convex lower-level problems [41, 71]. Assumptions 2 and 3 are common regularity conditions imposed on $f$ and $g$ in the bilevel optimization literature. This translates into the smoothness of the function $F$ and $\mathrm{D}y^*(x)$ (discussed in Appendix C.3).

Table 1: Comparison of first-order and second-order complexities for reaching $\epsilon$-stationarity. For stochastic algorithms, including HGD-NS, RSHGD-HINV, the complexities are measured with respect to the component functions $f_i, g_i$. Here, $G_f, G_g$ are the gradient complexities of function $f, g$, respectively, to reach an $\epsilon$-stationary point of (1). Also, we denote $JV_g, HV_g$ as the complexity of computing the second-order cross derivative and Hessian-vector product of function $g$.

| Methods | $G_f$ | $G_g$ | $JV_g$ | $HV_g$ |
|---|---|---|---|---|
| HGD-CG [40] | $O(\kappa_l^3\epsilon^{-1})$ | $\widetilde{O}(\kappa_l^4\epsilon^{-1})$ | $O(\kappa_l^3\epsilon^{-1})$ | $\widetilde{O}(\kappa_l^{3.5}\epsilon^{-1})$ |
| -AD [40] | $O(\kappa_l^3\epsilon^{-1})$ | $\widetilde{O}(\kappa_l^4\epsilon^{-1})$ | $\widetilde{O}(\kappa_l^4\epsilon^{-1})$ | $\widetilde{O}(\kappa_l^4\epsilon^{-1})$ |
| SHGD-NS [40, 11] | $O(\kappa_l^5\epsilon^{-2})$ | $\widetilde{O}(\kappa_l^9\epsilon^{-2})$ | $O(\kappa_l^5\epsilon^{-2})$ | $\widetilde{O}(\kappa^6\epsilon^{-2})$ |
| RHGD-HINV | $O(\kappa_l^3\epsilon^{-1})$ | $\widetilde{O}(\kappa_l^5\zeta\epsilon^{-1})$ | $O(\kappa_l^3\epsilon^{-1})$ | NA |
| -CG | $O(\kappa_l^4\epsilon^{-1})$ | $\widetilde{O}(\kappa_l^6\zeta\epsilon^{-1})$ | $O(\kappa_l^4\epsilon^{-1})$ | $\widetilde{O}(\kappa_l^{4.5}\epsilon^{-1})$ |
| -NS | $O(\kappa_l^3\epsilon^{-1})$ | $\widetilde{O}(\kappa_l^5\zeta\epsilon^{-1})$ | $O(\kappa_l^3\epsilon^{-1})$ | $\widetilde{O}(\kappa_l^4\epsilon^{-1})$. |
| -AD | $O(\kappa_l^3\epsilon^{-1})$ | $\widetilde{O}(\kappa_l^5\zeta\epsilon^{-1})$ | $\widetilde{O}(\kappa_l^5\zeta\epsilon^{-1})$ | $\widetilde{O}(\kappa_l^5\zeta\epsilon^{-1})$ |
| RSHGD-HINV | $O(\kappa_l^5\epsilon^{-2})$ | $\widetilde{O}(\kappa_l^9\zeta\epsilon^{-2})$ | $O(\kappa_l^5\epsilon^{-2})$ | NA |

We first bound the estimation error of the proposed schemes of approximated hypergradient as follows. For the hypergradient computed by automatic differentiation, we highlight that due to the presence of exponential map in the chain of differentiation, it is non-trivial to explicitly express $\mathrm{D}_{x_k}y_k^S$. Here, we adopt the property of exponential map (which is locally linear) in the ambient space [1], i.e., $\mathrm{Exp}_x(u) = x + u + O(\|u\|_x^2)$. This requires the use of tangent space projection of $\xi$ in the ambient space as $\mathcal{P}_x(\xi)$, which is solved for the $v$ such that $\langle v, \xi\rangle_x = \langle u, \xi\rangle$ for any $\xi \in T_x\mathcal{M}$.

For notation simplicity, we denote $\kappa_l := \frac{L}{\mu}$ and $\kappa_\rho := \frac{\rho}{\mu}$. For analysis, we consider $\kappa_\rho = \Theta(\kappa_l)$.

**Lemma 1** (Hypergradient approximation error bound). *Under Assumptions 1, 2, 3, we can bound the error for approximated hypergradient as*

1. **HINV:** $\|\widehat{\mathcal{G}}_{\mathrm{hinv}}F(x_k) - \mathcal{G}F(x_k)\|_{x_k} \le (L + \kappa_\rho M + \kappa_l L + \kappa_l\kappa_\rho M)d\big(y^*(x_k), y_{k+1}\big)$.

2. **CG:** $\|\widehat{\mathcal{G}}_{\mathrm{cg}}F(x_k) - \mathcal{G}F(x_k)\|_{x_k} \le \big(L + \kappa_\rho M + L(1 + 2\sqrt{\kappa_l})\big(\kappa_l + \frac{M\kappa_\rho}{\mu}\big)\big)d\big(y^*(x_k), y_{k+1}\big) +$
   $2L\sqrt{\kappa_l}\big(\frac{\sqrt{\kappa_l}-1}{\sqrt{\kappa_l}+1}\big)^T\|\hat{v}_k^0 - \Gamma_{y^*(x_k)}^{y_{k+1}}v_k^*\|_{y_{k+1}}$, *where* $v_k^* = \mathcal{H}_y^{-1}g(x_k, y^*(x_k))[\mathcal{G}_yf(x_k, y^*(x_k))]$.

3. **NS:** $\|\widehat{\mathcal{G}}_{\mathrm{ns}}F(x_k) - \mathcal{G}F(x_k)\|_{x_k} \le (L + \kappa_l L + \kappa_\rho M + \kappa_l\kappa_\rho M)d\big(y^*(x_k), y_{k+1}\big) + \kappa_l M(1 - \gamma\mu)^T$.

4. **AD:** *Suppose further there exist* $C_1, C_2, C_3 > 0$ *such that* $\|\mathrm{D}_{x_k}y_k^s\|_{y_k^s} \le C_1$, $\|\Gamma_x^y\mathcal{P}_xv - v\|_y \le C_2d(x,y)\|v\|_y$ *and* $\mathrm{D}_x\mathrm{Exp}_x(u) = \mathcal{P}_{\mathrm{Exp}_x(u)}\big(\mathrm{id} + \mathrm{D}_xu\big) + \mathcal{E}$ *where* $\|\mathcal{E}\|_{\mathrm{Exp}_x(u)} \le C_3\|\mathrm{D}_xu\|_x\|u\|_x$ *for any* $x, y \in \mathcal{U}$ *and* $v \in T_y\mathcal{M}_y$, $u \in T_x\mathcal{M}_x$. *Then,*

   $\|\widehat{\mathcal{G}}_{\mathrm{ad}}F(x_k) - \mathcal{G}F(x_k)\|_{x_k} \le \big(\frac{2M\widetilde{C}}{\mu - \eta_y\zeta L^2} + L(1 + \kappa_l)\big)(1 + \eta_y^2\zeta L^2 - \eta_y\mu)^{\frac{S-1}{2}}d(y_k, y^*(x_k)) +$
   $M\kappa_l(1 - \eta_y\mu)^S$, *where* $\widetilde{C} := (\kappa_l + 1)\rho + (C_2 + \eta_yC_3)L\big((1 - \eta_y\mu)C_1 + \eta_yL\big)$.

From Lemma 1, it is evident that the exact Hessian inverse exhibits the tightest bound, which is followed by conjugate gradient (CG) and truncated Neumann series (NS). Automatic differentiation (AD) presents the worst upper bound on the error due to the introduction of curvature constant $\zeta$, resulting in $(1 - \Theta(\frac{\mu^2}{L^2\zeta}))^S = (1 - \Theta(\frac{1}{\kappa_l^2\zeta}))^S$ for the trailing term, which could be much larger than $(1 - \gamma\mu)^T = (1 - \Theta(\frac{1}{\kappa_l}))^T$ for NS and $\big(\frac{\sqrt{\kappa_l}-1}{\sqrt{\kappa_l}+1}\big)^T = (1 - \Theta(\frac{1}{\sqrt{\kappa_l}}))^T$ for CG. Further, the error critically relies on the number of inner iterations $S$ compared with $T$ for CG and NS, and the constants $C_1, C_2, C_3$ can be large for manifolds with high curvature. We now present the main convergence result with the four proposed hypergradient estimation strategies.

**Theorem 1.** *Denote* $\Delta_0 := F(x_0) + d^2(y_0, y^*(x_0))$ *and* $L_F := \big(\frac{L}{\mu} + 1\big)\big(L + \frac{\tau M}{\mu} + \frac{\rho LM}{\mu^2} + \frac{L^2}{\mu}\big) = O(\kappa_l^3)$. *Under Assumptions 1, 2, 3, we have the following bounds on the hypergradient norm obtained by Algorithm 1.*

- **HINV:** *Let* $\eta_x = \frac{1}{20L_F}$ *and* $S \ge \widetilde{\Theta}(\kappa_l^2\zeta)$. *We have* $\min_{k=0,\ldots,K-1}\|\mathcal{G}F(x_k)\|_{x_k}^2 \le 80L_F\Delta_0/K$.

- **CG:** Let $\Lambda := C_v^2 + \kappa_l^2(\frac{5M^2 C_0^2 D^2}{\mu} + 1)$, where $C_v := \frac{M\kappa_\rho}{\mu} + \frac{M\kappa_\rho\kappa_l}{\mu} + \kappa_l^2 + \kappa_l$. Choosing $\eta_x = \frac{1}{24\Lambda}$, $S \geq \widetilde{\Theta}(\kappa_l^2\zeta)$, and $T_{\mathrm{cg}} \geq \widetilde{\Theta}(\sqrt{\kappa_l})$, we have $\min_{k=0,\ldots,K-1} \|\mathcal{G}F(x_k)\|_{x_k}^2 \leq \frac{96\Lambda}{K}(\Delta_0 + \|v_0^*\|_{y^*(x_0)}^2)$.

- **NS:** Choosing $\eta_x = \frac{1}{20L_F}$, $S \geq \widetilde{\Theta}(\kappa_l^2\zeta)$, and $T_{\mathrm{ns}} \geq \widetilde{\Theta}(\kappa\log(\frac{1}{\epsilon}))$ for an arbitrary $\epsilon > 0$, we have $\min_{k=0,\ldots,K-1} \|\mathcal{G}F(x_k)\|_{x_k}^2 \leq \frac{80L_F}{K}\Delta_0 + \frac{\epsilon}{2}$.

- **AD:** Choosing $\eta_x = \frac{1}{20L_F}$ and $S \geq \widetilde{\Theta}(\kappa_l^2\zeta\log(\frac{1}{\epsilon}))$ for an arbitrary $\epsilon > 0$, we have $\min_{k=0,\ldots,K-1} \|\mathcal{G}F(x_k)\|_{x_k}^2 \leq \frac{80L_F}{K}\Delta_0 + \frac{\epsilon}{2}$.

**Complexity analysis.** Based on the convergence guarantees in Theorem 1, we have analyzed (in Corollary 1), the computational complexity of the proposed algorithm with four different hypergradient estimation strategies in reaching the $\epsilon$-stationary point. The results are summarized in Table 1. For reference, we also provide the computational cost of Euclidean algorithms which solve bilevel Euclidean optimization problem [40]. We notice that except for CG, the gradient complexity for $f$ (i.e., $G_f$) matches the Euclidean version. For conjugate gradient, the complexity is higher by $O(\kappa_l)$, which is due to the additional distortion from the use of vector transport when tracking the error of conjugate gradient at each epoch. In terms of gradient complexity for $g$ (i.e., $G_g$), all deterministic methods require a higher complexity by at least $\widetilde{O}(\kappa_l\zeta)$ compared to the Euclidean baselines. This is because of the curvature distortion when analyzing the convergence for geodesic strongly convex functions. Similar comparisons can be also made with respect to the computations of cross-derivatives and Hessian vector products.

### 3.3 Extension to stochastic bilevel optimization

**Algorithm 2** Riemannian stochastic bilevel optimization with Hessian inverse.

---
1: Initialize $x_0 \in \mathcal{M}_x, y_0 \in \mathcal{M}_y$.
2: **for** $k = 0, \ldots, K-1$ **do**
3:    $y_k^0 = y_k$.
4:    **for** $s = 0, \ldots, S-1$ **do**
5:       Sample a batch $\mathcal{B}_1$.
6:       $y_k^{s+1} = \mathrm{Exp}_{y_k^s}(-\eta_y \, \mathcal{G}_y g_{\mathcal{B}_1}(x_k, y_k^s))$.
7:    **end for**
8:    Set $y_{k+1} = y_k^S$.
9:    Sample batches $\mathcal{B}_2, \mathcal{B}_3, \mathcal{B}_4$.
10:   Compute $\widehat{\mathcal{G}}F(x_k)$.
11:   Update $x_{k+1} = \mathrm{Exp}_{x_k}(-\eta_x\widehat{\mathcal{G}}F(x_k))$.
12: **end for**

---

In this section, we consider the bilevel optimization problem (1) in the stochastic setting, where $f(x, y^*(x)) := \frac{1}{n}\sum_{i=1}^{n} f_i(x, y^*(x))$ and $g(x, y) := \frac{1}{m}\sum_{i=1}^{m} g_i(x, y)$. The algorithm for solving the stochastic bilevel optimization problem is in Algorithm 2, where we sample $\mathcal{B}_1, \mathcal{B}_2, \mathcal{B}_3, \mathcal{B}_4$ afresh every iteration. The batch index is omitted for clarity. The batches are sampled uniformly at random with replacement such that the mini-batch gradient is an unbiased estimate of the full gradient. Here, we denote $f_{\mathcal{B}}(x, y) := \frac{1}{|\mathcal{B}|}\sum_{i\in\mathcal{B}} f_i(x, y)$ and similarly for $g$. We let $[n] := \{1, \ldots, n\}$.

In Step 10 of Algorithm 2, we can employ any hypergradient estimator proposed in Section 3.1. In this work, we only show convergence under the Hessian inverse approximation of hypergradient, i.e., $\widehat{\mathcal{G}}F(x_k) = \mathcal{G}_x f_{\mathcal{B}_2}(x_k, y_{k+1}) - \mathcal{G}_{xy}^2 g_{\mathcal{B}_3}(x_k, y_{k+1})[\mathcal{H}_y^{-1}g_{\mathcal{B}_4}(x_k, y_{k+1})[\mathcal{G}_y f_{\mathcal{B}_2}(x_k, y_{k+1})]]$. Similar analysis can be followed for other approximation strategies. The theoretical guarantees are in Theorem 2, where we require Assumption 4, which is common in existing works for analyzing stochastic algorithms on Riemannian manifolds [42, 23, 22].

**Assumption 4.** Under stochastic setting, Assumption 1 holds and Assumptions 2, 3 are satisfied for component functions $f_i(x, y), g_j(x, y)$, for all $i \in [n], j \in [m]$. Further, stochastic gradient, Hessian, and cross derivatives are unbiased estimates.

**Theorem 2.** Under Assumption 4, consider Algorithm 2. Suppose we choose $\eta_x = \frac{1}{20L_F}, S \geq \widetilde{\Theta}(\kappa_l^2\zeta)$, and $|\mathcal{B}_1|, |\mathcal{B}_2|, |\mathcal{B}_3|, |\mathcal{B}_4| \geq \Theta(\kappa_l^2\epsilon^{-1})$ for an arbitrary $\epsilon > 0$. Then we have $\min_{k=0,\ldots,K-1} \mathbb{E}\|\mathcal{G}F(x_k)\|_{x_k}^2 \leq \frac{80L_F\Delta_0}{K} + \frac{\epsilon}{2}$ and the gradient complexity to reach $\epsilon$-stationary solution is $G_f = O(\kappa_l^5\epsilon^{-2}), G_g = \widetilde{O}(\kappa_l^9\zeta\epsilon^{-2}), JV_g = O(\kappa_l^5\epsilon^{-2})$.

In Table 1, we compare our attained complexities with that of stocBiO [40], which makes use of a truncated Neumann series. With exact Hessian inverse, we can match the $G_f$ and $JV_g$ complexities

with stocBio. For the $G_g$ complexity, the additional curvature constant is inevitable from the convergence analysis for geodesic strongly convex functions. Nevertheless we observe the same order dependency on $\kappa_l$. This is mainly due to the analysis where we choose a smaller stepsize $\eta_y = \Theta(\frac{\mu}{L^2})$ compared to $\Theta(\frac{2}{L+\mu})$ in [40]. The larger stepsize, despite increasing the convergence rate, also increases the variance under stochastic setting. We believe an order of $\Theta(\kappa_l)$ lower can be established for stocBio, following our analysis.

### 3.4 Extension to retraction

Our analysis till now has been limited to the use of the exponential map. However, the retraction mapping is often preferred over the exponential map due to its lower computational cost. Here, we show that use of retraction in our algorithms also leads to similar convergence guarantees.

**Assumption 5.** There exist constants $\bar{c} \geq 1, c_R \geq 0$ such that $d^2(x, y) \leq \bar{c}\|u\|_x^2$ and $\|\mathrm{Exp}_x^{-1}(y) - u\|_x \leq c_R\|u\|^2$, for any $x, y = \mathrm{Retr}_x(u) \in \mathcal{U}$.

Assumption 5 is standard (e.g. in [42, 23]) in bounding the error between exponential map and retraction given that retraction is a first-order approximation to the exponential map.

**Theorem 3.** *Suppose Assumptions 1, 2, 3 and 5 hold and let $\widetilde{L}_F = 4\kappa_l c_R M + 5\bar{c}L_F$. Then consider Algorithm 1 with exponential map replaced with general retraction. We can obtain the following bounds.*

- **HINV:** *Let $\eta_x = \Theta(1/\widetilde{L}_F), S \geq \widetilde{\Theta}(\kappa_l^2 \zeta)$. Then $\min_{k=0,\ldots,K-1} \|\mathcal{G}F(x_k)\|_{x_k}^2 \leq 16\widetilde{L}_F \Delta_0/K$.*

- **CG:** *Let $\eta_x = \Theta(1/\widetilde{\Lambda}), S \geq \widetilde{\Theta}(\kappa_l^2 \zeta), T_{\mathrm{cg}} \geq \widetilde{\Theta}(\sqrt{\kappa_l})$, where $\widetilde{\Lambda} = C_v^2 \bar{c} + \kappa_l^2(\frac{5M^2 C_0^2 \bar{D}^2}{\mu} + \bar{c})$. Then*
  $\min_{k=0,\ldots,K-1} \|\mathcal{G}F(x_k)\|_{x_k}^2 \leq \frac{96\widetilde{\Lambda}}{K}(\Delta_0 + \|v_0^*\|_{y^*(x_0)}^2)$.

- **NS:** *Let $\eta_x = \Theta(1/\widetilde{L}_F), S \geq \widetilde{\Theta}(\kappa_l^2 \zeta)$. Then for an arbitrary $\epsilon > 0, T_{\mathrm{ns}} \geq \widetilde{\Theta}(\kappa_l \log(1/\epsilon))$, we have $\min_{k=0,\ldots,K-1} \|\mathcal{G}F(x_k)\|_{x_k}^2 \leq \frac{16\widetilde{L}_F}{K}\Delta_0 + \frac{\epsilon}{2}$.*

- **AD:** *Let $\eta_x = \Theta(1/\widetilde{L}_F), S \geq \widetilde{\Theta}(\kappa_l^2 \zeta \log(1/\epsilon))$. Then for an arbitrary $\epsilon > 0$, we have $\min_{k=0,\ldots,K-1} \|\mathcal{G}F(x_k)\|_{x_k}^2 \leq \frac{16\widetilde{L}_F}{K}\Delta_0 + \frac{\epsilon}{2}$.*

Theorem 3 demonstrates that employing a general retraction preserves the same order of convergence and complexity as the exponential map in Theorem 1. This is due to the fact that $\widetilde{L}_F = \Theta(L_F)$ and $\widetilde{\Lambda} = \Theta(\Lambda)$, where $L_F$ and $\Lambda$ are as defined in Theorem 1. In addition, when exponential map is used, Theorem 3 recovers the results in Theorem 1 as $c_R = 0$ and $\bar{c} = 1$.

## 4 Experiments

This section explores various applications of bilevel optimization problems over manifolds. All the experiments are implemented based on Geoopt [44] and the codes are available at `https://github.com/andyjm3/rhgd`.

### 4.1 Synthetic problem

We consider the following bilevel optimization problem on the Stiefel manifold $\mathrm{St}(d, r) = \{\mathbf{W} \in \mathbb{R}^{d \times r} : \mathbf{W}^\top \mathbf{W} = \mathbf{I}_r\}$ and SPD manifold $\mathbb{S}_{++}^d = \{\mathbf{M} \in \mathbb{R}^{d \times d} : \mathbf{M} \succ 0\}$ (in Appendix A):

$$\max_{\mathbf{W} \in \mathrm{St}(d,r)} \mathrm{tr}(\mathbf{M}^* \mathbf{X}^\top \mathbf{Y} \mathbf{W}^\top), \quad \text{s.t. } \mathbf{M}^* = \arg\min_{\mathbf{M} \in \mathbb{S}_{++}^d} \langle \mathbf{M}, \mathbf{X}^\top \mathbf{X} \rangle + \langle \mathbf{M}^{-1}, \mathbf{W}\mathbf{Y}^\top \mathbf{Y} \mathbf{W}^\top + \nu \mathbf{I} \rangle,$$

where $\mathbf{X} \in \mathbb{R}^{n \times d}, \mathbf{Y} \in \mathbb{R}^{n \times r}$, with $n \geq d \geq r$, are given matrices and $\nu > 0$ is the regularization parameter. The above is a synthetically constructed problem that aims to maximize the similarity between $\mathbf{X}$ and $\mathbf{Y}$ in different feature dimensions. We align $\mathbf{X}$ and $\mathbf{Y}$ to the same dimension via $\mathbf{W} \in \mathrm{St}(d, r)$ and also learn an appropriate geometric metric $\mathbf{M} \in \mathbb{S}_{++}^d$ in the lower-level problem [69]. The geodesic convexity of the lower-level problem and the Hessian inverse expression are discussed in Appendix H.1.

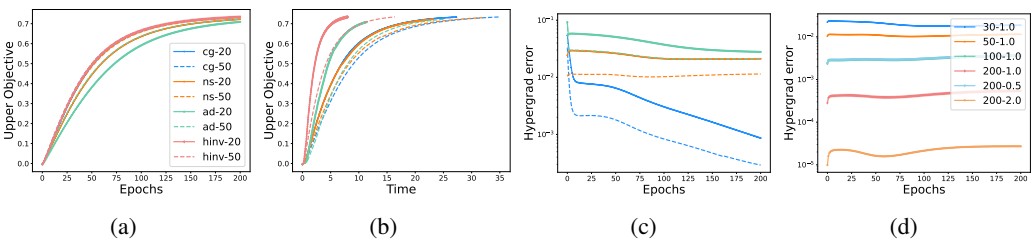

Figure 1: Figures (a) & (b) show the plot of objective of the upper-level problem (Upper Objective) for different strategies. HINV and CG strategies have fastest convergence, followed by NS and AD. The corresponding estimation errors are shown in (c). Figure (d) specifically shows the robustness of approximation error obtained by NS across different $\gamma$ and $T$ values.

**Results.** We generate random data matrices $\mathbf{X}, \mathbf{Y}$ with $n = 100, d = 50$, and $r = 20$. We set $\nu = 0.01$ and fix $\eta_x = \eta_y = 0.5$. We compare the three proposed strategies for approximating the hypergradient where we select $\gamma = 1.0$ and $T_{\text{ns}} = 50$ for Neumann series (NS) and set maximum iterations $T_{\text{cg}}$ for conjugate gradient (CG) to be 50 and break once the residual reaches a tolerance of $10^{-10}$. We set the number of outer iterations (epochs) $K$ to be 200. Figure 1 compares RHGD with different approximation strategies implemented with $S = 20$ or 50 number of inner iterations.

### 4.2 Hyper-representation over SPD manifolds

Hyper-representation [54, 61] aims to solve a regression/classification task while searching for the best representation of the data. It can be formulated as a bilevel optimization problem, where the lower-level optimizes the regression/classification parameters while the upper-level searches for the optimal embedding of the inputs. Suppose we are given a set of SPD matrices, $\mathcal{D} = \{\mathbf{A}_i\}_{i=1}^n$ where $\mathbf{A}_i \in \mathbb{S}_{++}^d$ and the task is to learn a low-dimensional embedding of $\mathbf{A}_i$ while remaining close to their semantics labels. In particular, we partition the set into a training set $\mathcal{D}_{\text{tr}}$ and validation set $\mathcal{D}_{\text{val}}$.

**Shallow hyper-representation for regression.** We consider a shallow learning paradigm over $\mathcal{D}$ through the regression task. The representation is parameterized with $\mathbf{W}^\top \mathbf{A}_i \mathbf{W}$ for $\mathbf{W} \in \text{St}(d, r)$. The requirement of orthogonality on $\mathbf{W}$ follows [38, 30, 33] that ensures the learned representations are SPD. The learned representation is then transformed to a Euclidean space for performing regression, namely through a matrix logarithm (that acts as a bijective map between the space of SPD matrices and symmetric matrices) and a vectorization operation $\text{vec}(\cdot)$ that extract the upper-triangular part of the symmetric matrix. The bilevel optimization problem is

$$\min_{\mathbf{W} \in \text{St}(d,r)} \sum_{i \in \mathcal{D}_{\text{val}}} \frac{(\text{vec}(\text{logm}(\mathbf{W}^\top \mathbf{A}_i \mathbf{W}))\boldsymbol{\beta}^* - y_i)^2}{2|\mathcal{D}_{\text{val}}|},$$
$$\text{s.t. } \boldsymbol{\beta}^* = \arg\min_{\boldsymbol{\beta} \in \mathbb{R}^{r(r+1)/2}} \sum_{i \in \mathcal{D}_{\text{tr}}} \frac{(\text{vec}(\text{logm}(\mathbf{W}^\top \mathbf{A}_i \mathbf{W}))\boldsymbol{\beta} - y_i)^2}{2|\mathcal{D}_{\text{tr}}|} + \frac{\lambda}{2}\|\boldsymbol{\beta}\|^2.$$

The regularization $\lambda > 0$ ensures the lower-level problem is strongly convex. The upper-level problem is on the validation set while the lower-level problem is on the training set. We generate random $\mathbf{W}, \mathbf{A}_i$ and $\boldsymbol{\beta}$ and construct $\mathbf{y}$ with $y_i = \text{vec}(\text{logm}(\mathbf{W}^\top \mathbf{A}_i \mathbf{W}))\boldsymbol{\beta} + \epsilon_i$, where $\epsilon_i \sim \mathcal{N}(0, 1)$. We generate 200 $\mathbf{A}_i$ with $|\mathcal{D}_{\text{val}}| = 100$ and $|\mathcal{D}_{\text{tr}}| = 100$. In Figure 2a, we show the loss on validation set (the upper loss) in terms of number of outer iterations. We compare both the deterministic (RHGD) and stochastic (RSHGD) versions of Riemannian hypergradient descent. We again observe that the best performance is attained by either the ground-truth Hessian inverse or the conjugate gradient. NS requires carefully selecting the hyperparameters $\gamma, T$, which pose difficulties in real applications. For the stochastic versions, all the methods perform similarly.

**Deep hyper-representation for classification.** We now explore a 2-layer SPD network [38] for classifying ETH-80 image set [46]. The dataset consists of 8 classes, each with 10 objects. Each object is represented by an image set consisting of images taken from different viewing angles. Here, we represent each image set by taking the covariance matrix of the images in the same set after resizing them into $10 \times 10$. This results in 80 SPD matrices $\mathbf{A}_i$ of size $100 \times 100$ for classification. Let $\Phi(\mathbf{A}_i) = \text{vec}(\text{logm}(\mathbf{W}_2^\top \text{ReEig}(\mathbf{W}_1^\top \mathbf{A}_i \mathbf{W}_1)\mathbf{W}_2))$ be the output of the 2 layer network where $\text{ReEig}(\mathbf{A}) = \mathbf{U}\max\{\epsilon\mathbf{I}, \boldsymbol{\Sigma}\}\mathbf{U}^\top$ is the eigenvalue rectifying activation with the eigenvectors $\mathbf{U}$ and

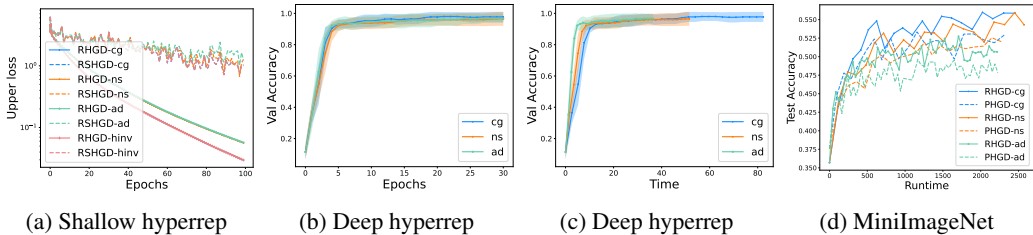

| (a) Shallow hyperrep | (b) Deep hyperrep | (c) Deep hyperrep | (d) MiniImageNet |

Figure 2: Figures (a), (b), and (c) show the performance of RHGD on the hyper-representation problems on SPD networks. Figure (d) shows the good generalization performance of our proposed RHGD algorithms over the projected gradient PHGD baselines on the MiniImageNet dataset.

Table 2: Classification accuracy on the Caltech-Office dataset.

| Methods | A→C | A→D | A→W | C→A | C→D | C→W | D→A | D→C | D→W | W→A | W→C | W→D |
|---|---|---|---|---|---|---|---|---|---|---|---|---|
| OT-EMD | 66.67 | 47.77 | 45.76 | 67.52 | 36.31 | 42.71 | 62.17 | 59.71 | 85.08 | 55.41 | 51.16 | 96.82 |
| OT-SKH | 76.83 | 75.80 | 69.83 | 84.35 | 78.34 | 68.14 | 80.92 | 71.57 | 93.90 | 74.17 | 67.02 | 87.26 |
| Proposed | 78.70 | 80.25 | 69.83 | 88.21 | 80.25 | 68.47 | 82.74 | 75.69 | 97.97 | 83.49 | 73.62 | 98.73 |

eigenvalues $\Sigma$ of $\mathbf{A}$. We consider the same bilevel optimization as above except the least-squares loss function becomes the cross-entropy loss. Here we sample 5 samples from each class to form the training set and the rest as the validation set. We set $d_1 = 20, d_2 = 5$, and fix learning rate to be $0.1$ for both lower and upper problems. Figures 2b and 2c show the good performance on the validation accuracy (upper-level loss).

### 4.3 Riemannian meta learning

Meta learning [16, 34] allows adaptation of models to new tasks with minimal amount of additional data and training, by distilling past learning experiences. A recent work [64] considers meta learning with orthogonality constraint. In particular, the upper-level optimization searches for the base parameters shared by all tasks while the lower level optimizes over the task-specific parameters to ensure generalization ability. Let $P_{\mathcal{T}}$ denote the distribution of meta tasks and for each training epoch, we sample $m$ tasks $\mathcal{D}^\ell \sim P_{\mathcal{T}}, \ell = 1, ..., m$. Each task is composed of a support and query set denoted by $\mathcal{D}^\ell_{\mathrm{s}}, \mathcal{D}^\ell_{\mathrm{q}}$, and the task is to learn a set of base parameters $\Theta$ such that the model can quickly adapt to the query set from the support set by adjusting only a few parameters $w$. For each task, the task-specific parameter $w^*_\ell$ is learned from the support set, which is used to update the base parameters by minimizing the loss over the query set. In standard settings, $w_\ell$ corresponds to the final linear layer of a neural network [39, 40]. Here, we adopt the setup with $w_\ell$ to be the last layer parameters in the Euclidean space while enforcing $\Theta$ on the Stiefel manifold. The problem of Riemannian meta-learning is $\min_{\Theta \in \mathrm{St}} \frac{1}{m} \sum_{\ell=1}^m \mathcal{L}(\Theta, w^*_\ell; \mathcal{D}^\ell_{\mathrm{q}})$ s.t. $w^*_\ell = \arg\min_{w_\ell} \frac{1}{m} \sum_{\ell=1}^m \mathcal{L}(\Theta, w_\ell; \mathcal{D}^\ell_{\mathrm{s}}) + \mathcal{R}(w_\ell)$, where $\mathcal{D}^\ell_{\mathrm{s}}, \mathcal{D}^\ell_{\mathrm{q}}$ are the support and query sets for task $\ell$ and $\mathcal{R}(\cdot)$ is a regularizer that ensures strong convexity of the lower-level problem.

**Results.** We consider 5-ways 5-shots meta learning over the MiniImageNet dataset [59] where the backbone network is a 4-block CNN with the kernel of the first 2 layers constrained to be orthogonal in terms of the output channel (following [48]). The kernel size is $3 \times 3$ and we consider 16 output channels with a padding of $1$. Each convolutional block consists of a convolutional layer, followed by a ReLU activation, a max-pooling and a batch normalization layer. $\Theta$, thus, has the dimension $(16 * 3 * 3) \times 16 = 144 \times 16$, which is constrained to the Stiefel manifold.

In Figure 2d, we plot the test accuracy averaged for over 200 tasks. We compare RHGD with an extrinsic update baseline PHGD, which projects the update from the Euclidean space to the Stiefel manifold at every iteration. We observe the RHGD converges faster compared to the extrinsic update PHGD, thereby showing the benefit of the Riemannian modeling.

### 4.4 Unsupervised domain adaptation

Given two marginals $\boldsymbol{\mu} \in \mathbb{R}^n, \boldsymbol{\nu} \in \mathbb{R}^m$ with equal total mass, i.e., $\boldsymbol{\mu}^\top \mathbf{1}_n = \boldsymbol{\nu}^\top \mathbf{1}_m = 1$ where we assume unit mass without loss of generality. Let $\Pi(\boldsymbol{\mu}, \boldsymbol{\nu}) := \{ \boldsymbol{\Gamma} \in \mathbb{R}^{n \times m} : \boldsymbol{\Gamma} > 0, \boldsymbol{\Gamma}\mathbf{1}_m =$

$\boldsymbol{\mu}, \boldsymbol{\Gamma}^\top \mathbf{1}_n = \boldsymbol{\nu}\}$ be the set of doubly stochastic matrices with strictly positive entries. From [15, 57], it is known that the set forms a Riemannian manifold with the Fisher metric.

Given a supervised source dataset $\mathbf{X} \in \mathbb{R}^{n \times d}$ and an unsupervised target dataset $\mathbf{Y} \in \mathbb{R}^{m \times d}$ $(n, m \geq d)$, we consider the unsupervised domain adaptation problem to classify target domain instances. Using the optimal transport framework [58, 12], we pose this as a bilevel problem:

$$\min_{\boldsymbol{\Gamma} \in \Pi(\boldsymbol{\mu}, \boldsymbol{\nu})} \langle \boldsymbol{\Gamma}, \mathcal{C}(\mathbf{X}\mathbf{M}^{*-1/2}, \mathbf{Y}\mathbf{M}^{*-1/2}) \rangle - \lambda H(\boldsymbol{\Gamma}),$$
$$\text{s.t. } \mathbf{M}^* = \arg\min_{\mathbf{M} \in \mathbb{S}_{++}^d} \alpha \text{dist}^2(\mathbf{M}, \mathbf{X}^\top\mathbf{X}) + (1-\alpha)\text{dist}^2(\mathbf{M}, \mathbf{Y}^\top\boldsymbol{\Gamma}^\top\boldsymbol{\Gamma}\mathbf{Y}), \tag{3}$$

where $H(\boldsymbol{\Gamma}) = -\langle \boldsymbol{\Gamma}, \log \boldsymbol{\Gamma} \rangle$ is the entropy regularization, $\mathcal{C}(\mathbf{X}, \mathbf{Y}) = \text{diag}(\mathbf{X}\mathbf{X}^\top)\mathbf{1}_m^\top + \mathbf{1}_n\text{diag}(\mathbf{Y}\mathbf{Y}^\top) - 2\mathbf{X}\mathbf{Y}^\top$ is the pairwise squared distance matrix, and $\Pi$ denotes the doubly stochastic manifold [15]. Here, dist is the geodesic distance between SPD matrices and $\alpha \in [0, 1]$.

The lower-level problem in (3) finds $\mathbf{M}^*$ which is the weighted geometric mean between $\mathbf{X}^\top\mathbf{X}$ and $\mathbf{Y}^\top\boldsymbol{\Gamma}^\top\boldsymbol{\Gamma}\mathbf{Y}$ [6]. Conceptually, learning of $\mathbf{M}$ allows to align the features of the source and target instances. The upper-level problem, on the other hand, minimizes the Mahalanobis distance between the source and target domain instances parameterized by $\mathbf{M}^{*-1}$. An interpretation is that the matrix $\mathbf{M}^*$ leads to whitening of the data (i.e., $\mathbf{X}\mathbf{M}^{*-1/2}$ is the whitened data) [5].

After the transport plan $\boldsymbol{\Gamma}^*$ is learned, we employ barycentric projection using $\boldsymbol{\Gamma}^*$ to transport the source points to the target domain and and employ the nearest neighbour (1-NN) classifier for the target dataset classification . For barycentric projection, we project the source samples $\mathbf{X}$ to the target $\mathbf{Y}$ by solving $\mathbf{x}_i = \arg\min_{\mathbf{x}_i \in \mathbb{R}^d} \sum_{j=1}^m \boldsymbol{\Gamma}_{i,j}^* \|\mathbf{M}^{*-1/2}\mathbf{x}_i - \mathbf{M}^{*-1/2}\mathbf{y}_j\|^2 = \mu_i^{-1}(\sum_{j=1}^m \boldsymbol{\Gamma}_{i,j}^*\mathbf{y}_j)$. Then, a nearest-neighbour (NN) classifier is used to classify the samples in the target given the source labels based on the distance computed with $\mathbf{M}^*$, i.e., $\mathcal{C}(\mathbf{X}\mathbf{M}^{*-1/2}, \mathbf{Y}\mathbf{M}^{*-1/2})$.

**Results.** We consider the Caltech-Office dataset [20], which is commonly used for domain adaptation. The dataset contains images from four domains in ten classes, i.e., Amazon (A), the Caltech image dataset (C), DSLR (D), and Webcam (W), each with containing 958, 1123, 157, and 295 samples respectively. Hence, there are 12 domain adaptation tasks, e.g., A→D implies A is the source and D is the target. Each domain has the same ten classes. The goal is to classify images from target domain given source domain. For preprocessing, we normalize the samples to have unit norm and reduce the dimensionality to 128 by mean pooling every 64 columns.

We compare our proposed bilevel approach (3) with single-level optimal transport baselines, i.e., solving $\min_{\boldsymbol{\Gamma} \in \Pi(\boldsymbol{\mu}, \boldsymbol{\nu})} \langle \boldsymbol{\Gamma}, \mathcal{C}(\mathbf{X}, \mathbf{Y}) \rangle - \lambda H(\boldsymbol{\Gamma})$, followed by the same barycentric projection. Specifically, the baselines are: (1) optimal transport where $\lambda = 0$ (labelled as OT-EMD) and (2) optimal transport with the Sinkhorn algorithm (labelled as OT-SKH). OT-EMD employs the earth mover distance while OT-SKH employs the Sinkhorn distance [13]. We implement the two OT baselines with the POT Python library [17]. $\lambda$ is tuned in OT-SKH for each source-target pair. The best validation results are obtained by setting $\lambda = 5 \times 10^{-3}$ for all the problem pairs except the W→D pair for which $\lambda = 10^{-3}$ gives the best result. For our proposed bilevel approach, we set $\lambda = 0$ and $\alpha = 0.5$.

In Table 2, we observe that the proposed bilevel approach obtains better generalization performance than the baselines across all the tasks. This showcases the utility of learning the whitening metric $\mathbf{M}^{-1}$ in a bilevel setting.

## 5  Conclusion

In this work, we have proposed a framework for tackling bilevel optimization over Riemannian manifolds. We discussed various hypergradient approximation strategies (conjugate gradients, truncated Neumann series, and automatic differentiation) and provide error bounds. Our proposed algorithms rely only on gradient updates and make use of retraction which scale well across problems. We illustrate the efficacy of the proposal approach in several machine learning applications.

Although in this work, we focus on geodesic strongly convex lower-level problems, our framework can be extended to relax such assumption to (geodesic) convexity with an extra strongly convex regularizer [2], or to (Riemannian) PL condition where a global minimizer exists [9]. Furthermore, we believe there is potential to improve the current results in stochastic bilevel optimization by reducing the strict requirements on batch size. Additionally, the dependency on the curvature constant could also be further optimized.

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

# Appendix

## A  Riemannian geometries of considered manifolds

**Symmetric positive definite (SPD) manifold.** The SPD manifold of size $d$ is denoted as $\mathbb{S}_{++}^d \coloneqq \{\mathbf{X} \in \mathbb{R}^{d \times d} : \mathbf{X}^\top = \mathbf{X}, \mathbf{X} \succ 0\}$ and the commonly considered Riemannian metric is the affine-invariant metric $\langle \mathbf{U}, \mathbf{V} \rangle_{\mathbf{X}} = \operatorname{tr}(\mathbf{X}^{-1}\mathbf{U}\mathbf{X}^{-1}\mathbf{V})$ [6], for $\mathbf{U}, \mathbf{V} \in T_{\mathbf{X}}\mathbb{S}_{++}^d$. Other Riemannian metrics, such as (generalized) Bures-Wasserstein [55, 24], log-Euclidean [4] and log-Cholesky [51] metrics can also be considered. The exponential map is given by $\operatorname{Exp}_{\mathbf{X}}(\mathbf{U}) = \mathbf{X}\operatorname{expm}(\mathbf{X}^{-1}\mathbf{U})$ where $\operatorname{expm}(\cdot)$ denotes the principal matrix exponential. The corresponding logarithm map is given by $\log_{\mathbf{X}}(\mathbf{Y}) = \mathbf{X}\operatorname{logm}(\mathbf{X}^{-1}\mathbf{Y})$. Its Riemannian gradient of a real-valued function $f$ is derived as $\operatorname{grad}f(\mathbf{X}) = \mathbf{X}\nabla f(\mathbf{X})\mathbf{X}$ and the Riemannian Hessian is $\operatorname{Hess}f(\mathbf{X})[\mathbf{U}] = \operatorname{Dgrad}f(\mathbf{X})[\mathbf{U}] - \{\mathbf{U}\mathbf{X}^{-1}\operatorname{grad}f(\mathbf{X})\}_{\mathrm{S}} = \mathbf{X}\nabla^2 f(\mathbf{X})[\mathbf{U}]\mathbf{X} + \{\mathbf{U}\nabla f(\mathbf{X})\mathbf{X}\}_{\mathrm{S}}$ where we use $\{\mathbf{A}\}_{\mathrm{S}} \coloneqq (\mathbf{A} + \mathbf{A}^\top)/2$.

**Stiefel manifold.** The Stiefel manifold is the set of orthonormal matrices, i.e., $\operatorname{St}(d, r) \coloneqq \{\mathbf{X} \in \mathbb{R}^{d \times r} : \mathbf{X}^\top \mathbf{X} = \mathbf{I}\}$. A common Riemannian metric is the Euclidean inner product. We consider the QR-based retraction in the experiment, which is $\operatorname{Retr}_{\mathbf{X}}(\mathbf{U}) = \operatorname{qf}(\mathbf{X} + \mathbf{U})$ where $\operatorname{qf}(\cdot)$ extracts the Q-factor from the QR decomposition. Let the orthogonal projection to the tangent space be denoted as $\operatorname{P}_{\mathbf{X}}(\mathbf{U}) = \mathbf{U} - \mathbf{X}\{\mathbf{X}^\top \mathbf{U}\}_{\mathrm{S}}$. Then, the Riemannian gradient and Riemannian Hessian are given by $\operatorname{grad}f(\mathbf{X}) = \operatorname{P}_{\mathbf{X}}(\nabla f(\mathbf{X}))$ and $\operatorname{Hess}f(\mathbf{X})[\mathbf{U}] = \operatorname{P}_{\mathbf{X}}(\nabla^2 f(\mathbf{X})[\mathbf{U}] - \mathbf{U}\{\mathbf{X}^\top \nabla f(\mathbf{X})\}_{\mathrm{S}})$.

**Doubly stochastic manifold.** The doubly stochastic manifold (or coupling manifold) between two discrete probability measures $\mu, \nu$ with marginals $\mathbf{a} \in \mathbb{R}^m, \mathbf{b} \in \mathbb{R}^n$ is the set $\Pi(\mu, \nu) = \{\mathbf{\Gamma} \in \mathbb{R}^{m \times n} : \Gamma_{ij} > 0, \mathbf{\Gamma}\mathbf{1}_n = \mathbf{a}, \mathbf{\Gamma}^\top \mathbf{1}_m = \mathbf{b}\}$. It can be equipped with the Fisher information metric, defined as $\langle \mathbf{U}, \mathbf{V} \rangle_{\mathbf{\Gamma}} = \sum_{i,j}(U_{ij}V_{ij})/\Gamma_{ij}$ for any $\mathbf{U}, \mathbf{V} \in T_{\mathbf{\Gamma}}\Pi(\mu, \nu)$. The retraction is given by $\operatorname{Retr}_{\mathbf{\Gamma}}(\mathbf{U}) = \operatorname{Sinkhorn}(\mathbf{\Gamma} \odot \exp(\mathbf{U} \oslash \mathbf{\Gamma}))$ where $\exp, \odot, \oslash$ are elementwise exponential, product, and division operations. $\operatorname{Sinkhorn}(\cdot)$ represents the Sinkhorn-Knopp iterations for balancing a matrix [43].

## B  Important Lemmas

**Proposition 2** ([8]). *In a totally normal neighbourhood $\mathcal{U} \subseteq \mathcal{M}$, a function $f : \mathcal{U} \to \mathbb{R}$ is $\mu$-geodesic strongly convex, then it satisfies for all $x, y \in \mathcal{U}$*

$$f(y) \geq f(x) + \langle \operatorname{grad}f(x), \operatorname{Exp}_x^{-1}(y)\rangle_x + \frac{\mu}{2}d^2(x, y).$$

*If a function $f$ has $L$-Lipschitz Riemannian gradient, then it satisfies for all $x, y \in \mathcal{U}$*

$$f(y) \leq f(x) + \langle \operatorname{grad}f(x), \operatorname{Exp}_x^{-1}(y)\rangle_x + \frac{L}{2}d^2(x, y).$$

**Lemma 2** ([63, 26]). *There exists a constant $C_0 > 0$ such that for any $y_1, y_2, y_3 \in \mathcal{U}_y$, $u \in T_{y_1}\mathcal{M}_y$, $\|\Gamma_{y_2}^{y_3}\Gamma_{y_1}^{y_2}u - \Gamma_{y_1}^{y_3}u\| \leq C_0 d(y_1, y_2)d(y_2, y_3)\|u\|_{y_1}$*

**Lemma 3** (Trigonometric distance bound [72, 71, 28]). *Let $x_a, x_b, x_c \in \mathcal{U} \subseteq \mathcal{M}$ and denote $a = d(x_b, x_c)$, $b = d(x_a, x_c)$ and $c = d(x_a, x_b)$ as the geodesic side lengths. Then,*

$$a^2 \leq \zeta b^2 + c^2 - 2\langle \operatorname{Exp}_{x_a}^{-1}(x_b), \operatorname{Exp}_{x_a}^{-1}(x_c)\rangle_{x_a}$$

*where $\zeta = \frac{\sqrt{|\kappa^-|}\bar{D}}{\tanh(\sqrt{|\kappa^-|}\bar{D})}$ if $\kappa^- < 0$ and $\zeta = 1$ if $\kappa^- \geq 0$. Here, $\bar{D}$ denotes the diameter of $\mathcal{U}$ and $\kappa^-$ denotes the lower bound of the sectional curvature of $\mathcal{U}$.*

## C  Proofs for Section 3.1

### C.1  Proof of Proposition 1

*Proof of Proposition 1.* By the first-order optimality condition, $y^*(x)$ satisfies $\mathcal{G}_y g(x, y^*(x)) = 0 \in T_{y^*(x)}\mathcal{M}_y$. Based on Theorem 5 in [25], taking the (implicit) derivative of the equality with respect to $x$ yields $\mathcal{G}_{yx}^2 g(x, y^*(x))[u] + \mathcal{H}_y g(x, y^*(x))[\operatorname{D}y^*(x)[u]] = 0$ for any $u \in T_x\mathcal{M}_x$. This gives

$\mathrm{D}y^*(x) = -\mathcal{H}_y^{-1}g(x, y^*(x)) \circ \mathcal{G}_{yx}^2 g(x, y^*(x))$. Notice that $\mathrm{D}y^*(x) : T_x\mathcal{M}_x \to T_{y^*(x)}\mathcal{M}_y(x)$, its adjoint operator $(\mathrm{D}y^*(x))^\dagger$ is derived as follows. For any $u \in T_x\mathcal{M}_x$, $v \in T_{y^*(x)}\mathcal{M}_y$

$$
\begin{aligned}
\langle (\mathrm{D}y^*(x))^\dagger[v], u \rangle_x = \langle \mathrm{D}y^*(x)[u], v \rangle_{y^*(x)} &= -\big\langle \big(\mathcal{H}_y^{-1}g(x, y^*(x)) \circ \mathcal{G}_{yx}^2 g(x, y^*(x))\big)[u], v \big\rangle_{y^*(x)} \\
&= -\big\langle \mathcal{G}_{yx}^2 g(x, y^*(x))[u], \mathcal{H}_y^{-1}g(x, y^*(x))[v] \big\rangle_{y^*(x)} \\
&= -\big\langle \big(\mathcal{G}_{xy}^2 g(x, y^*(x)) \circ \mathcal{H}_y^{-1}g(x, y^*(x))\big)[v], u \big\rangle_x
\end{aligned}
$$

where the first equality uses the definition of adjoint operator and the third equality is due to the self-adjointness of Riemannian Hessian (inverse) and the last equality is due to Proposition D.2 in [27] that $\mathcal{G}_{xy}^2 g$ and $\mathcal{G}_{yx}^2 g$ are adjoint operators. By identification, we have $(\mathrm{D}y^*(x))^\dagger = -\mathcal{G}_{xy}^2 g(x, y^*(x)) \circ \mathcal{H}_y^{-1}g(x, y^*(x))$.

Finally by the chain rule, we obtain (from the definition of Riemannian gradient), for any $u \in T_x\mathcal{M}_x$

$$
\begin{aligned}
\langle \mathcal{G}F(x), u \rangle_x &= \langle \mathcal{G}_x f(x, y^*(x)), u \rangle_x + \mathrm{D}_y f(x, y^*(x))[\mathrm{D}y^*(x)[u]] \\
&= \langle \mathcal{G}_x f(x, y^*(x)), u \rangle_x + \langle \mathcal{G}_y f(x, y^*(x)), \mathrm{D}y^*(x)[u] \rangle_y \\
&= \langle \mathcal{G}_x f(x, y^*(x)), u \rangle_x + \langle (\mathrm{D}y^*(x))^\dagger \mathcal{G}_y f(x, y^*(x)), u \rangle_x \\
&= \langle \mathcal{G}_x f(x, y^*(x)) - \mathcal{G}_{xy}^2 g(x, y^*(x))[\mathcal{H}_y^{-1}g(x, y^*(x))[\mathcal{G}_y f(x, y^*(x))]], u \rangle_x.
\end{aligned}
$$

By identification the proof is complete. $\qquad\square$

## C.2 On Lipschitzness of gradients

**Proposition 3.** *If a bifunction $f(x, y)$ has $L$-Lipschitz Riemannian gradient, then it satisfies $\|\mathcal{G}f(z_1) - \Gamma_{z_2}^{z_1}\mathcal{G}f(z_2)\|_{z_1} \le 2Ld(z_1, z_2)$, where we let $z = (x, y)$. If an operator $\mathcal{G}(x, y) : T_y\mathcal{M}_y \to T_x\mathcal{M}_x$ is $\rho$-Lipschitz, then it satisfies $\|\mathcal{G}(z_1) - \Gamma_{x_2}^{x_1}\mathcal{G}(z_2)\Gamma_{y_1}^{y_2}\|_{x_1} \le \rho\, d(z_1, z_2)$. If an operator $\mathcal{H}(x, y) : T_y\mathcal{M}_y \to T_x\mathcal{M}_x$ is $\rho$-Lipschitz, then it satisfies $\|\mathcal{H}(z_1) - \Gamma_{y_2}^{y_1}\mathcal{H}(z_2)\Gamma_{y_1}^{y_2}\|_{y_1} \le \rho\, d(z_1, z_2)$.*

*Proof of Proposition 3.* From the definition of Riemannian gradient of product manifold we have

$$
\begin{aligned}
\|\mathcal{G}f(z_1) - \Gamma_{z_2}^{z_1}\mathcal{G}f(z_2)\|_{z_1} &= \|\mathcal{G}_x f(x_1, y_1) - \Gamma_{x_2}^{x_1}\mathcal{G}_x f(x_2, y_2)\|_{x_1} + \|\mathcal{G}_y f(x_1, y_1) - \Gamma_{y_2}^{y_1}\mathcal{G}_y f(x_2, y_2)\|_{y_1} \\
&\le \|\mathcal{G}_x f(x_1, y_1) - \mathcal{G}_x f(x_1, y_2)\|_{x_1} + \|\mathcal{G}_x f(x_1, y_2) - \Gamma_{x_2}^{x_1}\mathcal{G}_x f(x_2, y_2)\|_{x_1} \\
&\quad + \|\mathcal{G}_y f(x_1, y_1) - \mathcal{G}_y f(x_2, y_1)\|_{y_1} + \|\mathcal{G}_y f(x_2, y_1) - \Gamma_{y_2}^{y_1}f(x_2, y_2)\|_{y_1} \\
&\le Ld(y_1, y_2) + Ld(x_1, x_2) + Ld(x_1, x_2) + Ld(y_1, y_2) = 2Ld(z_1, z_2)
\end{aligned}
$$

where we use triangle inequality of Riemannian norm.

Similarly, for the other two claims, we verify

$$
\begin{aligned}
\|\mathcal{G}(z_1) - \Gamma_{x_2}^{x_1}\mathcal{G}(z_2)\Gamma_{y_1}^{y_2}\|_{z_1} &= \|\mathcal{G}(x_1, y_1) - \mathcal{G}(x_1, y_2)\Gamma_{y_1}^{y_2}\|_{x_1} + \|\mathcal{G}(x_1, y_2)\Gamma_{y_1}^{y_2} - \Gamma_{x_2}^{x_1}\mathcal{G}(x_2, y_2)\Gamma_{y_1}^{y_2}\|_{x_1} \\
&\le \rho d(y_1, y_2) + \rho d(x_1, x_2) = \rho d(z_1, z_2).
\end{aligned}
$$

The same arguments also hold for $\mathcal{H}(x, y)$ and hence the proof is omitted. $\qquad\square$

## C.3 Boundedness of ingredients

**Lemma 4.** *Under Assumptions 1, 2, 3, we can show*

4.1 $\|\mathcal{G}_{yx}^2 g(x, y)\|_y = \|\mathcal{G}_{xy}^2 g(x, y)\|_x \le L$ *holds for any* $(x, y) \in \mathcal{U}_x \times \mathcal{U}_y$.

4.2 $\|\mathrm{D}y^*(x)\|_{y^*(x)} \le \kappa_l$ *and* $\|\mathrm{D}y^*(x_1) - \Gamma_{y^*(x_2)}^{y^*(x_1)}\mathrm{D}y^*(x_2)\Gamma_{x_1}^{x_2}\|_{y^*(x_1)} \le L_y d(x_1, x_2)$, *for any* $x, x_1, x_2 \in \mathcal{U}_x$, *where we let* $L_y := \kappa_l^2\kappa_\rho + 2\kappa_l\kappa_\rho + \kappa_\rho$.

4.3 $d(y^*(x_1), y^*(x_2)) \le \kappa_l d(x_1, x_2)$, *for any* $x_1, x_2 \in \mathcal{U}_x$

4.4 *For any* $x, x_1, x_2 \in \mathcal{U}_x, y, y_1, y_2 \in \mathcal{U}_y$

$$
\|\Gamma_{y_1}^{y_2}\mathcal{H}_y^{-1}g(x, y_1)\Gamma_{y_2}^{y_1} - \mathcal{H}_y^{-1}g(x, y_2)\|_{y_2} \le \frac{\kappa_\rho}{\mu}d(y_1, y_2),
$$

$$
\|\mathcal{H}^{-1}g(x_1, y) - \mathcal{H}^{-1}g(x_2, y)\|_y \le \frac{\kappa_\rho}{\mu}d(x_1, x_2).
$$

**4.5** Let $L_F := (\kappa_l + 1)(L + \kappa_\rho M + \kappa_\rho \kappa_l M + \kappa_l L)$. Then for any $x_1, x_2 \in \mathcal{U}_x$, $\|\Gamma_{x_1}^{x_2}\mathcal{G}F(x_1) - \mathcal{G}F(x_2)\|_{x_2} \leq L_F d(x_1, x_2)$.

*Proof of Lemma 4.* (4.1) First we have for any $v \in T_y \mathcal{M}_y$

$$\|\mathcal{G}^2_{xy}g(x,y)[v]\|_x = \|\mathrm{D}_y \mathcal{G}_x g(x,y)[v]\|_x \leq \lim_{t \to 0} \frac{\|\mathcal{G}_x g(x, \mathrm{Exp}_y(tv)) - \mathcal{G}_x g(x,y)\|_x}{|t|} \leq \lim_{t \to 0} \frac{L\|tv\|_y}{|t|} = L\|v\|_y,$$

where we use the fact that $d(\mathrm{Exp}_y(\xi), y) = \|\xi\|_y$. The operator norm is the same between $\mathcal{G}^2_{xy}g(x,y)$ and $\mathcal{G}^2_{yx}g(x,y)$ is due to the adjointness. This proves the first claim.

(4.2) We first verify $\mathrm{D}y^*(x)$ can be bounded as

$$\|\mathrm{D}y^*(x)\|_{y^*(x)} = \|\mathcal{H}_y^{-1}g(x, y^*(x))\|_{y^*(x)}\|\mathcal{G}^2_{yx}g(x, y^*(x))\|_{y^*(x)} \leq \frac{L}{\mu},$$

and $\mathrm{D}y^*(x)$ is also Lipschitz as

$$\|\mathrm{D}y^*(x_1) - \Gamma_{y^*(x_2)}^{y^*(x_1)}\mathrm{D}y^*(x_2)\Gamma_{x_1}^{x_2}\|_{y^*(x_1)}$$

$$\leq \|\mathcal{H}_y^{-1}g(x_1, y^*(x_1)) - \Gamma_{y^*(x_2)}^{y^*(x_1)}\mathcal{H}_y^{-1}g(x_2, y^*(x_2))\Gamma_{y^*(x_1)}^{y^*(x_2)}\|_{y^*(x_1)}\|\mathcal{G}^2_{yx}g(x_1, y^*(x_1))\|_{y^*(x_1)}$$

$$\quad + \|\mathcal{H}_y^{-1}g(x_2, y^*(x_2))\|_{x_2}\|\Gamma_{y^*(x_1)}^{y^*(x_2)}\mathcal{G}^2_{yx}g(x_1, y^*(x_1)) - \mathcal{G}^2_{yx}g(x_2, y^*(x_2))\Gamma_{x_1}^{x_2}\|_{y^*(x_2)}$$

$$\leq L\|\mathcal{H}_y^{-1}g(x_1, y^*(x_1)) - \mathcal{H}_y^{-1}g(x_2, y^*(x_1))\|_{y^*(x_1)} + L\|\mathcal{H}_y^{-1}g(x_2, y^*(x_1)) - \Gamma_{y^*(x_2)}^{y^*(x_1)}\mathcal{H}_y^{-1}g(x_2, y^*(x_2))\Gamma_{y^*(x_1)}^{y^*(x_2)}\|_{y^*(x_1)}$$

$$\quad + \frac{1}{\mu}\|\mathcal{G}^2_{yx}g(x_1, y^*(x_1)) - \mathcal{G}^2_{yx}g(x_2, y^*(x_1))\Gamma_{x_1}^{x_2}\|_{y^*(x_1)} + \frac{1}{\mu}\|\Gamma_{y^*(x_1)}^{y^*(x_2)}\mathcal{G}^2_{yx}g(x_2, y^*(x_1)) - \mathcal{G}^2_{yx}g(x_2, y^*(x_2))\|_{y^*(x_1)}$$

$$\leq \frac{L\rho}{\mu^2}d(x_1, x_2) + \frac{L\rho}{\mu^2}d(y^*(x_1), y^*(x_2)) + \frac{\rho}{\mu}d(x_1, x_2) + \frac{\rho}{\mu}d(y^*(x_1), y^*(x_2))$$

$$\leq \left(\frac{L^2\rho}{\mu^3} + \frac{2L\rho}{\mu^2} + \frac{\rho}{\mu}\right)d(x_1, x_2).$$

(4.3) Now suppose we let $c : [0,1] \to \mathcal{M}_y$, defined as $c(t) := y^*(\gamma(t))$ where $\gamma : [0,1] \to \mathcal{M}_x$ is a geodesic that connects $x_1, x_2$, i.e., $\gamma(0) = x_1, \gamma(1) = x_2$. Then

$$d(y^*(x_1), y^*(x_2)) = \int_0^1 \|c'(t)\|_{c(t)}dt = \int_0^1 \|\mathrm{D}y^*(\gamma(t))[\gamma'(t)]\|_{c(t)}dt \leq \frac{L}{\mu}\int_0^1 \|\gamma'(t)\|_{\gamma(t)}dt = \frac{L}{\mu}d(x_1, x_2),$$

where we use the fact that the manifold is complete.

(4.4) For the second claim, we first notice for any (invertible) linear operators $A, B$, $A^{-1} - B^{-1} = A^{-1}(B-A)B^{-1}$ and thus $\|A^{-1} - B^{-1}\| \leq \|A^{-1}\|\|A-B\|\|B^{-1}\|$ for some well-defined norm $\|\cdot\|$. Here substituting $A = \Gamma_{y_1}^{y_2}\mathcal{H}_y g(x, y_1)\Gamma_{y_2}^{y_1}$, $B = \mathcal{H}_y g(x, y_2)$, we have

$$\|\Gamma_{y_1}^{y_2}\mathcal{H}_y^{-1}g(x, y_1)\Gamma_{y_2}^{y_1} - \mathcal{H}_y^{-1}g(x, y_2)\|_{y_2}$$

$$= \|\mathcal{H}_y^{-1}g(x, y_1)\|_{y_2}\|\Gamma_{y_1}^{y_2}\mathcal{H}_y g(x, y_1)\Gamma_{y_2}^{y_1} - \mathcal{H}_y g(x, y_2)\|_{y_2}\|\mathcal{H}_y^{-1}g(x, y_2)\|_{y_2}$$

$$\leq \frac{\rho}{\mu^2}d(y_1, y_2),$$

where we notice $(\Gamma_{y_1}^{y_2}\mathcal{H}_y g(x, y_1)\Gamma_{y_2}^{y_1})^{-1} = \Gamma_{y_1}^{y_2}\mathcal{H}_y g(x, y_1)^{-1}\Gamma_{y_2}^{y_1}$ and use the isometry property of parallel transport. The same argument applies for $\|\mathcal{H}^{-1}g(x_1, y) - \mathcal{H}^{-1}g(x_2, y)\|_y$.

(4.5) we have

$$\|\Gamma_{x_1}^{x_2}\mathcal{G}F(x_1) - \mathcal{G}F(x_2)\|_{x_2}$$

$$\leq \|\Gamma_{x_1}^{x_2}\mathcal{G}_x f(x_1, y^*(x_1)) - \mathcal{G}_x f(x_2, y^*(x_2))\|_{x_2}$$

$$+ \|\Gamma_{x_1}^{x_2}\mathcal{G}^2_{xy}g(x_1, y^*(x_1)) - \mathcal{G}^2_{xy}g(x_2, y^*(x_2))\Gamma_{y^*(x_1)}^{y^*(x_2)}\|_{x_2}\|\mathcal{H}_y^{-1}g(x_1, y^*(x_1))\|_{y^*(x_1)}\|\mathcal{G}_y f(x_1, y^*(x_1))\|_{y^*(x_1)}$$

$$+ \|\mathcal{G}^2_{xy}g(x_2, y^*(x_2))\|_{x_2}\|\Gamma_{y^*(x_1)}^{y^*(x_2)}\mathcal{H}_y^{-1}g(x_1, y^*(x_1)) - \mathcal{H}_y^{-1}g(x_2, y^*(x_2))\Gamma_{y^*(x_1)}^{y^*(x_2)}\|_{y^*(x_2)}\|\mathcal{G}_y f(x_1, y^*(x_1))\|_{y^*(x_1)}$$

$$+ \|\mathcal{G}^2_{xy}g(x_2, y^*(x_2))\|_{x_2}\|\mathcal{H}_y^{-1}g(x_2, y^*(x_2))\|_{y^*(x_2)}\|\Gamma_{y^*(x_1)}^{y^*(x_2)}\mathcal{G}_y f(x_1, y^*(x_1)) - \mathcal{G}_y f(x_2, y^*(x_2))\|_{y^*(x_2)}.$$

From Assumption 2, 3 and Lemma 4, we can obtain

$$\|\Gamma_{x_1}^{x_2}\mathcal{G}_x f(x_1, y^*(x_1)) - \mathcal{G}_x f(x_2, y^*(x_2))\|_{x_2}$$
$$\leq \|\mathcal{G}_x f(x_1, y^*(x_1)) - \mathcal{G}_x f(x_1, y^*(x_2))\|_{x_1} + \|\Gamma_{x_1}^{x_2}\mathcal{G}_x f(x_1, y^*(x_2)) - \mathcal{G}_x f(x_2, y^*(x_2))\|_{x_2}$$
$$\leq Ld(y^*(x_1), y^*(x_2)) + Ld(x_1, x_2) = \left(\frac{L^2}{\mu} + L\right)d(x_1, x_2).$$

$$\|\Gamma_{y^*(x_1)}^{y^*(x_2)}\mathcal{G}_y f(x_1, y^*(x_1)) - \mathcal{G}_y f(x_2, y^*(x_2))\|_{y^*(x_2)}$$
$$\leq \|\Gamma_{y^*(x_1)}^{y^*(x_2)}\mathcal{G}_y f(x_1, y^*(x_1)) - \mathcal{G}_y f(x_1, y^*(x_2))\|_{y^*(x_2)} + \|\mathcal{G}_y f(x_1, y^*(x_2)) - \mathcal{G}_y f(x_2, y^*(x_2))\|_{y^*(x_2)}$$
$$\leq \left(\frac{L^2}{\mu} + L\right)d(x_1, x_2).$$

Similarly, we have

$$\|\Gamma_{x_1}^{x_2}\mathcal{G}_{xy}^2 g(x_1, y^*(x_1)) - \mathcal{G}_{xy}^2 g(x_2, y^*(x_2))\Gamma_{y^*(x_1)}^{y^*(x_2)}\|_{x_2}$$
$$\leq \|\mathcal{G}_{xy}^2 g(x_1, y^*(x_1)) - \mathcal{G}_{xy}^2 g(x_1, y^*(x_2))\Gamma_{y^*(x_1)}^{y^*(x_2)}\|_{x_1} + \|\Gamma_{x_1}^{x_2}\mathcal{G}_{xy}^2 g(x_1, y^*(x_2)) - \mathcal{G}_{xy}^2 g(x_2, y^*(x_2))\|_{x_2}$$
$$\leq \left(\frac{\rho L}{\mu} + \rho\right)d(x_1, x_2)$$

and

$$\|\Gamma_{y^*(x_1)}^{y^*(x_2)}\mathcal{H}_y^{-1} g(x_1, y^*(x_1)) - \mathcal{H}_y^{-1} g(x_2, y^*(x_2))\Gamma_{y^*(x_1)}^{y^*(x_2)}\|_{y^*(x_2)}$$
$$\leq \|\mathcal{H}_y^{-1} g(x_1, y^*(x_1)) - \mathcal{H}_y^{-1} g(x_2, y^*(x_1))\|_{y^*(x_1)} + \|\Gamma_{y^*(x_1)}^{y^*(x_2)}\mathcal{H}_y^{-1} g(x_2, y^*(x_1))\Gamma_{y^*(x_2)}^{y^*(x_1)} - \mathcal{H}_y^{-1} g(x_2, y^*(x_2))\|_{y^*(x_2)}$$
$$\leq \left(\frac{\rho}{\mu^2} + \frac{\rho L}{\mu^3}\right)d(x_1, x_2).$$

Combining all the results together, we can show

$$\|\Gamma_{x_1}^{x_2}\mathcal{G}F(x_1) - \mathcal{G}F(x_2)\|_{x_2} \leq \left(\frac{L^2}{\mu} + L + \left(\frac{\rho L}{\mu} + \rho\right)\frac{M}{\mu} + LM\left(\frac{\rho}{\mu^2} + \frac{\rho L}{\mu^3}\right) + \frac{L}{\mu}\left(\frac{L^2}{\mu} + L\right)\right)d(x_1, x_2)$$
$$= \left(\frac{L}{\mu} + 1\right)\left(L + \frac{\rho M}{\mu} + \frac{\rho LM}{\mu^2} + \frac{L^2}{\mu}\right)d(x_1, x_2),$$

which completes the proof. □

## C.4 On strong convexity of the lower-level problem

**Lemma 5** (Convergence under strong convexity). *Under Assumptions 1, 2, 3, suppose $\eta_y < \frac{\mu}{L^2\zeta}$, where $\zeta \geq 1$ is a curvature constant defined in Lemma 3, then we have $d^2(y_k^{s+1}, y^*(x_k)) \leq (1 + \eta_y^2\zeta L^2 - \eta_y\mu)d^2(y_k^s, y^*(x_k))$.*

*Proof of Lemma 5.* We apply the trigonometric distance bound from Lemma 3 to obtain

$$d^2(y_k^{s+1}, y^*(x_k)) \leq d^2(y_k^s, y^*(x_k)) + \eta_y^2\zeta\|\mathcal{G}_y g(x_k, y_k^s)\|_{y_k^s}^2 + 2\eta_y\langle\mathcal{G}_y g(x_k, y_k^s), \text{Exp}_{y_k^s}^{-1} y^*(x_k)\rangle_{y_k^s}$$
$$\leq d^2(y_k^s, y^*(x_k)) + \eta_y^2\zeta\|\mathcal{G}_y g(x_k, y_k^s)\|_{y_k^s}^2$$
$$+ 2\eta_y\left(g(x_k, y^*(x_k)) - g(x_k, y_k^s) - \frac{\mu}{2}d^2(y_k^s, y^*(x_k))\right)$$
$$\leq (1 + \eta_y^2\zeta L^2 - \eta_y\mu)d^2(y_k^s, y^*(x_k)),$$

where the second inequality is due to geodesic strong convexity and the third inequality is due to $\|\mathcal{G}_y g(x_k, y_k^s)\|_{y_k^s}^2 = \|\mathcal{G}_y g(x_k, y_k^s) - \Gamma_{y^*(x_k)}^{y_k^s}\mathcal{G}_y g(x_k, y^*(x_k))\|_{y_k^s}^2 \leq L^2 d^2(y_k^s, y^*(x_k))$ and the fact that $y^*(x_k)$ is optimal. Here, we require $\eta_y < \frac{\mu}{L^2\zeta}$ in order for $1 + \eta_y^2\zeta L^2 - \eta_y\mu < 1$. □

## C.5 Proof of Lemma 1

*Proof of Lemma 1.* *Hessian inverse*: for the Hessian inverse approximation, we let

$$\overline{\mathcal{G}}f(x,y) = \mathcal{G}_x f(x,y) - \mathcal{G}_{xy}^2 g(x,y)\big[\mathcal{H}_y^{-1} g(x,y)[\mathcal{G}_y f(x,y)]\big].$$

It can be seen that $\mathcal{G}F(x_k) = \overline{\mathcal{G}}f(x_k, y^*(x_k))$ and $\widehat{\mathcal{G}}_{\text{hinv}}F(x_k) = \overline{\mathcal{G}}f(x_k, y_{k+1})$. Then for any $x \in \mathcal{U}_x, y_1, y_2 \in \mathcal{U}_y$, we have

$$\|\overline{\mathcal{G}}f(x,y_1) - \overline{\mathcal{G}}f(x,y_2)\|_x$$
$$\leq \|\mathcal{G}_x f(x,y_1) - \mathcal{G}_x f(x,y_2)\|_x + \|\mathcal{G}_{xy}^2 g(x,y_1) - \mathcal{G}_{xy}^2 g(x,y_2)\Gamma_{y_1}^{y_2}\|_x \|\mathcal{H}_y^{-1} g(x,y_1)\|_{y_1} \|\mathcal{G}_y f(x,y_1)\|_y$$
$$\quad + \|\mathcal{G}_{xy}^2 g(x,y_2)\|_x \|\Gamma_{y_1}^{y_2} \mathcal{H}_y^{-1} g(x,y_1)[\mathcal{G}_y f(x,y_1)] - \mathcal{H}^{-1} g(x,y_2)[\mathcal{G}_y f(x,y_2)]\|_{y_2}$$
$$\leq \big(L + \frac{\rho M}{\mu}\big) d(y_1, y_2) + L\|\Gamma_{y_1}^{y_2} \mathcal{H}_y^{-1} g(x,y_1) - \mathcal{H}_y^{-1} g(x,y_2)\Gamma_{y_1}^{y_2}\|_{y_2} \|\mathcal{G}_y f(x,y_1)\|_y$$
$$\quad + L\|\mathcal{H}_y^{-1} g(x,y_2)\|_y \|\Gamma_{y_1}^{y_2} \mathcal{G}_y f(x,y_1) - \mathcal{G}_y f(x,y_2)\|_{y_2}$$
$$\leq \big(L + \frac{\rho M + L^2}{\mu} + \frac{LM\rho}{\mu^2}\big) d(y_1, y_2),$$

where we use Assumption 2, 3 and Lemma 4.

*Conjugate gradient*: we let $v_k^* = \mathcal{H}_y^{-1} g(x_k, y^*(x_k))[\mathcal{G}_y f(x_k, y^*(x_k))] \in T_{y^*(x_k)}\mathcal{M}_y$ and let $\hat{v}_k^* = \mathcal{H}_y^{-1} g(x_k, y_{k+1})[\mathcal{G}_y f(x_k, y_{k+1})] \in T_{y_{k+1}}\mathcal{M}_y$. We first bound

$$\|\widehat{\mathcal{G}}_{\text{cg}}F(x_k) - \mathcal{G}F(x_k)\|_{x_k}$$
$$\leq \|\mathcal{G}_x f(x_k, y_{k+1}) - \mathcal{G}_x f(x_k, y^*(x_k))\|_{x_k} + \|\mathcal{G}_{xy}^2 g(x_k, y_{k+1})\|_{x_k} \|\hat{v}_k^T - \Gamma_{y^*(x_k)}^{y_{k+1}} v_k^*\|_{y_{k+1}}$$
$$\quad + \|\mathcal{G}_{xy}^2 g(x_k, y^*(x_k)) - \mathcal{G}_{xy}^2 g(x_k, y_{k+1})\Gamma_{y^*(x_k)}^{y_{k+1}}\|_{x_k} \|v_k^*\|_{y^*(x_k)}$$
$$\leq Ld(y^*(x_k), y_{k+1}) + L\|\hat{v}_k^T - \Gamma_{y^*(x_k)}^{y_{k+1}} v_k^*\|_{y_{k+1}} + \rho\, d(y^*(x_k), y_{k+1})\|v_k^*\|_{y^*(x_k)}$$
$$\leq \big(L + \kappa_\rho M\big) d(y^*(x_k), y_{k+1}) + L\|\hat{v}_k^T - \Gamma_{y^*(x_k)}^{y_{k+1}} v_k^*\|_{y_{k+1}},$$

where $\|v_k^*\|_{y^*(x_k)} \leq M/\mu$. From standard convergence result eq. 6.19 in [8], we have

$$\|\hat{v}_k^T - \hat{v}_k^*\|_{y_{k+1}}^2 \leq 4\kappa_l \Big(\frac{\sqrt{\kappa_l} - 1}{\sqrt{\kappa_l} + 1}\Big)^{2T} \|\hat{v}_k^0 - \hat{v}_k^*\|_{y_{k+1}}^2.$$

This leads to

$$\|\hat{v}_k^T - \Gamma_{y^*(x_k)}^{y_{k+1}} v_k^*\|_{y_{k+1}} \tag{4}$$
$$\leq \|\hat{v}_k^T - \hat{v}_k^*\|_{y_{k+1}} + \|\Gamma_{y^*(x_k)}^{y_{k+1}} v_k^* - \hat{v}_k^*\|_{y_{k+1}}$$
$$\leq 2\sqrt{\kappa_l}\Big(\frac{\sqrt{\kappa_l} - 1}{\sqrt{\kappa_l} + 1}\Big)^T \|\hat{v}_k^0 - \hat{v}_k^*\|_{y_{k+1}} + \|\Gamma_{y^*(x_k)}^{y_{k+1}} v_k^* - \hat{v}_k^*\|_{y_{k+1}}$$
$$\leq 2\sqrt{\kappa_l}\Big(\frac{\sqrt{\kappa_l} - 1}{\sqrt{\kappa_l} + 1}\Big)^T \|\hat{v}_k^0 - \Gamma_{y^*(x_k)}^{y_{k+1}} v_k^*\|_{y_{k+1}} + \Big(1 + 2\sqrt{\kappa_l}\Big(\frac{\sqrt{\kappa_l} - 1}{\sqrt{\kappa_l} + 1}\Big)^T\Big)\|\Gamma_{y^*(x_k)}^{y_{k+1}} v_k^* - \hat{v}_k^*\|_{y_{k+1}}$$
$$\leq 2\sqrt{\kappa_l}\Big(\frac{\sqrt{\kappa_l} - 1}{\sqrt{\kappa_l} + 1}\Big)^T \|\hat{v}_k^0 - \Gamma_{y^*(x_k)}^{y_{k+1}} v_k^*\|_{y_{k+1}} + \Big(1 + 2\sqrt{\kappa_l}\Big)\Big(\kappa_l + \frac{M\kappa_\rho}{\mu}\Big) d\big(y^*(x_k), y_{k+1}\big), \tag{5}$$

where in the last inequality, we use the definition of $v_k^*$ and $\hat{v}_k^*$ and the Lipschitzness assumptions. Combining the results yield the desired result.

*Neumann series*: let $\widehat{\mathcal{H}}_k(y) := \gamma \sum_{i=0}^{T-1} (\text{id} - \gamma \mathcal{H}_y g(x_k, y))^i$. Then we can bound

$$\|\widehat{\mathcal{G}}_{\text{ns}}F(x_k) - \mathcal{G}F(x_k)\|_{x_k}$$
$$\leq L\, d(y^*(x_k), y_{k+1})$$
$$+ \|\mathcal{G}_{xy}^2 g(x_k, y_{k+1})\|_{x_k} \|\widehat{\mathcal{H}}_k(y_{k+1}) - \Gamma_{y^*(x_k)}^{y_{k+1}} \mathcal{H}_y^{-1} g(x_k, y^*(x_k))\Gamma_{y_{k+1}}^{y^*(x_k)}\|_{y_{k+1}} \|\mathcal{G}_y f(x_k, y_{k+1})\|_{y_{k+1}}$$
$$+ \|\mathcal{G}_{xy}^2 g(x_k, y_{k+1})\|_{x_k} \|\mathcal{H}_y^{-1} g(x_k, y^*(x_k))\|_{y^*(x_k)} \|\Gamma_{y_{k+1}}^{y^*(x_k)} \mathcal{G}_y f(x_k, y_{k+1}) - \mathcal{G}_y f(x_k, y^*(x_k))\|_{y^*(x_k)}$$

$$+ \|\mathcal{G}^2_{xy}g(x_k, y_{k+1}) - \mathcal{G}^2_{xy}g(x_k, y^*(x_k))\Gamma^{y^*(x_k)}_{y_{k+1}}\|_{x_k}\|\mathcal{H}^{-1}_y g(x_k, y^*(x_k))\|_{y^*(x_k)}\|\mathcal{G}_y f(x_k, y^*(x_k))\|_{y^*(x_k)}$$

$$\leq (L + \kappa_l L + \kappa_\rho M)d(y^*(x_k), y_{k+1}) + LM\|\widehat{\mathcal{H}}_k(y_{k+1}) - \Gamma^{y_{k+1}}_{y^*(x_k)}\mathcal{H}^{-1}_y g(x_k, y^*(x_k))\Gamma^{y^*(x_k)}_{y_{k+1}}\|_{y_{k+1}}.$$

We now bound

$$\|\widehat{\mathcal{H}}_k(y_{k+1}) - \Gamma^{y_{k+1}}_{y^*(x_k)}\mathcal{H}^{-1}_y g(x_k, y^*(x_k))\Gamma^{y^*(x_k)}_{y_{k+1}}\|_{y_{k+1}}$$

$$\leq \|\widehat{\mathcal{H}}_k(y_{k+1}) - \mathcal{H}^{-1}_y g(x_k, y_{k+1})\|_{y_{k+1}} + \|\mathcal{H}^{-1}_y g(x_k, y_{k+1}) - \Gamma^{y_{k+1}}_{y^*(x_k)}\mathcal{H}^{-1}_y g(x_k, y^*(x_k))\Gamma^{y^*(x_k)}_{y_{k+1}}\|_{y_{k+1}}$$

$$\leq \|\gamma \sum_{i=T}^{\infty}(\mathrm{id} - \gamma\mathcal{H}_y g(x_k, y_{k+1}))^i\|_{y_{k+1}} + \frac{\kappa_\rho}{\mu}d(y^*(x_k), y_{k+1})$$

$$\leq \frac{(1 - \gamma\mu)^T}{\mu} + \frac{\kappa_\rho}{\mu}d(y^*(x_k), y_{k+1}),$$

where we use the lower bound on $\mathcal{H}_y g(x_k, y_{k+1})$. Substituting the results back the bound yields the desired result.

*Automatic differentiation*: Given $y^{s+1}_k = \mathrm{Exp}_{y^s_k}(-\eta_y \mathcal{G}_y g(x_k, y^s_k))$, we can show its differential is

$$\mathrm{D}_{x_k}y^{s+1}_k = \mathcal{P}_{y^{s+1}_k}\left(\mathrm{D}_{x_k}y^s_k - \eta_y\mathcal{G}^2_{yx}g(x_k, y^s_k) - \eta_y\mathcal{H}_y g(x_k, y^s_k)\mathrm{D}_{x_k}y^s_k\right) + \mathcal{E}^s_k$$

$$= \mathcal{P}_{y^{s+1}_k}\left((\mathrm{id} - \eta_y\mathcal{H}_y g(x_k, y^s_k))\mathrm{D}_{x_k}y^s_k - \eta_y\mathcal{G}^2_{yx}g(x_k, y^s_k)\right) + \mathcal{E}^s_k$$

where

$$\|\mathcal{E}^s_k\|_{y^{s+1}_k} \leq C_3\|(\mathrm{id} - \eta_y\mathcal{H}_y g(x_k, y^s_k))\mathrm{D}_{x_k}y^s_k - \eta_y\mathcal{G}^2_{yx}g(x_k, y^s_k)\|_{y^s_k}\|\mathcal{G}_y f(x_k, y^s_k)\|_{y^s_k}$$

$$\leq \eta^2_y C_3\left((1 - \eta_y\mu)C_1 + \eta_y L\right)\|\mathcal{G}_y f(x_k, y^s_k)\|_{y^s_k}.$$

In addition, we notice $\widehat{\mathcal{G}}_{\mathrm{ad}}F(x_k) = \mathcal{G}_x f(x_k, y^S_k) + (\mathrm{D}_{x_k}y^S_k)^\dagger[\mathcal{G}_y f(x_k, y^S_k)]$ and we can bound

$$\|\widehat{\mathcal{G}}_{\mathrm{ad}}F(x_k) - \mathcal{G}F(x_k)\|_{x_k}$$

$$\leq \|\mathcal{G}_x f(x_k, y^S_k) - \mathcal{G}_x f(x_k, y^*(x_k))\|_{x_k} + \|(\mathrm{D}_{x_k}y^S_k)^\dagger - (\mathrm{D}_{x_k}y^*(x_k))^\dagger\Gamma^{y^*(x_k)}_{y^S_k}\|_{x_k}\|\mathcal{G}_y f(x_k, y^S_k)\|_{y^S_k}$$

$$+ \|\mathrm{D}_{x_k}y^*(x_k)\|_{y^*(x_k)}\|\Gamma^{y^*(x_k)}_{y^S_k}\mathcal{G}_y f(x_k, y^S_k) - \mathcal{G}_y f(x_k, y^*(x_k))\|_{y^*(x_k)}$$

$$\leq (L + L\kappa_l)(1 + \eta^2_y\zeta L^2 - \eta_y\mu)^{\frac{S}{2}}d(y^*(x_k), y^0_k) + M\|\Gamma^{y^S_k}_{y^*(x_k)}\mathrm{D}_{x_k}y^*(x_k) - \mathrm{D}_{x_k}y^S_k\|_{y^S_k}, \tag{6}$$

where the second inequality uses Lemma 4.2. Then we bound

$$\|\Gamma^{y^{s+1}_k}_{y^*(x_k)}\mathrm{D}_{x_k}y^*(x_k) - \mathrm{D}_{x_k}y^{s+1}_k\|_{y^{s+1}_k}$$

$$\leq \|\Gamma^{y^{s+1}_k}_{y^*(x_k)}\mathrm{D}_{x_k}y^*(x_k) - \mathcal{P}_{y^{s+1}_k}\left((\mathrm{id} - \eta_y\mathcal{H}_y g(x_k, y^s_k))[\mathrm{D}_{x_k}y^s_k] - \eta_y\mathcal{G}^2_{yx}g(x_k, y^s_k)\right)\|_{y^{s+1}_k}$$

$$+ \eta^2_y C_3\left((1 - \eta_y\mu)C_1 + \eta_y L\right)\|\mathcal{G}_y f(x_k, y^s_k)\|_{y^s_k}$$

$$= \|\Gamma^{y^{s+1}_k}_{y^*(x_k)}\left(\mathrm{D}_{x_k}y^*(x_k) + \eta_y\mathcal{H}_y g(x_k, y^*(x_k))\mathrm{D}_{x_k}y^*(x_k) + \eta_y\mathcal{G}^2_{yx}g(x_k, y^*(x_k))\right)$$

$$- \mathcal{P}_{y^{s+1}_k}\left((\mathrm{id} - \eta_y\mathcal{H}_y g(x_k, y^s_k))[\mathrm{D}_{x_k}y^s_k] - \eta_y\mathcal{G}^2_{yx}g(x_k, y^s_k)\right)\|_{y^{s+1}_k}$$

$$+ \eta^2_y C_3\left((1 - \eta_y\mu)C_1 + \eta_y L\right)\|\mathcal{G}_y f(x_k, y^s_k)\|_{y^s_k}$$

$$= \|\Gamma^{y^s_k}_{y^{s+1}_k}\Gamma^{y^{s+1}_k}_{y^*(x_k)}\left(\mathrm{D}_{x_k}y^*(x_k) - \eta_y\mathcal{H}_y g(x_k, y^*(x_k))\mathrm{D}_{x_k}y^*(x_k) - \eta_y\mathcal{G}^2_{yx}g(x_k, y^*(x_k))\right)$$

$$- \Gamma^{y^s_k}_{y^{s+1}_k}\mathcal{P}_{y^{s+1}_k}\left((\mathrm{id} - \eta_y\mathcal{H}_y g(x_k, y^s_k))[\mathrm{D}_{x_k}y^s_k] - \eta_y\mathcal{G}^2_{yx}g(x_k, y^s_k)\right)\|_{y^s_k}$$

$$+ \eta^2_y C_3\left((1 - \eta_y\mu)C_1 + \eta_y L\right)\|\mathcal{G}_y f(x_k, y^s_k)\|_{y^s_k}$$

$$\leq \|\Gamma^{y^s_k}_{y^*(x_k)}(\mathrm{id} - \eta_y\mathcal{H}_y g(x_k, y^*(x_k)))[\mathrm{D}_{x_k}y^*(x_k)] - \eta_y\Gamma^{y^s_k}_{y^*(x_k)}\mathcal{G}^2_{yx}g(x_k, y^*(x_k))$$

$$- ((\mathrm{id} - \eta_y\mathcal{H}_y g(x_k, y^s_k))[\mathrm{D}_{x_k}y^s_k] - \eta_y\mathcal{G}^2_{yx}g(x_k, y^s_k))\|_{y^s_k} + (\eta_y C_2 + \eta^2_y C_3)\left((1 - \eta_y\mu)C_1 + \eta_y L\right)\|\mathcal{G}_y f(x_k, y^s_k)\|_{y^s_k}$$

$$= \left\|\Gamma^{y^s_k}_{y^*(x_k)}(\mathrm{id} - \eta_y\mathcal{H}_y g(x_k, y^*(x_k)))[\mathrm{D}_{x_k}y^*(x_k) - \Gamma^{y^*(x_k)}_{y^s_k}\mathrm{D}_{x_k}y^s_k]\right.$$

$$+ \eta_y\big(\mathcal{H}_y g(x_k, y_k^s) - \Gamma_{y^*(x_k)}^{y_k^s}\mathcal{H}_y g(x_k, y^*(x_k))\big)\Gamma_{y_k^s}^{y^*(x_k)}\big)[D_{x_k}y_k^s] + \eta_y\mathcal{G}_{yx}^2 g(x_k, y_k^s) - \Gamma_{y^*(x_k)}^{y_k^s}\mathcal{G}_{yx}^2 g(x_k, y^*(x_k))\Big\|_{y_k^s}$$

$$+ (\eta_y C_2 + \eta_y^2 C_3)\big((1-\eta_y\mu)C_1 + \eta_y L\big)\|\mathcal{G}_y f(x_k, y_k^s)\|_{y_k^s}$$

$$\leq (1-\eta_y\mu)\|\Gamma_{y^*(x_k)}^{y_k^s}D_{x_k}y^*(x_k) - D_{x_k}y_k^s\|_{y_k^s} + \eta_y(\kappa_l + 1)\rho d(y_k^s, y^*(x_k))$$

$$+ (\eta_y C_2 + \eta_y^2 C_3)\big((1-\eta_y\mu)C_1 + \eta_y L\big)\|\mathcal{G}_y f(x_k, y_k^s)\|_{y_k^s}$$

$$\leq (1-\eta_y\mu)\|\Gamma_{y^*(x_k)}^{y_k^s}D_{x_k}y^*(x_k) - D_{x_k}y_k^s\|_{y_k^s} + \eta_y(\kappa_l + 1)\rho(1 + \eta_y^2\zeta L^2 - \eta_y\mu)^{\frac{s}{2}}d(y_k^0, y^*(x_k))$$

$$+ \eta_y(C_2 + \eta_y C_3)\big((1-\eta_y\mu)C_1 + \eta_y L\big)\|\mathcal{G}_y f(x_k, y_k^s)\|_{y_k^s}$$

$$\leq (1-\eta_y\mu)\|\Gamma_{y^*(x_k)}^{y_k^s}D_{x_k}y^*(x_k) - D_{x_k}y_k^s\|_{y_k^s}$$

$$+ \eta_y\Big((\kappa_l + 1)\rho + (C_2 + \eta_y C_3)L\big((1-\eta_y\mu)C_1 + \eta_y L\big)\Big)(1 + \eta_y^2\zeta L^2 - \eta_y\mu)^{\frac{s}{2}}d(y_k^0, y^*(x_k)),$$

where the first equality uses the expression of $D_{x_k}y^*(x_k)$ (Proposition 1). The second last inequality follows from Lemma 5 and the last inequality is due to the smoothness of Riemannian gradient and Lemma 5, i.e., $\|\mathcal{G}_y f(x_k, y_k^s)\|_{y_k^s} = \|\Gamma_{y^*(x_k)}^{y_k^s}\mathcal{G}_y f(x_k, y^*(x_k)) - \mathcal{G}_y f(x_k, y_k^s)\|_{y_k^s} \leq Ld(y_k^s, y^*(x_k)) \leq L(1 + \eta_y^2\zeta L^2 - \eta_y\mu)^{\frac{s}{2}}d(y_k^0, y^*(x_k))$.

Finally, applying the bound recursively, we obtain

$$\|\Gamma_{y^*(x_k)}^{y_k^S}D_{x_k}y^*(x_k) - D_{x_k}y_k^S\|_{y_k^S} \leq (1-\eta_y\mu)^S\|\Gamma_{y^*(x_k)}^{y_k^0}D_{x_k}y^*(x_k) - D_{x_k}y_k^0\|_{y_k^0}$$

$$+ \eta_y\widetilde{C}\sum_{s=0}^{S-1}(1-\eta_y\mu)^{S-1-s}(1 + \eta_y^2\zeta L^2 - \eta_y\mu)^{\frac{s}{2}}d(y_k^0, y^*(x_k))$$

$$\leq \kappa_l(1-\eta_y\mu)^S + \eta_y\widetilde{C}\sum_{s=0}^{S-1}(1 + \eta_y^2\zeta L^2 - \eta_y\mu)^{S-1-\frac{s}{2}}d(y_k^0, y^*(x_k))$$

$$\leq \kappa_l(1-\eta_y\mu)^S + \eta_y\widetilde{C}\frac{(1 + \eta_y^2\zeta L^2 - \eta_y\mu)^{\frac{S-1}{2}}}{1 - (1 + \eta_y^2\zeta L^2 - \eta_y\mu)^{\frac{1}{2}}}d(y_k^0, y^*(x_k))$$

$$\leq \kappa_l(1-\eta_y\mu)^S + \frac{2\widetilde{C}}{\mu - \eta_y\zeta L^2}(1 + \eta_y^2\zeta L^2 - \eta_y\mu)^{\frac{S-1}{2}}d(y_k^0, y^*(x_k)),$$

where we let $\widetilde{C} := (\kappa_l + 1)\rho + (C_2 + \eta_y C_3)L\big((1-\eta_y\mu)C_1 + \eta_y L\big)$ and we note that $D_{x_k}y_k^0 = 0$.

Combining the above result with (6) gives

$$\|\widehat{\mathcal{G}}_{\text{ad}}F(x_k) - \mathcal{G}F(x_k)\|_{x_k}$$

$$\leq (L + L\kappa_l)(1 + \eta_y^2\zeta L^2 - \eta_y\mu)^{\frac{S}{2}}d(y^*(x_k), y_k^0) + M\|\Gamma_{y^*(x_k)}^{y_k^S}D_{x_k}y^*(x_k) - D_{x_k}y_k^S\|_{y_k^S}$$

$$\leq \Big(\frac{2M\widetilde{C}}{\mu - \eta_y\zeta L^2} + L(1 + \kappa_l)\Big)(1 + \eta_y^2\zeta L^2 - \eta_y\mu)^{\frac{S-1}{2}}d(y_k, y^*(x_k)) + M\kappa_l(1-\eta_y\mu)^S,$$

where we use the fact that $1 + \eta_y^2\zeta L^2 - \eta_y\mu \leq 1$. $\qquad\square$

### C.6 Proof of Theorem 1

*Proof of Theorem 1.* By smoothness of $F(x)$ (Lemma 4.5), we have

$$F(x_{k+1}) - F(x_k) \leq -\eta_x\langle\mathcal{G}F(x_k), \widehat{\mathcal{G}}F(x_k)\rangle_{x_k} + \frac{\eta_x^2 L_F}{2}\|\widehat{\mathcal{G}}F(x_k)\|_{x_k}^2$$

$$\leq -\big(\frac{\eta_x}{2} - \eta_x^2 L_F\big)\|\mathcal{G}F(x_k)\|_{x_k}^2 + \big(\frac{\eta_x}{2} + \eta_x^2 L_F\big)\|\mathcal{G}F(x_k) - \widehat{\mathcal{G}}F(x_k)\|_{x_k}^2. \quad (7)$$

Now we consider the different hypergradient estimator separately.

*1. Hessian inverse*: Let $C_{\text{hinv}} := L + \kappa_\rho M + \kappa_l L + \kappa_l\kappa_\rho M$.

$$\|\mathcal{G}F(x_k) - \widehat{\mathcal{G}}_{\text{hinv}}F(x_k)\|_{x_k}^2 \leq C_{\text{hinv}}^2 d^2(y^*(x_k), y_{k+1}) \leq C_{\text{hinv}}^2(1 + \eta_y^2\zeta L^2 - \eta_y\mu)^S d^2(y^*(x_k), y_k),$$
$$(8)$$

where we notice $y_{k+1} = y_k^S$ and apply Lemma 5. Furthermore,

$$d^2(y_k, y^*(x_k))$$
$$\leq 2d^2(y_{k-1}^S, y^*(x_{k-1})) + 2d^2(y^*(x_k), y^*(x_{k-1}))$$
$$\leq 2(1 + \eta_y^2 \zeta L^2 - \eta_y \mu)^S d^2(y^*(x_{k-1}), y_{k-1}) + 2\eta_x^2 \kappa_l^2 \|\widehat{\mathcal{G}}_{\mathrm{hinv}} F(x_{k-1})\|_{x_k}^2$$
$$\leq 2(1 + \eta_y^2 \zeta L^2 - \eta_y \mu)^S d^2(y^*(x_{k-1}), y_{k-1}) + 4\eta_x^2 \kappa_l^2 \|\widehat{\mathcal{G}}_{\mathrm{hinv}} F(x_{k-1}) - \mathcal{G}F(x_{k-1})\|_{x_{k-1}}^2$$
$$\quad + 4\eta_x^2 \kappa_l^2 \|\mathcal{G}F(x_{k-1})\|_{x_{k-1}}^2$$
$$\leq 2(1 + \eta_y^2 \zeta L^2 - \eta_y \mu)^S d^2(y^*(x_{k-1}), y_{k-1}) + 4\eta_x^2 \kappa_l^2 C_{\mathrm{hinv}}^2 (1 + \eta_y^2 \zeta L^2 - \eta_y \mu)^S d^2(y^*(x_{k-1}), y_{k-1})$$
$$\quad + 4\eta_x^2 \kappa_l^2 \|\mathcal{G}F(x_{k-1})\|_{x_{k-1}}^2$$
$$= 2(1 + 2\eta_x^2 \kappa_l^2 C_{\mathrm{hinv}}^2)(1 + \eta_y^2 \zeta L^2 - \eta_y \mu)^S d^2(y^*(x_{k-1}), y_{k-1}) + 4\eta_x^2 \kappa_l^2 \|\mathcal{G}F(x_{k-1})\|_{x_{k-1}}^2 \qquad (9)$$

where we apply Lemma 5 and 4.3 in the second inequality.

Construct a Lyapunov function $R_k := F(x_k) + d^2(y_k, y^*(x_k))$. Then,

$$R_{k+1} - R_k = F(x_{k+1}) - F(x_k) + \big(d^2(y_{k+1}, y^*(x_{k+1})) - d^2(y_k, y^*(x_k))\big)$$
$$\leq -\big(\frac{\eta_x}{2} - \eta_x^2 L_F\big)\|\mathcal{G}F(x_k)\|_{x_k}^2 + \big(\frac{\eta_x}{2} + \eta_x^2 L_F\big)\|\mathcal{G}F(x_k) - \widehat{\mathcal{G}}_{\mathrm{hinv}} F(x_k)\|_{x_k}^2$$
$$\quad + \Big(\big((2 + 4\eta_x^2 \kappa_l^2 C_{\mathrm{hinv}}^2)(1 + \eta_y^2 \zeta L^2 - \eta_y \mu)^S - 1\big)d^2(y^*(x_k), y_k) + 4\eta_x^2 \kappa_l^2 \|\mathcal{G}F(x_k)\|_{x_k}^2\Big)$$
$$\leq -\big(\frac{\eta_x}{2} - \eta_x^2 L_F - 4\eta_x^2 \kappa_l^2\big)\|\mathcal{G}F(x_k)\|_{x_k}^2$$
$$\quad + \Big(\big(2 + C_{\mathrm{hinv}}^2(\frac{\eta_x}{2} + \eta_x^2 L_F) + 4\eta_x^2 \kappa_l^2 C_{\mathrm{hinv}}^2\big)(1 + \eta_y^2 \zeta L^2 - \eta_y \mu)^S - 1\Big)d^2(y^*(x_k), y_k)$$
$$\leq -\big(\frac{\eta_x}{2} - 5\eta_x^2 L_F\big)\|\mathcal{G}F(x_k)\|_{x_k}^2$$
$$\quad + \Big(\big(2 + C_{\mathrm{hinv}}^2(\frac{\eta_x}{2} + 5\eta_x^2 L_F)\big)(1 + \eta_y^2 \zeta L^2 - \eta_y \mu)^S - 1\Big)d^2(y^*(x_k), y_k)$$

where we combine (7) and (9) in the first inequality and use $\kappa_l^2 \leq L_F$ in the third inequality. Now setting $\eta_x = \frac{1}{20L_F}$, we can simplify the inequality as

$$R_{k+1} - R_k \leq -\frac{1}{80L_F}\|\mathcal{G}F(x_k)\|_{x_k}^2 + \Big((2 + \frac{3C_{\mathrm{hinv}}^2}{80L_F})(1 + \eta_y^2 \zeta L^2 - \eta_y \mu)^S - 1\Big)d^2(y^*(x_k), y_k)$$
$$\leq -\frac{1}{80L_F}\|\mathcal{G}F(x_k)\|_{x_k}^2$$

where we choose $S \geq \log(\frac{80L_F}{160L_F + 3C_{\mathrm{hinv}}^2})/\log(1 + \eta_y^2 \zeta L^2 - \eta_y \mu) = \widetilde{\Theta}(\kappa_l^2 \zeta)$ for the last inequality.

Summing over $k = 0, ...K - 1$ yields

$$\frac{1}{K}\sum_{k=0}^{K-1}\|\mathcal{G}F(x_k)\|_{x_k}^2 \leq \frac{80L_F(R_0 - R_K)}{K} \leq \frac{80L_F \Delta_0}{K},$$

which suggests $\min_{k=0,...,K-1}\|\mathcal{G}F(x_k)\|_{x_k}^2 \leq \frac{80L_F \Delta_0}{K}$.

2. *Conjugate gradient*: Let $C_{\mathrm{cg}} := L + \kappa_\rho M + L\big(1 + 2\sqrt{\kappa_l}\big)\big(\kappa_l + \frac{M\kappa_\rho}{\mu}\big)$. Then we can show

$$\|\mathcal{G}F(x_k) - \widehat{\mathcal{G}}_{\mathrm{cg}} F(x_k)\|_{x_k}^2$$
$$\leq 2C_{\mathrm{cg}}^2(1 + \eta_y^2 \zeta L^2 - \eta_y \mu)^S d^2(y^*(x_k), y_k) + 8L^2 \kappa_l \Big(\frac{\sqrt{\kappa_l} - 1}{\sqrt{\kappa_l} + 1}\Big)^{2T} \|\hat{v}_k^0 - \Gamma_{y^*(x_k)}^{y_{k+1}} v_k^*\|_{y_{k+1}}^2, \qquad (10)$$

where it follows from Lemma 1 and Lemma 5. Then following similar analysis as in Hessian inverse case

$$d^2(y_{k+1}, y^*(x_{k+1})) \leq 2(1 + \eta_y^2 \zeta L^2 - \eta_y \mu)^S d^2(y^*(x_k), y_k) + 4\eta_x^2 \kappa_l^2 \|\widehat{\mathcal{G}}_{\mathrm{cg}} F(x_k) - \mathcal{G}F(x_k)\|_{x_k}^2$$

$$+ 4\eta_x^2 \kappa_l^2 \|\mathcal{G}F(x_k)\|_{x_k}^2$$
$$\leq (2 + 8\eta_x^2 \kappa_l^2 C_{\mathrm{cg}}^2)(1 + \eta_y^2 \zeta L^2 - \eta_y \mu)^S d^2(y^*(x_k), y_k)$$
$$+ 32\eta_x^2 \kappa_l^3 L^2 \Big(\frac{\sqrt{\kappa_l}-1}{\sqrt{\kappa_l}+1}\Big)^{2T} \|\hat{v}_k^0 - \Gamma_{y^*(x_k)}^{y_{k+1}} v_k^*\|_{y_{k+1}}^2 + 4\eta_x^2 \kappa_l^2 \|\mathcal{G}F(x_k)\|_{x_k}^2.$$

$$(11)$$

Further, noticing $\hat{v}_k^0 = \Gamma_{y_k}^{y_{k+1}} \hat{v}_{k-1}^T$, we bound $\|\hat{v}_k^0 - \Gamma_{y^*(x_k)}^{y_{k+1}} v_k^*\|_{y_{k+1}} = \|\hat{v}_{k-1}^T - \Gamma_{y_{k+1}}^{y_k} \Gamma_{y^*(x_k)}^{y_{k+1}} v_k^*\|_{y_k}$ as

$$\|\hat{v}_k^0 - \Gamma_{y^*(x_k)}^{y_{k+1}} v_k^*\|_{y_{k+1}}$$

$$\leq \|\hat{v}_{k-1}^T - \Gamma_{y^*(x_k)}^{y_k} v_k^*\|_{y_k} + \frac{MC_0\bar{D}}{\mu} d(y_{k+1}, y^*(x_k))$$

$$\leq \|\hat{v}_{k-1}^T - \Gamma_{y^*(x_{k-1})}^{y_k} v_{k-1}^*\|_{y_k} + \big\|v_k^* - \Gamma_{y_k}^{y^*(x_k)} \Gamma_{y^*(x_{k-1})}^{y_k} v_{k-1}^*\big\|_{y_k} + \frac{MC_0\bar{D}}{\mu} d(y_{k+1}, y^*(x_k))$$

$$\leq \|\hat{v}_{k-1}^T - \Gamma_{y^*(x_{k-1})}^{y_k} v_{k-1}^*\|_{y_k} + \big\|v_k^* - \Gamma_{y^*(x_{k-1})}^{y^*(x_k)} v_{k-1}^*\big\|_{y_k} + \frac{MC_0\bar{D}}{\mu} \big(d(y_k, y^*(x_k)) + d(y_{k+1}, y^*(x_k))\big)$$

$$\leq 2\sqrt{\kappa_l}\Big(\frac{\sqrt{\kappa_l}-1}{\sqrt{\kappa_l}+1}\Big)^T \|\hat{v}_{k-1}^0 - \Gamma_{y^*(x_{k-1})}^{y_k} v_{k-1}^*\|_{y_k} + (1 + \sqrt{\kappa_l})(\kappa_l + \frac{M\kappa_\rho}{\mu}) d(y^*(x_{k-1}), y_k)$$

$$+ \big\|v_k^* - \Gamma_{y^*(x_{k-1})}^{y^*(x_k)} v_{k-1}^*\big\|_{y_k} + \frac{2MC_0\bar{D}}{\mu} d(y_k, y^*(x_k))$$

$$\leq 2\sqrt{\kappa_l}\Big(\frac{\sqrt{\kappa_l}-1}{\sqrt{\kappa_l}+1}\Big)^T \|\hat{v}_{k-1}^0 - \Gamma_{y^*(x_{k-1})}^{y_k} v_{k-1}^*\|_{y_k} + 2\sqrt{\kappa_l}(\kappa_l + \frac{M\kappa_\rho}{\mu})(1 + \eta_y^2 \zeta L^2 - \eta_y \mu)^{\frac{S}{2}} d(y^*(x_{k-1}), y_{k-1})$$

$$+ \big\|v_k^* - \Gamma_{y^*(x_{k-1})}^{y^*(x_k)} v_{k-1}^*\big\|_{y_k} + \frac{2MC_0\bar{D}}{\mu} d(y_k, y^*(x_k))$$

$$(12)$$

where we use Lemma 2 in the first and third inequalities. The second last inequality follows from (5) and $d(y_{k+1}, y^*(x_k)) \leq d(y_k, y^*(x_k))$. The last inequality follows from Lemma 5 and $\kappa_l \geq 1$. Now we bound

$$\|v_k^* - \Gamma_{y^*(x_{k-1})}^{y^*(x_k)} v_{k-1}^*\|_{y^*(x_k)}$$

$$= \big\|\mathcal{H}_y^{-1} g(x_k, y^*(x_k))[\mathcal{G}_y f(x_k, y^*(x_k))] - \Gamma_{y^*(x_{k-1})}^{y^*(x_k)} \mathcal{H}_y^{-1} g(x_{k-1}, y^*(x_{k-1}))[\mathcal{G}_y f(x_{k-1}, y^*(x_{k-1}))]\big\|_{y^*(x_k)}$$

$$\leq M\big\|\mathcal{H}_y^{-1} g(x_k, y^*(x_k)) - \Gamma_{y^*(x_{k-1})}^{y^*(x_k)} \mathcal{H}_y^{-1} g(x_{k-1}, y^*(x_{k-1})) \Gamma_{y^*(x_k)}^{y^*(x_{k-1})}\big\|_{y^*(x_k)}$$

$$+ \frac{1}{\mu}\|\Gamma_{y^*(x_k)}^{y^*(x_{k-1})} \mathcal{G}_y f(x_k, y^*(x_k)) - \mathcal{G}_y f(x_{k-1}, y^*(x_{k-1}))\|_{y^*(x_{k-1})}$$

$$\leq M\|\mathcal{H}_y^{-1} g(x_k, y^*(x_k)) - \mathcal{H}_y^{-1} g(x_{k-1}, y^*(x_k))\|_{y^*(x_k)}$$

$$+ M\|\mathcal{H}_y^{-1} g(x_{k-1}, y^*(x_k)) - \Gamma_{y^*(x_{k-1})}^{y^*(x_k)} \mathcal{H}_y^{-1} g(x_{k-1}, y^*(x_{k-1})) \Gamma_{y^*(x_k)}^{y^*(x_{k-1})}\|_{y^*(x_k)}$$

$$+ \frac{1}{\mu}\|\Gamma_{y^*(x_k)}^{y^*(x_{k-1})} \mathcal{G}_y f(x_k, y^*(x_k)) - \mathcal{G}_y f(x_k, y^*(x_{k-1}))\|_{y^*(x_{k-1})}$$

$$+ \frac{1}{\mu}\|\mathcal{G}_y f(x_k, y^*(x_{k-1})) - \mathcal{G}_y f(x_{k-1}, y^*(x_{k-1}))\|_{y^*(x_{k-1})}$$

$$\leq \frac{M\kappa_\rho}{\mu} d(x_k, x_{k-1}) + \frac{M\kappa_\rho}{\mu} \kappa_l d(x_k, x_{k-1}) + \frac{L}{\mu}\kappa_l d(x_k, x_{k-1}) + \frac{L}{\mu} d(x_k, x_{k-1})$$

$$= \eta_x C_v \|\widehat{\mathcal{G}}_{\mathrm{cg}} F(x_{k-1}) - \mathcal{G}F(x_{k-1})\|_{x_{k-1}} + \eta_x C_v \|\mathcal{G}F(x_{k-1})\|_{x_{k-1}}$$

$$(13)$$

where we let $C_v := \frac{M\kappa_\rho}{\mu} + \frac{M\kappa_\rho \kappa_l}{\mu} + \kappa_l^2 + \kappa_l$. Combining (13) and (12), we obtain

$$\|\hat{v}_k^0 - \Gamma_{y^*(x_k)}^{y_{k+1}} v_k^*\|_{y_{k+1}}^2$$

$$\leq 20\kappa_l \Big(\frac{\sqrt{\kappa_l}-1}{\sqrt{\kappa_l}+1}\Big)^{2T} \|\hat{v}_{k-1}^0 - \Gamma_{y^*(x_{k-1})}^{y_k} v_{k-1}^*\|_{y_k}^2 + 20\kappa_l(\kappa_l + \frac{M\kappa_\rho}{\mu})^2(1 + \eta_y^2 \zeta L^2 - \eta_y \mu)^S d(y^*(x_{k-1}), y_{k-1})$$

$$+ 5\eta_x^2 C_v^2 \|\widehat{\mathcal{G}}_{\mathrm{cg}} F(x_{k-1}) - \mathcal{G}F(x_{k-1})\|_{x_{k-1}}^2 + 5\eta_x^2 C_v^2 \|\mathcal{G}F(x_{k-1})\|_{x_{k-1}}^2 + \frac{5M^2 C_0^2 \bar{D}^2}{\mu^2} d^2(y_k, y^*(x_k)).$$
$$\tag{14}$$

Now we define a Lyapunov function $R_k := F(x_k) + d^2(y_k, y^*(x_k)) + \|\hat{v}_k^0 - \Gamma_{y^*(x_k)}^{y_{k+1}} v_k^*\|_{y_{k+1}}^2$. Then

$$R_{k+1} - R_k$$
$$= (F(x_{k+1}) - F(x_k)) + \left(d^2(y_{k+1}, y^*(x_{k+1})) - d^2(y_k, y^*(x_k))\right)$$
$$\quad + \left(\|\hat{v}_{k+1}^0 - \Gamma_{y^*(x_{k+1})}^{y_{k+2}} v_{k+1}^*\|_{y_{k+2}}^2 - \|\hat{v}_k^0 - \Gamma_{y^*(x_k)}^{y_{k+1}} v_k^*\|_{y_{k+1}}^2\right)$$
$$\leq -\left(\frac{\eta_x}{2} - \eta_x^2 L_F\right)\|\mathcal{G}F(x_k)\|_{x_k}^2 + \left(\frac{\eta_x}{2} + \eta_x^2 L_F\right)\|\mathcal{G}F(x_k) - \widehat{\mathcal{G}}_{\mathrm{cg}} F(x_k)\|_{x_k}^2 + d^2(y_{k+1}, y^*(x_{k+1})) - d^2(y_k, y^*(x_k))$$
$$\quad + \left(20\kappa_l\left(\frac{\sqrt{\kappa_l} - 1}{\sqrt{\kappa_l} + 1}\right)^{2T} - 1\right)\|\hat{v}_k^0 - \Gamma_{y^*(x_k)}^{y_{k+1}} v_k^*\|_{y_{k+1}}^2 + 20\kappa_l(\kappa_l + \frac{M\kappa_\rho}{\mu})^2 (1 + \eta_y^2 \zeta L^2 - \eta_y \mu)^S d^2(y^*(x_k), y_k)$$
$$\quad + 5\eta_x^2 C_v^2 \|\widehat{\mathcal{G}}_{\mathrm{cg}} F(x_k) - \mathcal{G}F(x_k)\|_{x_k}^2 + 5\eta_x^2 C_v^2 \|\mathcal{G}F(x_k)\|_{x_k}^2 + \frac{5M^2 C_0^2 \bar{D}^2}{\mu^2} d^2(y_{k+1}, y^*(x_{k+1}))$$
$$= -\left(\frac{\eta_x}{2} - \eta_x^2 L_F - 5\eta_x^2 C_v^2\right)\|\mathcal{G}F(x_k)\|_{x_k}^2 + \left(\frac{\eta_x}{2} + \eta_x^2 L_F + 5\eta_x^2 C_v^2\right)\|\mathcal{G}F(x_k) - \widehat{\mathcal{G}}_{\mathrm{cg}} F(x_k)\|_{x_k}^2$$
$$\quad + \left(\frac{5M^2 C_0^2 \bar{D}^2}{\mu^2} + 1\right)d^2(y_{k+1}, y^*(x_{k+1})) + \left(20\kappa_l(\kappa_l + \frac{M\kappa_\rho}{\mu})^2 (1 + \eta_y^2 \zeta L^2 - \eta_y \mu)^S - 1\right)d^2(y^*(x_k), y_k)$$
$$\quad + \left(20\kappa_l\left(\frac{\sqrt{\kappa_l} - 1}{\sqrt{\kappa_l} + 1}\right)^{2T} - 1\right)\|\hat{v}_k^0 - \Gamma_{y^*(x_k)}^{y_{k+1}} v_k^*\|_{y_{k+1}}^2$$
$$\leq -\left(\frac{\eta_x}{2} - \eta_x^2 L_F - 5\eta_x^2 C_v^2 - 4\eta_x^2 \kappa_l^2\left(\frac{5M^2 C_0^2 D^2}{\mu^2} + 1\right)\right)\|\mathcal{G}F(x_k)\|_{x_k}^2$$
$$\quad + \left(\frac{\eta_x}{2} + \eta_x^2 L_F + 5\eta_x^2 C_v^2\right)\|\mathcal{G}F(x_k) - \widehat{\mathcal{G}}_{\mathrm{cg}} F(x_k)\|_{x_k}^2$$
$$\quad + \left(\left(\left(\frac{5M^2 C_0^2 D^2}{\mu^2} + 1\right)(2 + 8\eta_x^2 \kappa_l^2 C_{\mathrm{cg}}^2) + 20\kappa_l\left(\kappa_l + \frac{M\kappa_\rho}{\mu}\right)^2\right)(1 + \eta_y^2 \zeta L^2 - \eta_y \mu)^S - 1\right)d^2(y_k, y^*(x_k))$$
$$\quad + \left(\left(32\eta_x^2 \kappa_l^3 L^2\left(\frac{5M^2 C_0^2 \bar{D}^2}{\mu^2} + 1\right) + 20\kappa_l\right)\left(\frac{\sqrt{\kappa_l} - 1}{\sqrt{\kappa_l} + 1}\right)^{2T} - 1\right)\|\hat{v}_k^0 - \Gamma_{y^*(x_k)}^{y_{k+1}} v_k^*\|_{y_{k+1}}^2$$
$$\leq -\left(\frac{\eta_x}{2} - 6\eta_x^2 \Lambda\right)\|\mathcal{G}F(x_k)\|_{x_k}^2 + \left(\frac{\eta_x}{2} + 6\eta_x^2 \Lambda\right)\|\mathcal{G}F(x_k) - \widehat{\mathcal{G}}_{\mathrm{cg}} F(x_k)\|_{x_k}^2$$
$$\quad + \left(\left(\left(\frac{5M^2 C_0^2 \bar{D}^2}{\mu^2} + 1\right)(2 + 8\eta_x^2 \kappa_l^2 C_{\mathrm{cg}}^2) + 20\kappa_l\left(\kappa_l + \frac{M\kappa_\rho}{\mu}\right)^2\right)(1 + \eta_y^2 \zeta L^2 - \eta_y \mu)^S - 1\right)d^2(y_k, y^*(x_k))$$
$$\quad + \left(\left(32\eta_x^2 \kappa_l^3 L^2\left(\frac{5M^2 C_0^2 \bar{D}^2}{\mu^2} + 1\right) + 20\kappa_l\right)\left(\frac{\sqrt{\kappa_l} - 1}{\sqrt{\kappa_l} + 1}\right)^{2T} - 1\right)\|\hat{v}_k^0 - \Gamma_{y^*(x_k)}^{y_{k+1}} v_k^*\|_{y_{k+1}}^2$$
$$\leq -\frac{1}{96\Lambda}\|\mathcal{G}F(x_k)\|_{x_k}^2 + \left(\left(32\eta_x^2 \kappa_l^3 L^2\left(\frac{5M^2 C_0^2 \bar{D}^2}{\mu^2} + 1\right) + 20\kappa_l + \frac{L^2 \kappa_l}{4\Lambda}\right)\left(\frac{\sqrt{\kappa_l} - 1}{\sqrt{\kappa_l} + 1}\right)^{2T} - 1\right)\|\hat{v}_k^0 - \Gamma_{y^*(x_k)}^{y_{k+1}} v_k^*\|_{y_{k+1}}^2$$
$$\quad + \left(\left(\left(\frac{5M^2 C_0^2 \bar{D}^2}{\mu^2} + 1\right)(2 + 8\eta_x^2 \kappa_l^2 C_{\mathrm{cg}}^2) + 20\kappa_l\left(\kappa_l + \frac{M\kappa_\rho}{\mu}\right)^2 + \frac{C_{\mathrm{cg}}^2}{16\Lambda}\right)(1 + \eta_y^2 \zeta L^2 - \eta_y \mu)^S - 1\right)d^2(y_k, y^*(x_k))$$
$$\leq -\frac{1}{96\Lambda}\|\mathcal{G}F(x_k)\|_{x_k}^2$$

where we use (7), (14) in the first inequality and (11) in the second inequality. In the third inequality, we let $\Lambda := C_v^2 + \kappa_l^2\left(\frac{5M^2 C_0^2 \bar{D}^2}{\mu} + 1\right)$ and because $L_F = \Theta(\kappa_l^3)$ and $\Lambda = \Theta(\kappa_l^4)$, we can without loss of generality have $L_F \leq \Lambda$. We also choose $\eta_x = \frac{1}{24\Lambda}$ and use (10) for the fourth inequality. The last inequality follows by choosing

$$S \geq -\log\left(\left(\frac{5M^2 C_0^2 \bar{D}^2}{\mu^2} + 1\right)(2 + 8\eta_x^2 \kappa_l^2 C_{\mathrm{cg}}^2) + 20\kappa_l\left(\kappa_l + \frac{M\kappa_\rho}{\mu}\right)^2\right)/\log(1 + \eta_y^2 \zeta L^2 - \eta_y \mu) = \widetilde{\Theta}(\kappa_l^2 \zeta)$$

$$T \geq -\frac{1}{2}\log\left(32\eta_x^2 \kappa_l^3 L^2\left(\frac{5M^2 C_0^2 \bar{D}^2}{\mu^2} + 1\right) + 20\kappa_l\right)/\log\left(\frac{\sqrt{\kappa_l} - 1}{\sqrt{\kappa_l} + 1}\right) = \widetilde{\Theta}(\sqrt{\kappa_l}).$$

Finally, telescoping the inequality, we obtain

$$\frac{1}{K}\sum_{k=0}^{K-1}\|\mathcal{G}F(x_k)\|_{x_k} \leq \frac{96\Lambda R_0}{K} = \frac{96\Lambda}{K}\Big(F(x_k) + d^2(y_0, y^*(x_0)) + \|v_0^*\|_{y^*(x_0)}^2\Big),$$

where we use the fact that $\hat{v}_k^0 = 0$ and the isometry property of parallel transport.

*3. Truncated Neumann series*: Let $C_{\mathrm{ns}} := L + \kappa_l L + \kappa_\rho M + \kappa_l \kappa_\rho M$. Here we notice that $C_{\mathrm{ns}} = C_{\mathrm{hinv}}$. Then by Lemma 1, we see

$$\|\widehat{\mathcal{G}}_{\mathrm{ns}}F(x_k) - \mathcal{G}F(x_k)\|_{x_k}^2 \leq 2C_{\mathrm{ns}}^2(1 + \eta_y^2\zeta L^2 - \eta_y\mu)^S d^2(y^*(x_k), y_k) + 2\kappa_l^2 M^2(1 - \gamma\mu)^{2T}. \tag{15}$$

Similar in the previous analysis,

$$d^2(y_{k+1}, y^*(x_{k+1})) \leq 2(1 + \eta_y^2\zeta L^2 - \eta_y\mu)^S d^2(y^*(x_k), y_k) + 4\eta_x^2\kappa_l^2\|\widehat{\mathcal{G}}_{\mathrm{ns}}F(x_k) - \mathcal{G}F(x_k)\|_{x_k}^2$$
$$+ 4\eta_x^2\kappa_l^2\|\mathcal{G}F(x_k)\|_{x_k}^2 \tag{16}$$

Let the Lyapunov function be $R_k := F(x_k) + d^2(y_k, y^*(x_k))$. Then

$$R_{k+1} - R_k$$
$$\leq -\Big(\frac{\eta_x}{2} - \eta_x^2 L_F - 4\eta_x^2\kappa_l^2\Big)\|\mathcal{G}F(x_k)\|_{x_k}^2 + \Big(\frac{\eta_x}{2} + \eta_x^2 L_F + 4\eta_x^2\kappa_l^2\Big)\|\mathcal{G}F(x_k) - \widehat{\mathcal{G}}_{\mathrm{ns}}F(x_k)\|_{x_k}^2$$
$$+ \Big(2(1 + \eta_y^2\zeta L^2 - \eta_y\mu)^S - 1\Big)d^2(y_k, y^*(x_k))$$
$$\leq -\frac{1}{80L_F}\|\mathcal{G}F(x_k)\|_{x_k}^2 + \Big(\big(2 + \frac{3C_{\mathrm{ns}}^2}{40L_F}\big)(1 + \eta_y^2\zeta L^2 - \eta_y\mu)^S - 1\Big)d^2(y_k, y^*(x_k))$$
$$+ \frac{3}{40L_F}\kappa_l^2 M^2(1 - \gamma\mu)^{2T}$$

where we set $\eta_x = \frac{1}{20L_F}$ and apply (15) in the second inequality.

Now setting $S \geq \log(\frac{40L_F}{80L_F + 3C_{\mathrm{ns}}^2})/\log(1 + \eta_y^2\zeta L^2 - \eta_y\mu) = \widetilde{\Theta}(\kappa_l^2\zeta)$ and telescoping the results yields

$$\frac{1}{K}\sum_{k=0}^{K-1}\|\mathcal{G}F(x_k)\|_{x_k}^2 \leq \frac{80L_F R_0}{K} + 6\kappa_l^2 M^2(1 - \gamma\mu)^{2T} \leq \frac{80L_F R_0}{K} + \frac{\epsilon}{2}$$

where we set $T \geq -\frac{1}{2}\log(\frac{12\kappa_l^2 M^2}{\epsilon})/\log(1 - \gamma\mu) = \widetilde{\Theta}(\kappa\log(\frac{1}{\epsilon}))$.

*4. Automatic differentiation*: Let $C_{\mathrm{ad}} := \frac{2M\widetilde{C}}{\mu - \eta_y\zeta L^2} + L(1 + \kappa_l)$. Then

$$\|\mathcal{G}F(x_k) - \widehat{\mathcal{G}}_{\mathrm{ad}}F(x_k)\|_{x_k}^2 \leq 2C_{\mathrm{ad}}^2(1 + \eta_y^2\zeta L^2 - \eta_y\mu)^{S-1} d^2(y_k, y^*(x_k)) + 2M^2\kappa_l^2(1 - \eta_y\mu)^{2S}$$

and similarly

$$d^2(y_{k+1}, y^*(x_{k+1})) \leq 2(1 + \eta_y^2\zeta L^2 - \eta_y\mu)^S d^2(y^*(x_k), y_k) + 4\eta_x^2\kappa_l^2\|\widehat{\mathcal{G}}_{\mathrm{ad}}F(x_k) - \mathcal{G}F(x_k)\|_{x_k}^2$$
$$+ 4\eta_x^2\kappa_l^2\|\mathcal{G}F(x_k)\|_{x_k}^2$$

Let the Lyapunov function be $R_k := F(x_k) + d^2(y_k, y^*(x_k))$. Then

$$R_{k+1} - R_k \leq -\Big(\frac{\eta_x}{2} - \eta_x^2 L_F - 4\eta_x^2\kappa_l^2\Big)\|\mathcal{G}F(x_k)\|_{x_k}^2 + \Big(\frac{\eta_x}{2} + \eta_x^2 L_F + 4\eta_x^2\kappa_l^2\Big)\|\mathcal{G}F(x_k) - \widehat{\mathcal{G}}_{\mathrm{ad}}F(x_k)\|_{x_k}^2$$
$$+ \Big((2(1 + \eta_y^2\zeta L^2 - \eta_y\mu)^{S-1} - 1)d^2(y_k, y^*(x_k))\Big)$$
$$\leq -\frac{1}{80L_F}\|\mathcal{G}F(x_k)\|_{x_k}^2 + \Big(\big((2 + \frac{3C_{\mathrm{ad}}^2}{40L_F})(1 + \eta_y^2\zeta L^2 - \eta_y\mu)^{S-1} - 1\big)d^2(y_k, y^*(x_k))$$
$$+ \frac{3}{40L_F}M^2\kappa_l^2(1 - \eta_y\mu)^{2S}$$
$$\leq -\frac{1}{80L_F}\|\mathcal{G}F(x_k)\|_{x_k}^2 + \frac{3}{40L_F}M^2\kappa_l^2(1 - \eta_y\mu)^{2S}$$

where we set $\eta_x = \frac{1}{20L_F}$ and choose $S \geq \log \frac{40L_F}{80L_F + 3C_{ad}^2} / \log(1 + \eta_y^2 \zeta L^2 - \eta_y \mu) + 1 = \widetilde{\Theta}(\kappa_l^2 \zeta)$. Telescoping the result gives

$$\frac{1}{K} \sum_{k=0}^{K-1} \|\mathcal{G}F(x_k)\|_{x_k}^2 \leq \frac{80L_F R_0}{K} + 6M^2 \kappa_l^2 (1 - \eta_y \mu)^{2S} \leq \frac{80L_F R_0}{K} + \frac{\epsilon}{2},$$

by choosing $S \geq \frac{1}{2} \log(\frac{\epsilon}{12M^2 \kappa_l^2}) \log(1 - \eta_y \mu) = \widetilde{\Theta}(\kappa_l^2 \zeta \log(\frac{1}{\epsilon}))$. Hence we set $S \geq \widetilde{\Theta}(\kappa_l^2 \zeta \log(\frac{1}{\epsilon}))$ for both conditions to hold. $\square$

### C.7 Proof of Corollary 1

The computational cost of gradient and Hessian for each method for approximating the hypergradient are as follows.

**Corollary 1.** *The complexities of reaching an $\epsilon$-stationary solution are*

- *Hessian inverse: $G_f = O(\kappa_l^3 \epsilon^{-1})$, $G_g = \widetilde{O}(\kappa_l^5 \zeta \epsilon^{-1})$, $JV_g = O(\kappa_l^3 \epsilon^{-1})$, $HV_g = NA$.*

- *Conjugate gradient: $G_f = O(\kappa_l^4 \epsilon^{-1})$, $G_g = \widetilde{O}(\kappa_l^6 \zeta \epsilon^{-1})$, $JV_g = O(\kappa_l^4 \epsilon^{-1})$, $HV_g = \widetilde{O}(\kappa_l^{4.5} \epsilon^{-1})$.*

- *Truncated Neumann series: $G_f = O(\kappa_l^3 \epsilon^{-1})$, $G_g = \widetilde{O}(\kappa_l^5 \zeta \epsilon^{-1})$, $JV_g = O(\kappa_l^3 \epsilon^{-1})$, $HV_g = \widetilde{O}(\kappa_l^4 \epsilon^{-1} \log(\epsilon^{-1}))$.*

- *Automatic differentiation: $G_f = O(\kappa_l^3 \epsilon^{-1})$, $G_g = \widetilde{O}(\kappa_l^5 \zeta \epsilon^{-1} \log(\epsilon^{-1}))$, $JV_g = \widetilde{O}(\kappa_l^5 \zeta \epsilon^{-1} \log(\epsilon^{-1}))$, $HV_g = \widetilde{O}(\kappa_l^5 \zeta \epsilon^{-1} \log(\epsilon^{-1}))$.*

*Proof of Corollary 1.* From the convergence established in Theorem 1, we see the iterations in order to reach $\epsilon$-stationary solution are given by

- *(Hessian inverse)* $K = O(L_F \epsilon^{-1}) = O(\kappa_l^3 \epsilon^{-1})$, $S = \widetilde{O}(\kappa_l^2 \zeta)$.

- *(Conjugate gradient)* $K = O(\Lambda \epsilon^{-1}) = O(\kappa_l^4 \epsilon^{-1})$, $S = \widetilde{O}(\kappa_l^2 \zeta)$, $T = \widetilde{O}(\sqrt{\kappa_l})$.

- *(Truncated Neumann series)* $K = O(\kappa_l^3 \epsilon^{-1})$, $S = \widetilde{O}(\kappa_l^2 \zeta)$, $T = \widetilde{O}(\kappa_l \log(\epsilon^{-1}))$.

- *(Automatic differentiation)* $K = O(\kappa_l^3 \epsilon^{-1})$, $S = \widetilde{O}(\kappa_l^2 \zeta \log(\epsilon^{-1}))$.

Then based on Algorithm 1, the gradient complexities are $G_f = 2K$ and $G_g = KS$ and cross-derivative and Hessian product complexities are $JV_g = K$, $HV_g = KT$ for CG and NS and $JV_g = KS$, $HV_g = KS$ for AD (which we approximate based on the analysis in Lemma 1). We notice here for the Hessian inverse, because we do not compute Hessian vector product, we write NA for Hessian vector product based on the Neumann series. This completes the proof. $\square$

## D Proofs for Section 3.3

We first show Lemma 4 holds for each $f_i(x, y), g_i(x, y)$. Further, the variance of the estimate can be bounded as follows. We here use $[\cdot]$ to denote all possible derivatives, including $x, y, xy, yx$.

**Lemma 6.** *Under Assumption 4, we have for any $x, y \in \mathcal{U}$, (1) $\mathbb{E}\|\mathcal{G}_{[\cdot]} f_i(x, y) - \mathcal{G}_{[\cdot]} f(x, y)\|_{[\cdot]}^2 \leq M^2$. (2) $\mathbb{E}\|\mathcal{G}_{[\cdot]}^2 g_i(x, y) - \mathcal{G}_{[\cdot]}^2 g(x, y)\|_{[\cdot]}^2 \leq L^2$. (3) $\mathbb{E}\|\mathcal{H}_y^{-1} g_i(x, y) - \mathcal{H}_y^{-1} g(x, y)\|_y^2 \leq \mu^{-2}$.*

For notation, denote the filtration $\mathcal{F}_k := \{y_0, x_0, y_1, x_1, ..., x_k, y_{k+1}\}$ and here we let $\mathbb{E}_k := \mathbb{E}[\cdot|\mathcal{F}_k]$. With a slight abuse of notation, we further consider $\mathcal{F}_k^s := \{y_0, x_0, y_1, x_1, ..., y_k, y_k^1, ..., y_k^s\}$ and correspondingly let $\mathbb{E}_k^s := \mathbb{E}[\cdot|\mathcal{F}_k^s]$.

**Lemma 7** (Convergence under strong convexity and stochastic setting). *Under stochastic setting and under the Assumption that $g$ is geodesic strongly convex, we can show* $\mathbb{E}_k^s d^2(y_k^{s+1}, y^*(x_k)) \leq (1 + \eta_y^2\zeta L^2 - \eta_y\mu)d^2(y_k^s, y^*(x_k)) + \frac{\eta_y^2\zeta M^2}{|\mathcal{B}_1|}$ *and* $\mathbb{E}_{k-1}d^2(y_{k+1}, y^*(x_k)) \leq (1 + \eta_y^2\zeta L^2 - \eta_y\mu)^S d^2(y_k, y^*(x_k)) + \frac{\eta_y\zeta M^2}{\mu - \eta_y\zeta L^2}\frac{1}{|\mathcal{B}_1|}$.

**Lemma 8.** *Under Assumption 4, we can bound* $\mathbb{E}_k\|\widehat{\mathcal{G}}F(x_k) - \mathcal{G}F(x_k)\|_{x_k}^2 \leq \frac{4M^2 + 16M^2\kappa_l^2}{|\mathcal{B}_2|} + \frac{8M^2\kappa_l^2}{|\mathcal{B}_3|} + \frac{16M^2\kappa_l^2}{|\mathcal{B}_4|} + 2C_{\text{hinv}}^2 d^2(y_{k+1}, y^*(x_k))$.

### D.1 Proofs for the lemmas

*Proof of Lemma 6.* Here we only prove one and the rest follows exactly. Due to the unbiasedness of the stochastic estimate, we have

$$\mathbb{E}\|\mathcal{G}_x f_i(x,y) - \mathcal{G}f(x,y)\|_x^2 = \mathbb{E}\|\mathcal{G}_x f_i(x,y)\|_x^2 - \|\mathcal{G}_x f(x,y)\|_x^2 \leq \mathbb{E}\|\mathcal{G}_x f_i(x,y)\|_x^2 \leq M^2$$

where we use Assumption 4. $\qquad\square$

*Proof of Lemma 7.* Similarly from the proof of Lemma 5, we take expectation over $\mathcal{F}_k^s$

$$\mathbb{E}_k^s d^2(y_k^{s+1}, y^*(x_k))$$
$$\leq \mathbb{E}_k^s\big[d^2(y_k^s, y^*(x_k)) + \eta_y^2\zeta\mathbb{E}_k^s\|\mathcal{G}_y g_{\mathcal{B}_1}(x_k, y_k^s)\|_{y_k^s}^2 + 2\eta_y\langle\mathcal{G}_y g_{\mathcal{B}_1}(x_k, y_k^s), \text{Exp}_{y_k^s}^{-1} y^*(x_k)\rangle_{y_k^s}\big]$$
$$\leq d^2(y_k^s, y^*(x_k)) + \eta_y^2\zeta\mathbb{E}_k^s\|\mathcal{G}_y g_{\mathcal{B}_1}(x_k, y_k^s) - \mathcal{G}_y g(x_k, y_k^s)\|_{y_k^s}^2 + \eta_y^2\zeta\|\mathcal{G}_y g(x_k, y_k^s)\|_{y_k^s}^2$$
$$\quad + 2\eta_y\langle\mathcal{G}_y g(x_k, y_k^s), \text{Exp}_{y_k^s}^{-1} y^*(x_k)\rangle_{y_k^s}$$
$$\leq (1 + \eta_y^2\zeta L^2 - \eta_y\mu)d^2(y_k^s, y^*(x_k)) + \eta_y^2\zeta\mathbb{E}_k^s\frac{1}{|\mathcal{B}_1|^2}\sum_{i\in\mathcal{B}_1}\mathbb{E}_k^s\|\mathcal{G}_y g_i(x_k, y_k^s) - \mathcal{G}_y g(x_k, y_k^s)\|_{y_k^s}^2$$
$$\leq (1 + \eta_y^2\zeta L^2 - \eta_y\mu)d^2(y_k^s, y^*(x_k)) + \frac{\eta_y^2\zeta M^2}{|\mathcal{B}_1|},$$

where we use the strong convexity and the fact that $\mathbb{E}\|\mathcal{G}_y g_{\mathcal{B}_1}(x,y) - \mathcal{G}_y g(x,y)\|_y^2 = \frac{1}{|\mathcal{B}_1|^2}\mathbb{E}\|\sum_{i\in\mathcal{B}_1}(\mathcal{G}_y g_i(x,y) - \mathcal{G}_y g(x,y))\|_y^2 = \frac{1}{|\mathcal{B}_1|^2}\sum_{i\in\mathcal{B}_1}\mathbb{E}\|\mathcal{G}_y g_i(x,y) - \mathcal{G}_y g(x,y)\|_y^2$ in the third inequality and Lemma 6 in the last inequality. Further, we telescope the inequality and taking the expectation $\mathbb{E}_{k-1}$ gets

$$\mathbb{E}_{k-1}d^2(y_k^S, y^*(x_k)) \leq (1 + \eta_y^2\zeta L^2 - \eta_y\mu)^S d^2(y_k, y^*(x_k)) + \frac{\eta_y^2\zeta M^2}{|\mathcal{B}_1|}\sum_{s=0}^{S-1}(1 + \eta_y^2\zeta L^2 - \eta_y\mu)^s$$
$$\leq (1 + \eta_y^2\zeta L^2 - \eta_y\mu)^S d^2(y_k, y^*(x_k)) + \frac{\eta_y\zeta M^2}{\mu - \eta_y\zeta L^2}\frac{1}{|\mathcal{B}_1|},$$

where we use the fact that $\sum_{s=0}^{S-1}\theta^s \leq \frac{1}{1-\theta}$ for $0 < \theta < 1$. $\qquad\square$

*Proof of Lemma 8.* Recall that $\mathcal{G}F(x_k) = \mathcal{G}_x f(x, y^*(x)) - \mathcal{G}_{xy}^2 g(x, y^*(x))\big[\mathcal{H}_y^{-1} g(x, y^*(x))[\mathcal{G}_y f(x, y^*(x))]\big]$. Then

$$\mathbb{E}_k\|\widehat{\mathcal{G}}F(x_k) - \mathcal{G}F(x_k)\|_{x_k}^2$$
$$\leq 2\mathbb{E}_k\|\widehat{\mathcal{G}}F(x_k) - \widehat{\mathcal{G}}_{\text{hinv}}F(x_k)\|_{x_k}^2 + 2\|\widehat{\mathcal{G}}_{\text{hinv}}F(x_k) - \mathcal{G}F(x_k)\|_{x_k}^2$$
$$\leq 2\mathbb{E}_k\|\widehat{\mathcal{G}}F(x_k) - \widehat{\mathcal{G}}_{\text{hinv}}F(x_k)\|_{x_k}^2 + 2C_{\text{hinv}}^2 d^2(y_{k+1}, y^*(x_k)) \qquad (17)$$

where the second inequality uses Lemma 1. Now we bound the first term $\mathbb{E}_k\|\widehat{\mathcal{G}}F(x_k) - \widehat{\mathcal{G}}_{\text{hinv}}F(x_k)\|_{x_k}^2$ as follows.

First we bound

$$\mathbb{E}_k\|\mathcal{H}_y^{-1} g_{\mathcal{B}_4}(x_k, y_{k+1})[\mathcal{G}_y f_{\mathcal{B}_2}(x_k, y_{k+1})] - \mathcal{H}_y^{-1} g(x_k, y_{k+1})[\mathcal{G}_y f(x_k, y_{k+1})]\|_{y_{k+1}}^2$$

$$\leq 2\mathbb{E}_k\|\mathcal{H}_y^{-1}g(x_k,y_{k+1})[\mathcal{G}_y f_{\mathcal{B}_2}(x_k,y_{k+1}) - \mathcal{G}_y f(x_k,y_{k+1})]\|_{y_{k+1}}^2$$

$$+ 2\mathbb{E}_k\|(\mathcal{H}_y^{-1}g_{\mathcal{B}_4}(x_k,y_{k+1}) - \mathcal{H}_y^{-1}g(x_k,y_{k+1}))[\mathcal{G}_y f_{\mathcal{B}_2}(x_k,y_{k+1})]\|_{y_{k+1}}^2$$

$$\leq 2\|\mathcal{H}_y^{-1}g(x_k,y_{k+1})\|_{y_{k+1}}^2\mathbb{E}_k\|\mathcal{G}_y f_{\mathcal{B}_2}(x_k,y_{k+1}) - \mathcal{G}_y f(x_k,y_{k+1})\|_{y_{k+1}}^2$$

$$+ 2\mathbb{E}_k\|\mathcal{H}_y^{-1}g_{\mathcal{B}_4}(x_k,y_{k+1}) - \mathcal{H}_y^{-1}g(x_k,y_{k+1})\|_{y_{k+1}}^2\mathbb{E}_k\|\mathcal{G}_y f_{\mathcal{B}_2}(x_k,y_{k+1})\|_{y_{k+1}}^2$$

$$\leq \frac{2M^2}{\mu^2}\Big(\frac{1}{|\mathcal{B}_2|} + \frac{1}{|\mathcal{B}_4|}\Big)$$

where we notice that $\|\mathcal{G}_y f_{\mathcal{B}_2}(x_k,y_{k+1})\|_{y_{k+1}} \leq \frac{1}{|\mathcal{B}_2|}\sum_{i\in\mathcal{B}_2}\|\mathcal{G}_y f_i(x_k,y_{k+1})\|_{y_{k+1}} \leq M$.

Hence, we can bound

$$\mathbb{E}_k\|\widehat{\mathcal{G}}F(x_k) - \widehat{\mathcal{G}}_{\mathrm{hinv}}F(x_k)\|_{x_k}^2$$

$$\leq 2\mathbb{E}_k\|\mathcal{G}_x f_{\mathcal{B}_2}(x_k,y_{k+1}) - \mathcal{G}_x f(x_k,y_{k+1})\|_{x_k}^2 + \frac{8M^2}{\mu^2}\Big(\frac{1}{|\mathcal{B}_2|} + \frac{1}{|\mathcal{B}_4|}\Big)\mathbb{E}_k\|\mathcal{G}_{xy}^2 g_{\mathcal{B}_3}(x_k,y_{k+1})\|_{x_k}^2$$

$$+ 4\mathbb{E}_k\|\mathcal{G}_{xy}^2 g(x_k,y_{k+1}) - \mathcal{G}_{xy}^2 g_{\mathcal{B}_3}(x_k,y_{k+1})\|_{x_k}^2\|\mathcal{H}_y^{-1}g(x_k,y_{k+1})[\mathcal{G}_y f(x_k,y_{k+1})]\|_{y_{k+1}}^2$$

$$\leq \frac{2M^2}{|\mathcal{B}_2|} + 8M^2\kappa_l^2\Big(\frac{1}{|\mathcal{B}_2|} + \frac{1}{|\mathcal{B}_4|}\Big) + \frac{4M^2\kappa_l^2}{|\mathcal{B}_3|}, \tag{18}$$

where we use Lemma 6 in the last inequality. Combining (18) with (17) yields the desired result. $\quad\square$

### D.2  Proof of Theorem 2

*Proof of Theorem 2.* From the smoothness of $F(x)$ (i.e., (7)) and taking full expectation we obtain,

$$\mathbb{E}[F(x_{k+1}) - F(x_k)] \leq -\Big(\frac{\eta_x}{2} - \eta_x^2 L_F\Big)\mathbb{E}\|\mathcal{G}F(x_k)\|_{x_k}^2 + \Big(\frac{\eta_x}{2} + \eta_x^2 L_F\Big)\mathbb{E}\|\mathcal{G}F(x_k) - \widehat{\mathcal{G}}F(x_k)\|_{x_k}^2.$$

Further, we can bound

$$\mathbb{E}d^2(y_{k+1}, y^*(x_{k+1}))$$

$$\leq 2\mathbb{E}d^2(y_{k+1}, y^*(x_k)) + 4\eta_x^2\kappa_l^2\mathbb{E}\|\widehat{\mathcal{G}}F(x_k) - \mathcal{G}F(x_k)\|_{x_k}^2 + 4\eta_x^2\kappa_l^2\mathbb{E}\|\mathcal{G}F(x_k)\|_{x_k}$$

$$\leq 2(1 + \eta_y^2\zeta L^2 - \eta_y\mu)^S\mathbb{E}d^2(y_k, y^*(x_k)) + \frac{2\eta_y\zeta M^2}{\mu - \eta_y\zeta L^2}\frac{1}{|\mathcal{B}_1|} + 4\eta_x^2\kappa_l^2\mathbb{E}\|\mathcal{G}F(x_k)\|_{x_k}$$

$$+ 4\eta_x^2\kappa_l^2\mathbb{E}\|\widehat{\mathcal{G}}F(x_k) - \mathcal{G}F(x_k)\|_{x_k}^2$$

where we use Lemma 7 and 8 in the second inequality.

Next, we construct a Lyapunov function as $R_k := F(x_k) + d^2(y_k, y^*(x_k))$. Then

$$\mathbb{E}[R_{k+1} - R_k]$$

$$\leq \mathbb{E}[F(x_{k+1}) - F(x_k)] + \mathbb{E}[d^2(y_{k+1}, y^*(x_{k+1}) - d^2(y_k, y^*(x_k))]$$

$$\leq -\Big(\frac{\eta_x}{2} - \eta_x^2 L_F - 4\eta_x^2\kappa_l^2\Big)\mathbb{E}\|\mathcal{G}F(x_k)\|_{x_k}^2 + \Big(\frac{\eta_x}{2} + \eta_x^2 L_F + 4\eta_x^2\kappa_l^2\Big)\mathbb{E}\|\mathcal{G}F(x_k) - \widehat{\mathcal{G}}F(x_k)\|_{x_k}^2$$

$$+ \Big((2(1 + \eta_y^2\zeta L^2 - \eta_y\mu)^S - 1)\mathbb{E}d^2(y_k, y^*(x_k)) + \frac{2\eta_y\zeta M^2}{\mu - \eta_y\zeta L^2}\frac{1}{|\mathcal{B}_1|}\Big)$$

$$= -\frac{1}{80L_F}\mathbb{E}\|\mathcal{G}F(x_k)\|_{x_k}^2 + \frac{3}{80L_F}\mathbb{E}[\mathbb{E}_k\|\mathcal{G}F(x_k) - \widehat{\mathcal{G}}F(x_k)\|_{x_k}^2]$$

$$+ \Big((2(1 + \eta_y^2\zeta L^2 - \eta_y\mu)^S - 1)\mathbb{E}d^2(y_k, y^*(x_k)) + \frac{2\eta_y\zeta M^2}{\mu - \eta_y\zeta L^2}\frac{1}{|\mathcal{B}_1|}\Big)$$

$$\leq -\frac{1}{80L_F}\mathbb{E}\|\mathcal{G}F(x_k)\|_{x_k}^2 + \frac{3}{80L_F}\Big(\frac{4M^2 + 16M^2\kappa_l^2}{|\mathcal{B}_2|} + \frac{8M^2\kappa_l^2}{|\mathcal{B}_3|} + \frac{16M^2\kappa_l^2}{|\mathcal{B}_4|}\Big) + \frac{3C_{\mathrm{hinv}}^2}{40L_F}\mathbb{E}d^2(y_{k+1}, y^*(x_k))$$

$$+ \Big((2(1 + \eta_y^2\zeta L^2 - \eta_y\mu)^S - 1)\mathbb{E}d^2(y_k, y^*(x_k)) + \frac{2\eta_y\zeta M^2}{\mu - \eta_y\zeta L^2}\frac{1}{|\mathcal{B}_1|}\Big)$$

$$\leq -\frac{1}{80L_F}\mathbb{E}\|\mathcal{G}F(x_k)\|_{x_k}^2 + \Big((2 + \frac{3C_{\mathrm{hinv}}^2}{40L_F})(1 + \eta_y^2\zeta L^2 - \eta_y\mu)^S - 1\Big)\mathbb{E}d^2(y_k, y^*(x_k)) +$$

$$+ \frac{3}{80L_F}\Big(\frac{4M^2 + 16M^2\kappa_l^2}{|\mathcal{B}_2|} + \frac{8M^2\kappa_l^2}{|\mathcal{B}_3|} + \frac{16M^2\kappa_l^2}{|\mathcal{B}_4|}\Big) + \Big(\frac{3C_{\text{hinv}}^2}{40L_F} + 2\Big)\frac{\eta_y\zeta M^2}{\mu - \eta_y\zeta L^2}\frac{1}{|\mathcal{B}_1|}$$

$$\leq -\frac{1}{80L_F}\mathbb{E}\|\mathcal{G}F(x_k)\|_{x_k}^2 + \frac{3}{80L_F}\Big(\frac{4M^2 + 16M^2\kappa_l^2}{|\mathcal{B}_2|} + \frac{8M^2\kappa_l^2}{|\mathcal{B}_3|} + \frac{16M^2\kappa_l^2}{|\mathcal{B}_4|}\Big) + \Big(\frac{3C_{\text{hinv}}^2}{40L_F} + 2\Big)\frac{\eta_y\zeta M^2}{\mu - \eta_y\zeta L^2}\frac{1}{|\mathcal{B}_1|}$$

where we choose $\eta_x = \frac{1}{20L_F}$ in the first equality and $S \geq \log(\frac{40L_F}{80L_F + 3C_{\text{hinv}}^2})/\log(1 + \eta_y^2\zeta L^2 - \eta_y\mu) = \widetilde{\Theta}(\kappa_l^2\zeta)$ for the last inequality. Telescoping the result gives

$$\frac{1}{K}\sum_{k=0}^{K-1}\mathbb{E}\|\mathcal{G}F(x_k)\|_{x_k}^2 \leq \frac{80L_F R_0}{K} + \Big(\frac{12M^2 + 48M^2\kappa_l^2}{|\mathcal{B}_2|} + \frac{24M^2\kappa_l^2}{|\mathcal{B}_3|} + \frac{48M^2\kappa_l^2}{|\mathcal{B}_4|}\Big)$$
$$+ (6C_{\text{hinv}}^2 + 160L_F)\frac{\eta_y\zeta M^2}{\mu - \eta_y\zeta L^2}\frac{1}{|\mathcal{B}_1|}$$
$$\leq \frac{80L_F R_0}{K} + \frac{\epsilon}{2},$$

where the last inequality follows from the choice that $|\mathcal{B}_1| \geq (24C_{\text{hinv}}^2 + 640L_F)\frac{8\eta_y\zeta M^2}{\mu - \eta_y\zeta L^2}/\epsilon = \Theta(\kappa_l^4/\epsilon)$, $|\mathcal{B}_2| \geq \frac{144M^2 + 576M^2\kappa_l^2}{\epsilon} = \Theta(\kappa_l^2/\epsilon)$, $|\mathcal{B}_3| \geq \frac{288M^2\kappa_l^2}{\epsilon} = \Theta(\kappa_l^2/\epsilon)$, $|\mathcal{B}_4| \geq \frac{576M^2\kappa_l^2}{\epsilon} = \Theta(\kappa_l^2/\epsilon)$ in the last inequality.

In order to reach $\epsilon$-stationary solution, we require $K = O(\kappa_l^3\epsilon^{-1})$ and thus the (stochastic) gradient complexity for $f$ is $G_f = 2K|\mathcal{B}_2| = O(\kappa_l^5\epsilon^{-2})$ and for $g$ is $G_g = KS|\mathcal{B}_1| = \widetilde{O}(\kappa_l^9\zeta\epsilon^{-2})$. The complexity for cross-derivative is $K|\mathcal{B}_3| = O(\kappa_l^5\epsilon^{-2})$. $\qquad\square$

# E Proofs for Section 3.4

*Proof of Theorem 3.* We first give a complete proof for the Hessian inverse estimator as follows. For the other estimators, we only provide a proof sketch.

**Proof for HINV.** (1) First, we derive the convergence under strong convexity using retraction. By the trigonometric distance bound

$$d^2(y_k^{s+1}, y^*(x_k))$$
$$\leq d^2(y_k^s, y^*(x_k)) + \zeta d^2(y_k^s, y_k^{s+1}) - 2\langle \text{Exp}_{y_k^s}^{-1}y_k^{s+1}, \text{Exp}_{y_k^s}^{-1}y^*(x_k)\rangle_{y_k^s}$$
$$\leq d^2(y_k^s, y^*(x_k)) + \eta_y^2\zeta\bar{c}\|\mathcal{G}_y g(x_k, y_k^s)\|_{y_k^s}^2 - 2\langle\text{Exp}_{y_k^s}^{-1}y_k^{s+1} - \text{Retr}_{y_k^s}^{-1}y_k^{s+1}, \text{Exp}_{y_k^s}^{-1}y^*(x_k)\rangle_{y_k^s}$$
$$\quad + 2\eta_y\langle\mathcal{G}_y g(x_k, y_k^s), \text{Exp}_{y_k^s}^{-1}y^*(x_k)\rangle_{y_k^s}$$
$$\leq d^2(y_k^s, y^*(x_k)) + \eta_y^2\zeta\bar{c}\|\mathcal{G}_y g(x_k, y_k^s)\|_{y_k^s}^2 + 2\eta_y\langle\mathcal{G}_y g(x_k, y_k^s), \text{Exp}_{y_k^s}^{-1}y^*(x_k)\rangle_{y_k^s}$$
$$\quad + 2\bar{D}\|\text{Exp}_{y_k^s}^{-1}y_k^{s+1} - \text{Retr}_{y_k^s}^{-1}y_k^{s+1}\|_{y_k^s}$$
$$\leq d^2(y_k^s, y^*(x_k)) + \eta_y^2(\zeta\bar{c} + 2\bar{D}c_R)\|\mathcal{G}_y g(x_k, y_k^s)\|_{y_k^s}^2 + 2\eta_y\langle\mathcal{G}_y g(x_k, y_k^s), \text{Exp}_{y_k^s}^{-1}y^*(x_k)\rangle_{y_k^s}$$
$$\leq \big(1 + \eta_y^2(\zeta\bar{c} + 2\bar{D}c_R)L^2 - \mu\eta_y\big)d^2(y_k^s, y^*(x_k))$$

where we use Assumption 5 in the second inequality and fourth inequality. We require $\eta_y < \frac{\mu}{(\zeta\bar{c} + 2\bar{D}c_R)L^2}$ in order to achieve linear convergence. For simplicity, we let $\tau = \mu\eta_y - \eta_y^2(\zeta\bar{c} + 2\bar{D}c_R)$. This leads to $d^2(y_k^{s+1}, y^*(x_k)) \leq (1 - \tau)d^2(y_k^s, y^*(x_k))$.

(2) Next, we notice the bound on hypergradient approximation error still holds as $\|\widehat{\mathcal{G}}_{\text{hinv}}F(x_k) - \mathcal{G}F(x_k)\|_{x_k} \leq C_{\text{hinv}}d\big(y^*(x_k), y_{k+1}\big)$, where $C_{\text{hinv}} = L + \kappa_\rho M + \kappa_l L + \kappa_l\kappa_\rho M$. Further, by $L$-smoothness,

$$F(x_{k+1}) - F(x_k)$$
$$\leq \langle\mathcal{G}F(x_k), \text{Exp}_{x_k}^{-1}x_{k+1}\rangle_{x_k} + \frac{L_F}{2}d^2(x_k, x_{k+1})$$

$$\leq \langle \mathcal{G}F(x_k), \mathrm{Exp}_{x_k}^{-1} x_{k+1} - \mathrm{Retr}_{x_k}^{-1} x_{k+1} \rangle_{x_k} - \eta_x \langle \mathcal{G}F(x_k), \widehat{\mathcal{G}}F(x_k) \rangle_{x_k} + \frac{\bar{c} L_F \eta_x^2}{2} \|\widehat{\mathcal{G}}F(x_k)\|_{x_k}^2$$

$$\leq \left(2\kappa_l M c_R + \frac{\bar{c} L_F}{2}\right) \eta_x^2 \|\widehat{\mathcal{G}}F(x_k)\|_{x_k}^2 - \eta_x \langle \mathcal{G}F(x_k), \widehat{\mathcal{G}}F(x_k) \rangle_{x_k}$$

$$\leq (4\kappa_l M c_R + \bar{c} L_F)\eta_x^2 \|\widehat{\mathcal{G}}F(x_k) - \mathcal{G}F(x_k)\|_{x_k}^2 + (4\kappa_l M c_R + \bar{c} L_F)\eta_x^2 \|\mathcal{G}F(x_k)\|_{x_k}^2$$

$$\quad + \frac{\eta_x}{2} \|\mathcal{G}F(x_k) - \widehat{\mathcal{G}}F(x_k)\|_{x_k}^2 - \frac{\eta_x}{2} \|\mathcal{G}F(x_k)\|_{x_k}^2$$

$$= -\left(\frac{\eta_x}{2} - (4\kappa_l M c_R + \bar{c} L_F)\eta_x^2\right)\|\mathcal{G}F(x_k)\|_{x_k}^2 + \left(\frac{\eta_x}{2} + (4\kappa_l M c_R + \bar{c} L_F)\eta_x^2\right)\|\mathcal{G}F(x_k) - \widehat{\mathcal{G}}F(x_k)\|_{x_k}^2.$$

where in the third inequality, we bound $\|\mathcal{G}F(x_k)\|_{x_k} \leq M + \frac{L}{\mu}M \leq \frac{2LM}{\mu}$.

(3) Then we can bound

$$d^2(y_k, y^*(x_k)) \leq 2d^2(y_{k-1}^S, y^*(x_{k-1})) + 2d^2(y^*(x_k), y^*(x_{k-1}))$$

$$\leq 2(1-\tau)^S d^2(y^*(x_{k-1}), y_{k-1}) + 2\eta_x^2 \kappa_l^2 \bar{c}\|\widehat{\mathcal{G}}_{\mathrm{hinv}}F(x_{k-1})\|_{x_k}^2$$

$$\leq 2(1 + 2\eta_x^2 \kappa_l^2 C_{\mathrm{hinv}}^2 \bar{c})(1-\tau)^S d^2(y^*(x_{k-1}), y_{k-1}) + 4\eta_x^2 \kappa_l^2 \bar{c}\|\mathcal{G}F(x_{k-1})\|_{x_{k-1}}^2,$$

where the last inequality follows similarly as (9).

Let a Lyapunov function be $R_k := F(x_k) + d^2(y_k, y^*(x_k))$. Then

$$R_{k+1} - R_k$$

$$\leq -\left(\frac{\eta_x}{2} - (4\kappa_l M c_R + \bar{c} L_F)\eta_x^2\right)\|\mathcal{G}F(x_k)\|_{x_k}^2 + \left(\frac{\eta_x}{2} + (4\kappa_l M c_R + \bar{c} L_F)\eta_x^2\right)\|\mathcal{G}F(x_k) - \widehat{\mathcal{G}}F(x_k)\|_{x_k}^2$$

$$\quad + \left(\left((2 + 4\eta_x^2 \kappa_l^2 C_{\mathrm{hinv}}^2 \bar{c})(1-\tau)^S - 1\right)d^2(y^*(x_k), y_k) + 4\eta_x^2 \kappa_l^2 \bar{c}\|\mathcal{G}F(x_k)\|_{x_k}^2\right)$$

$$\leq -\left(\frac{\eta_x}{2} - (4\kappa_l M c_R + \bar{c} L_F)\eta_x^2 - 4\eta_x^2 \kappa_l^2 \bar{c}\right)\|\mathcal{G}F(x_k)\|_{x_k}^2$$

$$\quad + \left(\left(2 + C_{\mathrm{hinv}}^2(\frac{\eta_x}{2} + (4\kappa_l M c_R + \bar{c} L_F)\eta_x^2) + 4\eta_x^2 \kappa_l^2 C_{\mathrm{hinv}}^2 \bar{c}\right)(1-\tau)^S - 1\right)d^2(y^*(x_k), y_k)$$

$$\leq -\left(\frac{\eta_x}{2} - \tilde{L}_F \eta_x^2\right)\|\mathcal{G}F(x_k)\|_{x_k}^2 + \left(\left(2 + C_{\mathrm{hinv}}^2(\frac{\eta_x}{2} + \tilde{L}_F \eta_x^2)\right)(1-\tau)^S - 1\right)d^2(y^*(x_k), y_k)$$

$$\leq -\frac{1}{16\tilde{L}_F}\|\mathcal{G}F(x_k)\|_{x_k}^2$$

where we use $\kappa_l^2 \bar{c} \leq L_F \bar{c}$ and let $\tilde{L}_F := 4\kappa_l c_R M + 5\bar{c} L_F$ in the second last inequality, and we choose $\eta_x = \frac{1}{4\tilde{L}_F}$, $S \geq \log\left(\frac{16\tilde{L}_F}{32\tilde{L}_F + 3C_{\mathrm{hinv}}^2}\right)/\log(1-\tau) = \widetilde{\Theta}(\kappa_l^2 \zeta)$, in the last inequality. Then telescoping the results yields Finally, we sum over $k = 0, ..., K-1$, which leads to

$$\frac{1}{K}\sum_{k=0}^{K-1}\|\mathcal{G}F(x_k)\|_{x_k}^2 \leq \frac{16\tilde{L}_F R_0}{K}.$$

Thus in order to achieve $\epsilon$-stationary solution, we require $K = O(\tilde{L}_F \epsilon^{-1}) = O(\kappa_l^3 \epsilon^{-1})$ and hence the order of gradient and second-order complexities remain unchanged.

**Extensions to other estimators.** To extend the proof to other hypergradient estimators, we first notice that the convergence of inner iterations for solving lower-level problems is agnostic to the choice of hypergradient estimators, i.e.,

$$d^2\left(y_k^{s+1}, y^*(x_k)\right) \leq (1-\tau)d^2\left(y_k^s, y^*(x_k)\right), \quad \tau = \mu\eta_y - \eta_y^2(\zeta\bar{c} + 2Dc_R).$$

(1) *Hypergradient approximation error.* For hypergradient estimator based on Hessian inverse, conjugate gradient, truncated Neumann series, the hypegradient approximation error remains the same as in Lemma 1, given no retraction is involved in the computation. That is,

- **CG**: $\|\widehat{\mathcal{G}}_{\mathrm{cg}}F(x_k) - \mathcal{G}F(x_k)\|_{x_k} \leq C_{\mathrm{cg}}d(y^*(x_k), y_{k+1}) + 2L\sqrt{\kappa_l}\left(\frac{\sqrt{\kappa_l}-1}{\sqrt{\kappa_l}+1}\right)^T\|\hat{v}_k^0 - \Gamma_{y^*(x_k)}^{y_{k+1}}v_k^*\|_{y_{k+1}}$,
  where $C_{\mathrm{cg}} = L + \kappa_\rho M + L(1 + 2\sqrt{\kappa_l})(\kappa_l + \frac{M\kappa_\rho}{\mu})$.

- **NS**: $\|\widehat{\mathcal{G}}_{\text{ns}}F(x_k) - \mathcal{G}F(x_k)\|_{x_k} \leq C_{\text{ns}}d(y^*(x_k), y_{k+1}) + \kappa_l M(1-\gamma\mu)^T$, where $C_{\text{ns}} = C_{\text{hinv}}$.

For hypergradient based on automatic differentiation (**AD**), we first show that there exists a constant $C_4$ (that depends on $C_3, \bar{c}, c_R$) such that $D_x\text{Retr}_x(u) = \mathcal{P}_{\text{Retr}_x(u)}(\text{id} + D_x u) + \mathcal{E}$. with $\|\mathcal{E}\|_{\text{Retr}_x(u)} \leq C_4 \|D_x u\|_x \|u\|_x$. Such a result can be derived by bounding the difference between retraction and exponential map. Then we follow the analysis for Lemma 1 as follows. Given $y_k^{s+1} = \text{Retr}_{y_k^s}(-\eta_y \mathcal{G}_y g(x_k, y_k^s))$, we have

$$D_{x_k} y_k^{s+1} = \mathcal{P}_{y_k^{s+1}}\big((\text{id} - \eta_y \mathcal{H}_y g(x_k, y_k^s))D_{x_k} y_k^s - \eta_y \mathcal{G}_{yx}^2 g(x_k, y_k^s)\big) + \mathcal{E}_k^s$$

where $\|\mathcal{E}_k^s\|_{y_k^{s+1}} \leq \eta_y^2 C_4\big((1-\eta_y\mu)C_1 + \eta_y L\big)\|\mathcal{G}_y f(x_k, y_k^s)\|_{y_k^s}$. The rest of the proof follows exactly from Lemma 1, where we replace the convergence of inner iteration with the updated rate. This gives

$$\|\widehat{\mathcal{G}}_{\text{ad}}F(x_k) - \mathcal{G}F(x_k)\|_{x_k}$$
$$\leq \Big(\frac{2M\widetilde{C}'}{\mu - \eta_y(\zeta\bar{c} + 2Dc_R)} + L(1+\kappa_l)\Big)(1-\tau)^{\frac{S-1}{2}}d(y_k, y^*(x_k)) + M\kappa_l(1-\eta_y\mu)^S,$$

where $\widetilde{C}' := (\kappa_l + 1)\rho + (C_2 + \eta_y C_4)L\big((1-\eta_y\mu)C_1 + \eta_y L\big)$

In summary

- **AD**: $\|\widehat{\mathcal{G}}_{\text{ad}}F(x_k) - \mathcal{G}F(x_k)\|_{x_k} \leq C_{\text{ad}}(1-\tau)^{\frac{S-1}{2}}d(y_k, y^*(x_k)) + M\kappa_l(1-\eta_y\mu)^S$, where $C_{\text{ad}} := \frac{2M\widetilde{C}'}{\mu - \eta_y(\zeta\bar{c} + 2Dc_R)} + L(1+\kappa_l)$.

(2) *Objective decrement.* This part is also the same across all hypergradient estimators, i.e.,

$$F(x_{k+1}) - F(x_k)$$
$$\leq -\Big(\frac{\eta_x}{2} - (4\kappa_l Mc_R + \bar{c}L_F)\eta_x^2\Big)\|\mathcal{G}F(x_k)\|_{x_k}^2 + \Big(\frac{\eta_x}{2} + (4\kappa_l Mc_R + \bar{c}L_F)\eta_x^2\Big)\|\mathcal{G}F(x_k) - \widehat{\mathcal{G}}F(x_k)\|_{x_k}^2$$

(3) *Lyapunov function decrement.* The definition of Lyapunov function depends on the choice of hypergradient estimator.

For **CG**, we define $R_k := F(x_k) + d^2(y_k, y^*(x_k)) + \|\hat{v}_k^0 - \Gamma_{y^*(x_k)}^{y_{k+1}} v_k^*\|_{y_{k+1}}^2$. Then following similar analysis, we first bound

$$d^2(y_{k+1}, y^*(x_{k+1})) \leq (2 + 8\eta_x^2\kappa_l^2 C_{\text{cg}}^2)(1-\tau)^S d^2(y^*(x_k), y_k)$$
$$+ 32\eta_x^2\kappa_l^3 L^2\Big(\frac{\sqrt{\kappa_l}-1}{\sqrt{\kappa_l}+1}\Big)^{2T}\|\hat{v}_k^0 - \Gamma_{y^*(x_k)}^{y_{k+1}} v_k^*\|_{y_{k+1}}^2 + 4\eta_x^2\kappa_l^2\bar{c}\|\mathcal{G}F(x_k)\|_{x_k}^2.$$
$$\tag{19}$$

and

$$\|\hat{v}_k^0 - \Gamma_{y^*(x_k)}^{y_{k+1}} v_k^*\|_{y_{k+1}}$$
$$\leq 2\sqrt{\kappa_l}\Big(\frac{\sqrt{\kappa_l}-1}{\sqrt{\kappa_l}+1}\Big)^T\|\hat{v}_{k-1}^0 - \Gamma_{y^*(x_{k-1})}^{y_k} v_{k-1}^*\|_{y_k} + 2\sqrt{\kappa_l}\Big(\kappa_l + \frac{M\kappa_\rho}{\mu}\Big)(1-\tau)^{\frac{S}{2}}d(y^*(x_{k-1}), y_{k-1})$$
$$+ \big\|v_k^* - \Gamma_{y^*(x_{k-1})}^{y^*(x_k)} v_{k-1}^*\big\|_{y_k} + \frac{2MC_0 D}{\mu}d(y_k, y^*(x_k))$$

Then similarly,

$$\|v_k^* - \Gamma_{y^*(x_{k-1})}^{y^*(x_k)} v_{k-1}^*\|_{y^*(x_k)} \leq C_v d(x_k, x_{k-1})$$
$$\leq \eta_x\bar{c}C_v\|\widehat{\mathcal{G}}_{\text{cg}}F(x_{k-1}) - \mathcal{G}F(x_{k-1})\|_{x_{k-1}} + \eta_x\bar{c}C_v\|\mathcal{G}F(x_{k-1})\|_{x_{k-1}}$$

where in the second inequality, we use the bound between retraction and exponential map as well as triangle inequality. Then combining the above two results, gives

$$\|\hat{v}_k^0 - \Gamma_{y^*(x_k)}^{y_{k+1}} v_k^*\|_{y_{k+1}}^2$$

$$\leq 20\kappa_l\big(\frac{\sqrt{\kappa_l}-1}{\sqrt{\kappa_l}+1}\big)^{2T}\|\hat{v}_{k-1}^0 - \Gamma_{y^*(x_{k-1})}^{y_k}v_{k-1}^*\|_{y_k}^2 + 20\kappa_l(\kappa_l + \frac{M\kappa_\rho}{\mu})^2(1-\tau)^S d(y^*(x_{k-1}), y_{k-1})$$

$$+ 5\bar{c}\eta_x^2 C_v^2\|\widehat{\mathcal{G}}_{\text{cg}}F(x_{k-1}) - \mathcal{G}F(x_{k-1})\|_{x_{k-1}}^2 + 5\bar{c}\eta_x^2 C_v^2\|\mathcal{G}F(x_{k-1})\|_{x_{k-1}}^2 + \frac{5M^2C_0^2D^2}{\mu^2}d^2(y_k, y^*(x_k)).$$

Then we can show

$R_{k+1} - R_k$

$$\leq -\big(\frac{\eta_x}{2} - \eta_x^2\tilde{L}_F - 5\eta_x^2\bar{c}C_v^2\big)\|\mathcal{G}F(x_k)\|_{x_k}^2 + \big(\frac{\eta_x}{2} + \eta_x^2\tilde{L}_F + 5\eta_x^2\bar{c}C_v^2\big)\|\mathcal{G}F(x_k) - \widehat{\mathcal{G}}_{\text{cg}}F(x_k)\|_{x_k}^2$$

$$+ \big(\frac{5M^2C_0^2D^2}{\mu^2} + 1\big)d^2(y_{k+1}, y^*(x_{k+1})) + \big(20\kappa_l(\kappa_l + \frac{M\kappa_\rho}{\mu})^2(1-\tau)^S - 1\big)d^2(y^*(x_k), y_k)$$

$$+ \big(20\kappa_l\big(\frac{\sqrt{\kappa_l}-1}{\sqrt{\kappa_l}+1}\big)^{2T} - 1\big)\|\hat{v}_k^0 - \Gamma_{y^*(x_k)}^{y_{k+1}}v_k^*\|_{y_{k+1}}^2$$

$$\leq -\big(\frac{\eta_x}{2} - 6\eta_x^2\widetilde{\Lambda}\big)\|\mathcal{G}F(x_k)\|_{x_k}^2 + \big(\frac{\eta_x}{2} + 6\eta_x^2\widetilde{\Lambda}\big)\|\mathcal{G}F(x_k) - \widehat{\mathcal{G}}_{\text{cg}}F(x_k)\|_{x_k}^2$$

$$+ \big(\big(\big(\frac{5M^2C_0^2D^2}{\mu^2} + 1\big)(2 + 8\eta_x^2\kappa_l^2C_{\text{cg}}^2) + 20\kappa_l\big(\kappa_l + \frac{M\kappa_\rho}{\mu}\big)^2\big)(1 + \eta_y^2\zeta L^2 - \eta_y\mu)^S - 1\big)d^2(y_k, y^*(x_k))$$

$$+ \big(\big(32\eta_x^2\kappa_l^3 L^2(\frac{5M^2C_0^2D^2}{\mu^2} + 1) + 20\kappa_l\big)\big(\frac{\sqrt{\kappa_l}-1}{\sqrt{\kappa_l}+1}\big)^{2T} - 1\big)\|\hat{v}_k^0 - \Gamma_{y^*(x_k)}^{y_{k+1}}v_k^*\|_{y_{k+1}}^2$$

$$\leq -\frac{1}{96\widetilde{\Lambda}}\|\mathcal{G}F(x_k)\|_{x_k}^2$$

where we let $\widetilde{\Lambda} := C_v^2\bar{c} + \kappa_l^2(\frac{5M^2C_0^2D^2}{\mu} + \bar{c})$ and without loss of generality $\tilde{L}_F \leq \widetilde{\Lambda}$. The second inequality is by (19). The last inequality is by appropriately choosing $S, T$, which is on the same order as the exponential map case. Then telescoping the result yields

$$\frac{1}{K}\sum_{k=0}^{K-1}\|\mathcal{G}F(x_k)\|_{x_k}^2 \leq \frac{96\widetilde{\Lambda}R_0}{K}.$$

For **NS**, we define $R_k = F(x_k) + d^2(y_k, y^*(x_k))$ and derive

$$d^2(y_{k+1}, y^*(x_{k+1})) \leq 2(1-\tau)^S d^2(y^*(x_k), y_k) + 4\eta_x^2\kappa_l^2\bar{c}\|\widehat{\mathcal{G}}_{\text{ns}}F(x_k) - \mathcal{G}F(x_k)\|_{x_k}^2 + 4\eta_x^2\kappa_l^2\bar{c}\|\mathcal{G}F(x_k)\|_{x_k}^2$$

Then we can follow exactly the same proof as HINV that

$$\frac{1}{K}\sum_{k=0}^{K-1}\|\mathcal{G}F(x_k)\|_{x_k}^2 \leq \frac{16\tilde{L}_F R_0}{K} + \frac{\epsilon}{2}$$

by appropriately choosing $S, T$ and $\eta_x$.

For **AD**, we define $R_k := F(x_k) + d^2(y_k, y^*(x_k))$. Then following the same analysis except for the choice of $S$, we can show,

$$\frac{1}{K}\sum_{k=0}^{K-1}\|\mathcal{G}F(x_k)\|_{x_k}^2 \leq \frac{16\tilde{L}_F R_0}{K} + \frac{\epsilon}{2}$$

Thus the proof is now complete. $\qquad\square$

# F  Tangent space conjugate gradient

In Algorithm 3, we show the tangent space conjugate gradient algorithm for solving the linear system $\mathcal{H}[v] = \mathcal{G}$. Similar to [40], we set the initialization to be the transported output of $\hat{v}_{k-1}^T$ from last iteration, where $\hat{v}_{-1}^T = 0$, which is beneficial for convergence analysis. For practical purposes, we notice setting $v_0 = 0$ provides sufficient accurate solution without the expensive parallel transport operation.

---

**Algorithm 3** Tangent space conjugate gradient TSCG($\mathcal{H}, \mathcal{G}, v_0, T$)

---

1: Set $r_0 = \mathcal{G} \in T_x\mathcal{M}, p_0 = r_0$.
2: **for** $t = 0, ..., T - 1$ **do**
3:     Compute $\bar{r}_{t+1} = \mathcal{H}[v_t]$.
4:     $\alpha_{t+1} = \frac{\|r_t\|_x^2}{\langle p_t, \mathcal{H}[p_t]\rangle_x}$.
5:     $v_{t+1} = v_t + \alpha_{t+1}p_t$.
6:     $r_{t+1} = r_t - \alpha_{t+1}\mathcal{H}[p_t]$.
7:     $\beta_{t+1} = \frac{\|r_{t+1}\|_x^2}{\|r_t\|_x^2}$.
8:     $p_{t+1} = r_{t+1} + \beta_{t+1}p_t$.
9: **end for**
10: **Output**: $v_T$

---

**Algorithm 4** Riemannian bilevel solver for min-max optimization

---

1: Initialize $x_0 \in \mathcal{M}_x, y_0 \in \mathcal{M}_y$.
2: **for** $k = 0, ..., K - 1$ **do**
3:     $y_k^0 = y_k$.
4:     **for** $s = 0, ..., S - 1$ **do**
5:         $y_k^{s+1} = \text{Exp}_{y_k^s}(-\eta_y \mathcal{G}_y g(x_k, y_k^s))$.
6:     **end for**
7:     Update $x_{k+1} = \text{Exp}_{x_k}\left(-\eta_x \mathcal{G}_x f(x_k, y_{k+1})\right)$, where $y_{k+1} = y_k^S$.
8: **end for**

---

# G    Extensions: on Riemannian mix-max and compositional optimization

The bilevel optimization considered in the paper (1) generalizes the two other widely studied problems, namely the min-max optimization and compositional optimization.

## G.1    Min-max optimization on Riemannian manifolds

Riemannian min-max problems have gained increasing interest over the recent years [37, 41, 27, 73, 67, 25, 56, 35], which takes the form of

$$\min_{x \in \mathcal{M}_x} \max_{y \in \mathcal{M}_y} f(x, y),$$

and can be seen as a special case of bilevel optimization problem (1) where $g(x, y) = -f(x, y)$. Because the problem is nonconvex in $x$, the order of minimization and maximization matters [73, 25]. Nevertheless, under the assumption where $f$ is geodesic strongly convex in $y$, the optimal solution $x^*$ satisfies $\mathcal{G}F(x^*) = 0$, where $\mathcal{G}F(x) = \mathcal{G}_x f(x, y^*(x))$ due to $\mathcal{G}_y f(x, y^*(x)) = \mathcal{G}_y g(x, y^*(x)) = 0$. Thus Algorithm 1 reduces to alternating gradient descent ascent over Riemannian manifolds, as outlined in Algorithm 4.

Here we adapt the convergence analysis to the min-max optimization setting. Given we no longer require second-order derivatives, we restate assumptions for functions $f, g$ below.

**Assumption 6.** (1) Assumption 1 holds. (2) Function $f(x, y), g(x, y)$ have $L$-Lipschitz Riemannian gradients. (3) Further, $g(x, y)$ is $\mu$-geodesic strongly convex in $y$.

Under the min-max setup and Assumption 6, we see $\mathcal{G}F(x) = \mathcal{G}_x f(x, y^*(x))$ and thus the Lipschitz constant can be derived as $L_F = (\kappa_l + 1)L = \Theta(\kappa_l)$. Further we can directly apply Theorem 1 for the Hessian inverse with $C_{\text{hinv}} = L$, which leads to the following convergence result.

**Theorem 4.** *Under Assumption 6, choosing $S \geq \widetilde{\Theta}(\kappa_l^2\zeta)$, $\eta_x = \frac{1}{20L_F}$, we have $\min_{k=0,...,K-1} \|\mathcal{G}F(x_k)\|_{x_k}^2 \leq \frac{80(\kappa_l+1)L\Delta_0}{K}$ and to reach $\epsilon$-stationary solution, we require gradient complexities as $G_f = O(\kappa_l \epsilon^{-1})$ and $G_g = \widetilde{O}(\kappa_l^3 \zeta \epsilon^{-1})$.*

### G.2   Compositional optimization on Riemannian manifolds

Compositional problems on Riemannian manifolds have been considered in [36, 70], which requires to solve

$$\min_{x \in \mathcal{M}_x} \psi(\phi(x)), \tag{20}$$

where $\psi : \mathcal{M}_y \to \mathbb{R}$ and $\phi : \mathcal{M}_x \to \mathcal{M}_y$. It is worth noting that in both works [36, 70], the inner function $\phi : \mathcal{M}_x \to \mathbb{R}^d$ is vector-valued. In contrast, we consider a general manifold-valued function $\phi$. Because the function $\phi$ can be potentially complex and may be stochastic, we follow [11] to reformulate (20) into a bilevel optimization problem by letting

$$f(x, y) := \psi(y^*(x)), \text{ s.t. } y^*(x) = \underset{y \in \mathcal{M}_y}{\arg\min} \{ g(x, y) := \frac{1}{2} d^2(\phi(x), y) \}.$$

As long as the squared Riemannian distance is geodesic strongly convex, the reformulation is equivalent to the original problem (20). As formally stated in Lemma 9, this is satisfied for non-positively curved space, like Euclidean space, hyperbolic manifold, SPD manifold with affine invariant metric. For positively curved space, the strong convexity is guaranteed when restricting the domain relative to the curvature.

**Lemma 9.** *Let $\mathcal{U} \subseteq \mathcal{M}$ has sectional curvature lower and upper bounded by $\kappa^-$ and $\kappa^+$ respectively. Further $\mathcal{U}$ has diameter upper bounded by $\bar{D}$, which satisfies $\bar{D} < \frac{\pi}{\sqrt{\kappa^+}}$ if $\kappa^+ > 0$. Then let $\delta = 1$ when $\kappa^+ \leq 0$ and $\delta = \frac{\sqrt{\kappa^+}\bar{D}}{\tan(\sqrt{\kappa^+}\bar{D})}$ when $\kappa^+ > 0$ and consider $\zeta$ be the same curvature constant as in Lemma 3. Then function $\mathcal{H}_y g(x, y)$ has Riemannian Hessian bounded within $[\delta, \zeta]$ in spectrum.*

*Proof of Lemma 9.* The proof follows from Lemma 2 in [3]. Consider an arbitrary curve $\gamma : [0, 1] \to \mathcal{M}$, and let $f(x) = \frac{1}{2} d^2(x, p)$, for some $p \in \mathcal{M}$. From [3], we know that $\mathcal{H}f(\gamma(t))[\gamma'(t)] = -\boldsymbol{\nabla}_{\gamma'(t)} \mathrm{Exp}_{\gamma(t)}^{-1}(p)$ and under the conditions, $\delta\|\gamma'(t)\|_{\gamma(t)}^2 \leq \langle \boldsymbol{\nabla}_{\gamma'(t)} \mathrm{Exp}_{\gamma(t)}^{-1}(p), -\gamma'(t) \rangle_{\gamma(t)} \leq \zeta\|\gamma'(t)\|_{\gamma(t)}^2$, where we denote $\boldsymbol{\nabla}$ as the covariant derivative. This immediately leads to

$$\delta\|\gamma'(t)\|_{\gamma(t)}^2 \leq \langle \gamma'(t), \mathcal{H}f(\gamma(t))[\gamma'(t)] \rangle_{\gamma(t)} \leq \zeta\|\gamma'(t)\|_{\gamma(t)}^2$$

which completes the proof. $\qquad\qquad\square$

Thus, for positively curved manifold, if $\bar{D} < \frac{\pi}{2\sqrt{\kappa^+}}$, we have $\delta > 0$, which ensures geodesic strong convexity of the inner problem. As shown in Lemma 12 in [3], $\mathcal{G}_y d^2(\phi(x), y) = 2\mathrm{Exp}_y^{-1}\phi(x)$ and the Riemannian gradient descent on $y$ lead to

$$y_k^{s+1} = \mathrm{Exp}_{y_k^s}\big( -\eta_y \mathrm{Exp}_{y_k^s}^{-1}\phi(x_k) \big),$$

which suggests $y_k^{s+1}$ lies on a geodesic that connects $y_k^s$ and $\phi(x_k)$. When $S = 1$ and when the lower-level function $g$ is vector-valued, the algorithm recovers the deterministic version of SCGD [66].

However, unlike in the Euclidean space, the Riemannian Hessian does not simplify to the identity operator, but rather the covariant derivative of inverse exponential map and the cross derivatives $\mathcal{G}_{xy}^2 g(x, y) \neq -(\mathrm{D}\phi(x))^\dagger$.

**Assumption 7.** (1) Assumption 1 holds and further $\bar{D} < \frac{\pi}{2\sqrt{\kappa^+}}$ if $\kappa^+ > 0$. (2) Function $f(x, y)$ has Riemannian gradients that are bounded by $M$ and are $L$-Lipschitz. (3) Function $g$ has $\rho$-Lipschitz Riemannian Hessian and cross derivatives.

We notice that for function $g$ we only require second-order derivatives to be Lipschitz because the first-order Lipschitzness can be inferred from Lemma 9.

**Theorem 5.** *Under Assumption 7, Theorem 1 holds with $L = \zeta, \mu = \delta$.*

To prove the convergence, we only need to show Lemma 4 holds. It can be readily proved from Lemma 9 and Assumption 7 that Lemma 4 holds with $L = \zeta, \mu = \delta$. Hence the convergence follows directly.

## H Experimental details

### H.1 Synthetic problem

We first verify the lower-level problem is geodesic strongly convex.

**Proposition 4.** *For any $\mathbf{A}, \mathbf{B} \succ 0$, function $f(\mathbf{M}) = \langle \mathbf{M}, \mathbf{A} \rangle + \langle \mathbf{M}^{-1}, \mathbf{B} \rangle$ is $\mu$-geodesic strongly convex in $\mathcal{U} \subset \mathbb{S}_{++}^d$ with $\mu = \lambda_{a,-} \lambda_- + \frac{\lambda_{b,-} \lambda_-}{\lambda_+^2}$, where $\lambda_{a,-}, \lambda_{b,-}$ are the minimum eigenvalue of $\mathbf{A}, \mathbf{B}$ and $\lambda_{\pm}$ are the bounds for maximum and minimum eigenvalues for $\mathbf{M} \in \mathcal{U}$.*

*The inverse of Riemannian Hessian of function $f(\mathbf{M}) = \langle \mathbf{M}, \mathbf{A} \rangle + \langle \mathbf{M}^{-1}, \mathbf{B} \rangle$ is derived as, for any symmetric $\mathbf{U}$, $\mathcal{H}^{-1} f(\mathbf{M})[\mathbf{U}] = \mathbf{M}^{1/2} \mathbf{G} \mathbf{M}^{1/2}$ where $\mathbf{G}$ is the solution to the Lyapunov equation $\mathbf{G}(\mathbf{M}^{1/2}\mathbf{A}\mathbf{M}^{1/2} + \mathbf{M}^{-1/2}\mathbf{B}\mathbf{M}^{-1/2}) + (\mathbf{M}^{1/2}\mathbf{A}\mathbf{M}^{1/2} + \mathbf{M}^{-1/2}\mathbf{B}\mathbf{M}^{-1/2})\mathbf{G} = \mathbf{M}^{-1/2}\mathbf{U}\mathbf{M}^{-1/2}$.*

*Proof of Proposition 4.* We first derive the Euclidean gradient and Hessian as

$$\nabla f(\mathbf{M}) = \mathbf{A} - \mathbf{M}^{-1}\mathbf{B}\mathbf{M}^{-1}, \quad \nabla^2 f(\mathbf{M})[\mathbf{U}] = \mathbf{M}^{-1}\mathbf{U}\mathbf{M}^{-1}\mathbf{B}\mathbf{M}^{-1} + \mathbf{M}^{-1}\mathbf{B}\mathbf{M}^{-1}\mathbf{U}\mathbf{M}^{-1}$$

for any $\mathbf{U} = \mathbf{U}^{\top}$. The Riemannian gradient and Hessian are derived as

$$\mathcal{G} f(\mathbf{M}) = \mathbf{M}\mathbf{A}\mathbf{M} - \mathbf{B}$$

$$\mathcal{H} f(\mathbf{M})[\mathbf{U}] = \mathbf{U}\mathbf{M}^{-1}\mathbf{B} + \mathbf{B}\mathbf{M}^{-1}\mathbf{U} + \{\mathbf{U}\nabla f(\mathbf{M})\mathbf{M}\}_{\mathrm{S}}$$

$$= \mathbf{U}\mathbf{M}^{-1}\mathbf{B} + \mathbf{B}\mathbf{M}^{-1}\mathbf{U} + \frac{1}{2}\big(\mathbf{U}\mathbf{A}\mathbf{M} - \mathbf{U}\mathbf{M}^{-1}\mathbf{B} + \mathbf{M}\mathbf{A}\mathbf{U} - \mathbf{B}\mathbf{M}^{-1}\mathbf{U}\big)$$

$$= \frac{1}{2}\big(\mathbf{U}\mathbf{A}\mathbf{M} + \mathbf{M}\mathbf{A}\mathbf{U} + \mathbf{U}\mathbf{M}^{-1}\mathbf{B} + \mathbf{B}\mathbf{M}^{-1}\mathbf{U}\big)$$

where we let $\{\mathbf{A}\}_{\mathrm{S}} = (\mathbf{A} + \mathbf{A}^{\top})/2$. To show the function is geodesic strongly convex, it suffices to show $\mathcal{H} f(\mathbf{M})$ is positive definite, which is to show $\langle \mathcal{H} f(\mathbf{M})[\mathbf{U}], \mathbf{U} \rangle_{\mathbf{M}} \geq \mu \|\mathbf{U}\|_{\mathbf{M}}^2 > 0$ for any $\mathbf{U} = \mathbf{U}^{\top}$. To this end, we vectorize the Riemannian Hessian in terms of $\mathbf{U}$ as $\mathrm{vec}(2\mathcal{H} f(\mathbf{M})[\mathbf{U}]) = (\mathbf{M}\mathbf{A} \otimes \mathbf{I} + \mathbf{I} \otimes \mathbf{M}\mathbf{A} + \mathbf{B}\mathbf{M}^{-1} \otimes \mathbf{I} + \mathbf{I} \otimes \mathbf{B}\mathbf{M}^{-1})\mathrm{vec}(\mathbf{U})$, where $\otimes$ denotes the Kronecker product. Then, we have

$$\langle \mathcal{H} f(\mathbf{M})[\mathbf{U}], \mathbf{U} \rangle_{\mathbf{M}}$$

$$= \mathrm{tr}(\mathbf{M}^{-1}\mathbf{U}\mathbf{M}^{-1}\mathcal{H} f(\mathbf{M})[\mathbf{U}])$$

$$= \frac{1}{2}\mathrm{vec}(\mathbf{U})^{\top}(\mathbf{M}^{-1} \otimes \mathbf{M}^{-1})(\mathbf{M}\mathbf{A} \otimes \mathbf{I} + \mathbf{I} \otimes \mathbf{M}\mathbf{A} + \mathbf{B}\mathbf{M}^{-1} \otimes \mathbf{I} + \mathbf{I} \otimes \mathbf{B}\mathbf{M}^{-1})\mathrm{vec}(\mathbf{U})$$

$$= \frac{1}{2}\mathrm{vec}(\mathbf{U})^{\top}(\mathbf{A} \otimes \mathbf{M}^{-1} + \mathbf{M}^{-1} \otimes \mathbf{A} + \mathbf{M}^{-1}\mathbf{B}\mathbf{M}^{-1} \otimes \mathbf{M}^{-1} + \mathbf{M}^{-1} \otimes \mathbf{M}^{-1}\mathbf{B}\mathbf{M}^{-1})\mathrm{vec}(\mathbf{U})$$

$$\geq (\lambda_{a,-}\lambda_{m,-} + \frac{\lambda_{b,-}\lambda_{m,-}}{\lambda_{m,+}^2})\mathrm{vec}(\mathbf{U})^{\top}(\mathbf{M}^{\top} \otimes \mathbf{M}^{-1})\mathrm{vec}(\mathbf{U}) = \mu\|\mathbf{U}\|_{\mathbf{M}}^2,$$

where we let $\lambda_{a,\pm}$ be the maximum/minimum eigenvalues of $\mathbf{A}$ and similarly for $\lambda_{b,\pm}, \lambda_{m,\pm}$.

The Hessian inverse can be derived subsequently. This completes the proof. $\qquad\square$

### H.2 Computational time for each hypergradient estimator

This section report the average runtime in seconds (over 10 runs) for single evaluation of hypergradient using four different strategies (for Hypergradient estimation) for the synthetic problem (Section 4.1, Figure 1). The hyper-parameters are set to be the same as the main experiment. We see in general automatic differentiation (AD) is the most efficient strategy. Nevertheless, according to Figure 1(a), it is less accurate compared to other strategies.

Table 3: Comparison of runtime for single computation of hypergradient.

| HINV | CG | NS | AD |
|---|---|---|---|
| 0.0154 | 0.1037 | 0.1030 | 0.0053 |

### H.3 Hyperparameter selection

The selection of hyper-parameters is performed to reflect the best performance. The stepsize is selected from the range [1e-3, 5e-3, 1e-2, 5e-2, 1e-1, 5e-1, 1] and for Neumann series is selected from [0.1, 0.5, 1.0, 1.5, 2.0] and number of inner iterations is selected from [5, 10, 30, 50, 100]. Figure 1(d) shows the sensitivity of hypergradient error as we vary and the number of inner iterations.

### H.4 Computational Complexity

This section lists out the computational complexity for each task considered in the experiment section. In Table 4, we present an estimate of the per-iteration complexity of computing the gradient, Hessian/Jacobian-vector products. We highlight that we only provide estimates of the complexities given that there may not exist closed form expressions for the gradient and second-order derivatives.

Here, $n_v, n_t$ denote the size of validation set and training set respectively. For meta learning, $m$ denotes the number tasks and $n$ denotes the number of samples for each task. For domain adaptation, $m, n$ denote the number of samples for two domains, $s$ denotes the number of Sinkhorn iterations.

Table 4: Per-iteration complexity estimate for each task

|          | Hyper-rep (shallow) | Meta learning | Domain adaptation |
|----------|---------------------|---------------|-------------------|
| $x$ size | $d \times r$ | $d \times r$ | $m \times n$ |
| $y$ size | $r(r+1)/2$ | $d \times r$ | $d \times d$ |
| $G_f$ | $O(n_v d^2 r + n_v r^3)$ | $O(mnd^2 r)$ | $O(smn)$ |
| $G_g$ | $O(n_t r^4)$ | $O(mnd^2 r)$ | $O(d^3 + md^2 + nd^2)$ |
| $JV_g$ | $O(n_t(r^4 + d^2 r))$ | $O(mnd^2 r)$ | $O(d^3 + md^2 + nd^2 + smn)$ |
| $HV_g$ | $O(n_t r^4)$ | $O(mndr^2)$ | $O(d^3 + md^2 + nd^2)$ |

## I Experiment Configurations

All the experiments are conducted on a single NVIDIA RTX 4060 GPU. All datasets used in the paper are publicly available, which are properly cited in the main paper. We include detailed setups for the experiments in the main paper as well as documented in code (provided as supplementary material).

## J Broader Impact

This paper proposes new algorithms and are of theoretical in nature. We do not foresee any immediate negative societal impact that we feel obliged to report.

