# OpenReview forum: "A Framework for Bilevel Optimization on Riemannian Manifolds"
_NeurIPS.cc/2024/Conference — NeurIPS 2024 poster_

### Official Review · Reviewer_JBnb · 2024-07-10

**Soundness:** 3
**Presentation:** 3
**Contribution:** 3
**Rating:** 7
**Confidence:** 3

**Summary:**

This paper studies Riemannian bilevel optimization, where variables of both lower and upper level problems are constrained on Riemannian manifolds. The authors propose several hypergradient based algorithms via Neumann series and automatic differentiation.
Convergence analysis is provided for the proposed approaches. Experiments on synthetic problems, hyper-representation learning, meta-learning, unsupervised domain adaptation are provided to demonstrate the effectiveness of the proposed methods.

**Strengths:**

1.	The studied topic is of interest to bilevel optimization research. As far as I know, this should be the first work to study this type of research.
2.	The analysis is comprehensive. The algorithms cover AID and ITD based hypergradient approximations.
3.	Quite a few experiments are provided to support the theory.

**Weaknesses:**

1.	More motivation examples should be provided to validate the importance of bilevel optimization on manifolds. In the experiments, for example, in manifold meta-learning, it would be good to compare the method with meta-learning methods without the orthogonality constraint.
2.	Assumption 1 may be quite strong. It requires all iterates z_1,…, are bounded in a compact space. The analysis in previous standard bilevel optimization does not require this assumption.
3.	Main results contain complex constants and are a little bit hard to parse. Some simplifications may be helpful here.
4.	The extension in 3.3 will be more interesting if Hessian-vector rather than Hessian-inverse is considered.

**Questions:**

See weakness for information.

**Limitations:**

See weakness for information.

---

> ### Author Rebuttal · Authors · 2024-08-06
>
> We thank the reviewer for appreciating the contributions of our work in terms of comprehensive analysis and supportive experiments, as well as providing the constructive comments.
>
>
>
> **1. (W1) More examples to validate the importance of bilevel optimization on manifolds.**
>
> Thank you for the suggestion. We believe  we have provided sufficient motivating applications, showcasing the importance of bilevel optimization on manifolds. One such example is in Section 4.4 that motivates the use of bilevel optimization formulation for unsupervised domain adaptation (under the optimal transport framework). As far as we know, this is a novel application of bilevel optimization to domain adaptation, where we learn a whitening matrix $\textbf{M}$ in the lower-level problem. Our numerical experiments show that employing Riemannian bilevel optimization yields more suitable adaptation by accounting for the metric structure (through whitening) in unsupervised domain adaptation.
>
>
>
>
>
> **2. (W2) Assumption 1 is strong and is non-standard in previous bilevel optimization.**
>
> Standard bilevel optimization in the Euclidean space does not require this assumption. However, for optimization on manifolds, especially when dealing with (geodesic) strongly convex objectives, such an assumption is standard and often unavoidable. See for example [21, 37, 62, 63, 64]. This assumption is to ensure bounded curvature, thus allowing to show fast convergence under strong convexity.
>
>
>
> **3. (W3) Simplification regarding complex constants.**
>
> Thank you for the suggestion. We will simplify the notations in our revision.
>
>
>
> **4. (W4) Extend Hessian-inverse to Hessian-vector in Section 3.3.**
>
> We agree that a full-spectrum analysis covering other hypergradient estimators (in particular the ones using Hessian-vector products) would be interesting. The analysis techniques would be similar to what we already did for Hessian inverse. In order to make the contents concise, we had decided to omit the analysis for other hypergradient estimation strategies. In the revised version, however, we would include a discussion in this regard.

---

> > ### Comment · Reviewer_JBnb · 2024-08-12
> >
> > I thank the authors for the response. My concerns are addressed. I keep my score.

---

> ### Comment · Reviewer_JBnb · 2024-08-13
>
> On Assumption 1:
>
> I agreed with the concerns raised by Reviewer 4rfq regarding Assumption 1, which is typically a strong assumption in bilevel optimization. As I am not very familiar with the literature on manifold optimization, I am not very confident in validating the correctness of this assumption. Hope the authors can provide more evidence here. I found several related works on bilevel optimization on manifolds, e.g., [1][2]. I wonder whether similar assumptions are also made there.
>
> [1] Riemannian Bilevel Optimization, Li and Ma, 2024.
> [2] Riemannian Bilevel Optimization, Sanchayan Dutta, Xiang Cheng, Suvrit Sra, 2024.

---

> > ### Author Response · Authors · 2024-08-13
> >
> > Thank you for your further comments. We kindly refer you to our responses to Reviewer 4rfq regarding clarifications on Assumption 1. In summary, we have revised Assumption 1 to only require domain compactness and unique geodesic for the lower-level problem. Such an assumption is unavoidable in the Riemannian optimization literature.
> >
> >
> > Regarding your question on whether similar assumptions are used in [1,2], **yes, both [1,2] use assumption on the bounded domain.** Specifically:
> >
> > - [1] states explicitly in Assumption 4.2 that $\tau(\iota, \text{dist}(y^{k,t}, y^*(x^k)))$ is bounded by $\tau$ for all $k, t$. Because by the definition of $\tau(\iota, c) = \frac{\sqrt{|\iota| } c}{\tanh(\sqrt{|\iota|}c)}$, $\tau(\iota, c) $ is an increasing function over $\iota, c$. Thus bounded $\tau(\iota, c)$ suggests a bounded domain, i.e., $\text{dist}(y^{k,t}, y^*(x^k)) \leq D$ for all $k, t$ for some constant $D$.
> >
> >
> > - [2] also requires such an assumption, which can be seen from Page 30 where they also require a constant upper bound on $\tau(\kappa, d_{\mathcal{N}} (z_k^{(t)}, y_k^*) )$ for all $k, t$, which implies a bounded domain.
> >
> >
> > --------------------------------------------------------
> >
> > We hope this addresses your concern, and we will incorporate additional discussions in our revised paper. As the discussion period deadline approaches, we would greatly appreciate your acknowledgment of our responses if there are no further questions. Thank you again for your time.
> >
> >
> >
> > [1] Riemannian Bilevel Optimization, Li and Ma, 2024.
> >
> > [2] Riemannian Bilevel Optimization, Sanchayan Dutta, Xiang Cheng, Suvrit Sra, 2024.

---

### Official Review · Reviewer_4rfq · 2024-07-11

**Soundness:** 4
**Presentation:** 4
**Contribution:** 4
**Rating:** 8
**Confidence:** 4

**Summary:**

The paper proposes an RHGD algorithm (Algo 1) to solve the bilevel optimization problems (line 11).
- Thm 1 proves the convergence to a stationary point of $F$ when using different approximations of the hypergradient.
- Thm 2 shows the convergence under stochastic setting.
- Thm 3 proves the convergence with cheaper retraction.
Both synthetic numerical tests and applications to real problem are provided.

**Strengths:**

The paper is well-written. The theoretical results are solid and the contributions are new to me. The algorithm is fully implementable. What I like is the convergence is not only established for the Hessian inverse but also its cheaper approximation. The synthetic experiment is carefully designed.

**Weaknesses:**

1. Assumption 1 is too strong: it requires the trajectory of $z$ to be in the unique geodesic neighborhood of $z^*$.
1. Assumption 2 is slightly too strong, does this mean many simple functions like quadratic functions: $f(x, y)=|x|^2+|y|^2$ do not satisfy this assumption?

**Questions:**

1. Could you please clarify the synthetic problem in Sec 4.1 satisfies all the assumptions? I only find that Assumption 3 is verified. Assumption 2 is automatically satisfied due to the compactness of the Stiefel manifold. However, could you clarify how to verify Assumption 1?
1. Maybe it is a dumb question, but in Sec. 3.1, why HINV and AD are different?

**Limitations:**

Small typo: In line 112 there are 2 negative signs.

---

> ### Author Rebuttal · Authors · 2024-08-06
>
> We thank the reviewer for the positive feedback on our work, particularly acknowledging that our paper is well-written with solid theoretical results. We also greatly appreciate the constructive comments.
>
>
> **1. (W1) Assumption 1 is strong.**
>
> We would like to highlight that Assumption 1 is often unavoidable for analyzing Riemannian optimization algorithms, especially when dealing with geodesic strongly convex functions, see for example [21, 37, 62, 63, 64] that also makes use of such an assumption. This assumption provides bound for the curvature, which is crucial for deploying a trigonometry distance bound (Lemma 3) and achieving linear convergence. This assumption can be satisfied for compact manifolds and by restricting the domain of interest.
>
>
>
> **2. (W2) Assumption 2 is strong.**
>
> Assumption 2 is not strong and can be inferred from Assumption 1 where we consider a compact domain. In this regard, the quadratic function you provided *satisfies* such an assumption when the domain is bounded.
>
>
>
>
> **3. (Q1) Verify the synthetic problem in Sec 4.1 satisfies Assumption 1.**
>
> Thank you for the question. Assumption 1 on the bounded domain can be satisfied directly for the Stiefel manifold.  For the SPD manifold, we can take the maximal domain encompassing all the iterates considered.
>
>
>
>
> **4. (Q2) Why HINV and AD are different.**
>
> HINV computes the analytic expression of hypergradient using the Hessian inverse, which does not depend on the inner iterations. However, AD computes an estimate of hypergradient by differentiating through the inner iterations.

---

> > ### Comment · Reviewer_4rfq · 2024-08-13
> >
> > Dear authors
> >
> > I appreciate your detailed rebuttal and I am so sorry for replying a little late.
> >
> > W1: I don't agree assumption 1 is an 'unavoidable requirement'. In the rebuttal, the author gives some examples of papers using the same condition, and I will take one of them [63] as an example. They require the manifold to be Hadamard, i.e., unique geodesics (I understand you must require unique geodesic for convex function to exist). However, I don't see any requirement for the iteration to be in a compact set in that paper. Could you please point out where is ' All iterates are contained in a compact neighbourhood' in that specific paper?
> >
> > The rebuttal also states 'This assumption can be satisfied for compact manifolds and by restricting the domain of interest.' I don't think the problem is that trivial because 1.  compactness manifolds with no boundary (e.g., Stiefel manifold) leads to the non-existence of convex functions and non-existance of unique geodesic. 2. if you restrict the domain, how can you guarantee the iteration stays in that domain? This is the reason I think the assumption is too strong.
> >
> > W2: I agree with you it can be inferred from assumption 1. But again, assumption 1 is too strong, making me think assumption 2 is too strong. But I can take assumption 2 if you convince me assumption 1 is not strong.
> >
> > Q1: I am so sorry I do not think my question is that trivial. For Stiefel manifold, compactness in Assumption 1 is automatically satisfied, but where is unique geodesic (smooth inverse of exponential)? For SPD manifold, it is not that simple. What you really do is, fix a domain -> fine a step size as stated in Thm. 2, i.e., step size depends on the selection of domain. How can you simply run the algorithm and then find the domain containing all iterations? What is your step size in this case?
> >
> > Q2: thank you for the clarification.
> >
> > Best wishes
> > reviewer

---

> > > ### Author Response · Authors · 2024-08-13
> > >
> > > We sincerely appreciate the follow-up questions and would like to take this opportunity to address them in detail.
> > >
> > >
> > > We understand that our previous response statement "This assumption can be satisfied for compact manifolds and by restricting the domain of interest" (given as part of our response) might not be appropriate. What we mean is compact **subset** of a manifold, which allows unique-geodesic and geodesic convex functions. This is, for example, considered in [63] for analyzing geodesic (strongly) convex functions. All the theorems, i.e., Theorems 9, 10, 11, 12, 13, 14, 15 assumes a bounded domain and make use of the diameter of the domain $D$, which is defined in Theorem 9.
> > >
> > >
> > > With the above clarifications, we proceed to answer your individual questions.
> > >
> > >
> > >
> > >
> > > **1. On Assumption 1.**
> > >
> > > Regarding our Assumption 1, we now believe that our current phrasing in the paper may have caused some confusion. We wish to clarify that our analysis only requires the unique geodesic specifically for the manifold associated with the lower-level problem. We also highlight that the compactness requirement for the upper-level problem is not strictly necessary for analysis. The compactness assumption on the upper-level problem was initially included to facilitate a more natural interpretation of Assumptions 2 and 3. However, in contrast, for the lower-level problem, compactness on the domain and the uniqueness of the geodesic are essential for establishing linear convergence, based on [63].
> > >
> > > We plan to rewrite Assumption 1 as follows.
> > >
> > > **Assumption 1.** All the iterates in the lower level problem are bounded in a compact subset that contains the optimal solution, i.e., there exists a constants $D_k > 0$, for all $k$ such that $d(y_k^s, y^*(x_k)) \leq D_k$ for all $s$. Such eighbourhoods admit unique geodesic. We take $D := \max_{k} (D_1, ..., D_k )$.
> > >
> > >
> > >
> > >
> > > **2. Could you please point out where is 'All iterates are contained in a compact neighbourhood' in that specific paper?**
> > >
> > >
> > > In Theorem 15 of [63], when analyzing convergence for geodesic-strongly convex functions, the the definition of parameter $\epsilon = \min (1 / \zeta(\kappa, D), \mu/ L_g )$ depends on the diameter of the domain $D$, which is defined in Theorem 9. This suggests the bounded domain is still assumed in the analysis, and thus corroborates our claim that the bounded domain assumption is unavoidable in this case.
> > >
> > >
> > > **3. On Assumption 2.**
> > >
> > > We remark that Assumption 2 is also made in the Euclidean bilevel analysis [35], even without assuming Assumption 1. See the first bullet point of Assumption 2 in [35] where function Lipschitzness implies bounded gradient.
> > >
> > >
> > > **4. Stepsize depend on selection of domain.**
> > >
> > > This is a misunderstanding. First, we do not 'select' a domain. The bounded domain is mainly introduced for analysis purpose as we have explained  above. Second, we do not choose stepsize based on the domain. The stepsize is selected based on the Lipschitz constants of the objective function, i.e., from Assumption 2, 3. This is also the case for Euclidean bilevel analysis [35]. See, for example, Theorem 1 in [35]. As commented above, Assumption 2 can be made independent of the Assumption 1, and thus, the stepsize can be chosen independent of the domain.
> > >
> > > -------------------------
> > >
> > > In summary, we sincerely thank you for all the questions, which have largely helped to improve the clarity of the paper. We will ensure a more clear presentation and discussions of the assumptions in our revised version. In particular, we will include the revised assumption as given above. Given the rebuttal discussion is coming to the end, we would greatly appreciate if you could kindly acknowledge our response if there are no further questions. Thank you again for your time.

---

> > > > ### Comment · Reviewer_4rfq · 2024-08-13
> > > >
> > > > Dear authors and reviewer JBnb
> > > >
> > > > I appreciate your comment and I learn a lot from it. I also thank reviewer JBnb for the insightful discussion.
> > > >
> > > > I am fully convinced that assumption 1 is not strong. I am not convinced by 4 in the last reply by the authors, but I don't think that stops me from liking the paper anymore. I still believe that assumptions 1 and 2 are uncheckable for your example but that happens in curved spaces since manifolds are complicated.
> > > >
> > > > In summary, I didn't check all the technical details in the paper, but I like the paper, I think the topic matches the conference and the contribution is clearly shown. I would increase my rating.
> > > >
> > > > Best wishes
> > > >
> > > > reviewer

---

### Official Review · Reviewer_Bnza · 2024-07-12

**Soundness:** 3
**Presentation:** 3
**Contribution:** 3
**Rating:** 6
**Confidence:** 4

**Summary:**

This paper presents a framework for addressing bilevel optimization problems in which variables of both lower- and upper-level problems are constrained on Riemannian manifolds. It introduces multiple hypergradient estimation strategies on manifolds and investigates their estimation error. The paper includes convergence and complexity analyses for the proposed hypergradient descent algorithm on manifolds. Furthermore, it extends these advancements to stochastic bilevel optimization and the utilization of general retraction on the manifolds. The paper also demonstrates the practicality of the proposed framework through several applications.

**Strengths:**

1. The paper is well-written and is easy to read.

2. This paper derives the intrinsic Riemannian hypergradient using the implicit function theorem, proposes four strategies for estimating the hypergradient, and provides estimation error bounds for all the proposed strategies.

3. This paper presents the Riemannian hypergradient descent algorithm for tackling bilevel optimization problems on manifolds, providing convergence guarantees. Furthermore, it extends the framework to the stochastic setting and incorporates retraction mapping on the manifolds.

**Weaknesses:**

1. Firstly, my primary concern pertains to the novelty of this paper. The algorithm proposed in this paper appears to mirror the framework introduced in a prior work [42]. Therefore, it is crucial to delineate the similarities and disparities between this paper and the referenced study [42].

2. The experimental results should include a comparison with the methods presented in [42].

3. The analysis of retraction on manifolds lacks depth and should include the convergence results of three additional hypergradient estimators.

4. Assumption 1 seems to be excessively strict, which is unusual in common bilevel optimization. Moreover, the notation $D$ in $d(x_1,x_2)\le D$ may be confusing. Furthermore, Assumptions 2 and 3 could be inferred from Assumption 1, indicating that this paper may not necessitate Assumptions 2 and 3.

**Questions:**

1. The notations $C_{hinv}$, $C_{cg}$, $c_{ns}$, and $C_{ad}$ in Theorem 1 are not utilized in the main body of the paper.

2. Can the stochastic bilevel problem studied in this paper be extended to the general stochastic setting, i.e.,  $\underset{{x\in \mathcal{M}_x}}{\min} F(x) = \mathbb{E}[f(x,y)]$ and $y^*(x) = \underset{y\in\mathcal{M}_y}{\text{argmin}}~ \mathbb{E}[g(x,y)]$?

3. For other questions, please see the weaknesses above.

---

> ### Author Rebuttal · Authors · 2024-08-06
>
> We thank the reviewer for the general appreciation of our work as well as constructive comments that we address below.
>
>
> **1. (W1)  Similarities and disparities compared to [42].**
>
> We would like to emphasize that [42] is a *concurrent* work and not a prior work. It became available (publicly) after we had completed the draft. Moreover, we would like to highlight several notable differences compared to [42], some of which have already been discussed in lines 48-61.
>
> (1) Our proposed framework is more general and subsumes the developments of [42]. In particular, Algorithms 1 and 2 in [42] are special cases of our proposed RHGD-CG and RSHGD-NS algorithms, respectively. While [42] use exponential map in their Algorithms 1 and 2, we study more general retraction, of which exponential map is one specific instance. It should be noted that our contributions is not limited to CG variants of proposed RHGD algorithm as we study other hypergradient estimation strategies as well. Overall, we provide hypergradient estimation error bound for a variety of estimation strategies and numerically compare them in terms of convergence, estimation error (Figure 1) and runtime (Table 3).
>
> (2) We present many applications that require to solve a Riemannian bilevel optimization problem. Many of them are of independent interest.
>
> (3) In Appendix G, we have shown how the proposed framework also leads to a convergence analysis for Riemannian min-max optimization and Riemannian compositional optimization problems, where they are special instances.
>
> To summarize, our work provides a more general and practical framework than [42].
>
>
>
> **2. (W2) Experiment comparison with the methods presented in [42].**
>
> We have now included numerical comparisons in the one-page supplementary material. Specifically, we compare the performance on the synthetic problem (Figure 1) and hyper-representation task (Figure 2). We see the use of retraction in our case improves the efficiency of the solvers, which is reflected in the reduced runtime to achieve convergence.
>
>
>
> **3. (W3) The analysis of retraction should include the results of three additional hypergradient estimators.**
>
> Thank you for the suggestion. Below we include the convergence results for the three additional hypergradient estimators and will make sure to include them in the final revision.
>
> **Theorem.** (Extension of Theorem 3) Under Assumptions 1,2,3,5 and let $\tilde L_F = 4 \kappa_l c_R M + 5 \bar c L_F$. Consider Algorithm 1 with the following settings.
>
> - HINV: Suppose $\eta_x = \Theta(1/\tilde L_F), S \geq \tilde \theta(\kappa_l^2 \zeta)$. Then $\min_{k=0,..., K-1} \| \mathcal{G} F(x_k) \|^2_{x_k} \leq 16 \tilde L_F \Delta_0/K$.
>
> - CG: Suppose $\eta_x = \Theta(1/\tilde \Lambda), S \geq \tilde \theta(\kappa_l^2 \zeta), T \geq \tilde \Theta(\sqrt{\kappa_l})$, where $\tilde \Lambda = C_v^2 \bar c + \kappa_l^2 ( \frac{5M^2 C_0^2 D^2}{\mu} + \bar c)$, then $\min_{k=0,..., K-1} \| \mathcal{G} F(x_k) \|^2_{x_k} \leq 96 \tilde \Lambda (\Delta_0 + \| v_0^* \|^2_{y^*(x_0)})/K$.
>
> - NS: Suppose $\eta_x = \Theta(1/\tilde L_F), S \geq \tilde \Theta(\kappa_l^2 \zeta)$. Then for any $\epsilon>0$, $T \geq\tilde \Theta(\kappa_l \log(1/\epsilon))$, we have $\min_{k=0,..., K-1} \| \mathcal{G} F(x_k) \|^2_{x_k} \leq 16 \tilde L_F \Delta_0/k + \epsilon/2$.
>
> - AD: Suppose $\eta_x = \Theta(1/\tilde L_F)$, $S \geq \tilde \Theta(\kappa_l^2 \zeta \log(1/\epsilon))$ for any $\epsilon > 0$, we have $\min_{k=0,..., K-1} \| \mathcal{G} F(x_k) \|^2_{x_k} \leq 16 \tilde L_F \Delta_0/k + \epsilon/2$.
>
>
> The proof basically contains three parts, where we first show the convergence of inner iterations in terms of retraction, which is agnostic to the choice of hypergradient estimator. Then we bound the hypergradient estimation error where only AD based hypergradient needs different treatment when using retraction because it requires to differentiate through the retraction. For other hypergradient estimators, the bounds in Lemma 1 still hold given no retraction is used in their computation. Lastly we show the defined Lyapunov function is monotonically decreasing.
>
>
>
>
>
> **4.  (1) Assumption 1 is strict and unusual in bilevel optimization. (2) The notation $D$ in $d(x_1, x_2) \leq D$ is confusing. (3) Assumptions 2 and 3 could be inferred from Assumption 1 and the necessity to include Assumption 2, 3.**
>
> (1) Assumption 1 is often unavoidable in the literature of Riemannian optimization, especially when dealing with geodesic (strongly) convex objectives [21, 37, 62, 63, 64]. This is required to ensure bounded curvature in order to apply the trigonometry distance bound (Lemma 3), which is crucial for achieving linear convergence of the inner iterations. (2) For notation clarity on $d(x_1, x_2) \leq D$, we will change the notation in the revised manuscript. (3) It is correct that Assumption 2 and 3 can be inferred from Assumption 1. Nevertheless, we explicitly state such assumptions for clarity and for properly defining the Lipschitz constants.
>
>
>
> **5. (Q1) The notations $C_{hinv}$, etc in Theorem 1 are not used in the main text.**
>
> We will move such notations to appendix.
>
>
> **6. (Q2) Extension to the general stochastic setting?**
>
> Yes, the proposed framework can accommodate the general stochastic setting. To this end, we require to construct unbiased gradient estimate and properly control its variance. A more formal treatment of such general case is left for future exploration.

---

> > ### Comment · Reviewer_Bnza · 2024-08-12
> >
> > I would like to thank the authors for their responses. I am raising my score to 6.

---

### Official Review · Reviewer_kqCt · 2024-07-12

**Soundness:** 4
**Presentation:** 2
**Contribution:** 3
**Rating:** 6
**Confidence:** 3

**Summary:**

This paper introduces a novel approach for solving bilevel optimization problems where both upper and lower-level variables are constrained on Riemannian manifolds. The authors propose four hypergradient estimation strategies (HINV, CG, NC, AD), analyze their estimation errors with convergence and complexity analysis, and generalize to stochastic bilevel optimization and general retraction. The paper also shows many applications of the framework including hyper-representation over SPD matrices, Riemannian meta-learning, and unsupervised domain adaptation.

**Strengths:**

1. The proposed framework extends bilevel optimization to Riemannian manifolds, which fills a gap in this field.
2. The paper introduces multiple hypergradient estimation strategies and provides a thorough theoretical analysis.
3. Practical Relevance: Demonstrates the applicability of the framework through several machine learning applications, including hyper-representation and meta-learning.

**Weaknesses:**

1. Assumptions: The framework relies on assumptions such as geodesic strong convexity and Lipschitz continuity, which may limit its applicability in some practical scenarios.
2. The practical performance evaluation is limited to synthetic and small-scale problems, which may not fully represent its utility in large-scale real-world applications.

**Questions:**

1. What are the computational trade-offs when choosing between the different hypergradient estimation strategies?

**Limitations:**

The authors acknowledge that the framework relies on some strong mathematical properties that limit its broader applicability and may be improved in the future.

---

> ### Author Rebuttal · Authors · 2024-08-06
>
> We thank the reviewer for acknowledging that our theoretical analysis is thorough and our work has practical relevance. We also appreciate the constructive feedback.
>
> **1. (W1) Assumption such as geodesic strong convexity and Lipschitz continuity.**
>
> The assumption of (geodesic) strong convexity is common for analyzing bilevel optimization, which is also the case even in the Euclidean space [35,40,53]. This is because without strong convexity, the lower-level problem can be ill-defined, where there may exist many local solutions. Studying bilevel problem with non-(geodesic)-strongly-convex lower-level problem is even challenging in the Euclidean space. Nevertheless, it is possible to relax such assumption to (geodesic) convexity with an extra strongly convex regularizer, or to (Riemannian) PL condition where a global minimizer exists.
> On the other hand, the assumption on Lipschitzness is also standard in the bilevel literature such as [35,40,46].
>
>
> **2. (W2) Empirical evaluation is limited to synthetic and small-scale problems.**
>
> We would like to emphasize that in addition to the synthetic and small-scale setting, we  evaluate our approach on real-world scenarios such as unsupervised domain adaptation on Office-Caltech dataset (Section 4.4) and meta-learning where we optimize a convolutional neural network (CNN) with orthogonal constraints on mini-Imagenet dataset (Section 4.3). In particular, the meta-learning example leads to a large-scale problem that involves training a 3-layer CNN,  with each layer kernel parameters constrained to Stiefel manifold. This results in a parameter set of $3 \times 144 \times 16$ parameters. In addition, we train over 20,000 tasks where each task contains 50 images of size $84 \times 84$ (5-ways, 5-shots). We will make sure to include the details on experiment setup in the revised version.
>
>
> **3. (Q1) Computational trade-offs when choosing between the different hypergradient estimators.**
>
> We have discussed computational trade-offs in Table 1 and Appendix H.4 (together with convergence plots in Figure 1), where we compare the runtime for evaluating hypergradient for different estimation strategies. In summary, CG is the most computational demanding but yields the best approximation while AD requires the least computational effort but gives a poor hypergradient estimation. NS balances the computation and estimation error.

---

> ### Comment · Reviewer_kqCt · 2024-08-12
>
> Thanks so much for the responses. I will keep my rating and raise my confidence.

---

### Official Review · Reviewer_CDbu · 2024-07-13

**Soundness:** 3
**Presentation:** 3
**Contribution:** 3
**Rating:** 6
**Confidence:** 4

**Summary:**

This paper addresses bilevel optimization problems where variables of both the lower and upper-level problems are constrained on Riemannian manifolds. To solve such problems, the authors propose a hypergradient descent algorithm under the key assumption that the lower function is geodesically strongly convex and establish its iteration complexity. To efficiently estimate the hypergradient, several methods with error bounds are also proposed. Additionally, the paper provides several interesting application examples of bilevel optimization on Riemannian manifolds and presents numerical results for these examples.

**Strengths:**

- This paper generalizes the widely used hypergradient descent algorithm to the Riemannian manifold setting and provides a comprehensive analysis of the proposed algorithms.
- The paper includes four approaches to estimating the inverse of the Riemannian Hessian and a stochastic version of the proposed algorithm.
- The theoretical analysis is solid, with estimation error bounds and the iteration complexity of the proposed algorithm presented.
- The numerical results are extensive, showcasing results on four interesting Riemannian bilevel optimization application examples.

**Weaknesses:**

- A closely related work is [42], and although the authors claimed key differences in Section 1, some points should be clarified.

     - The various hypergradient estimators beyond conjugate gradient and Neumann might be standard. Do the other estimators show better performance?

     - While the paper presents interesting applications (see instances given in Section 4.2), these problems can also be solved by the methods in [42]. The authors should provide a numerical comparison of the methods.

     - Shall the authors include the complexities of the methods in [42] in Table 1?

- The estimations of the inverse Hessian are also widely used in Euclidean space.

- The assumption of geodesic strong convexity for the lower function is too strong for the Riemannian setting, making the application examples in the paper less typical. Additionally, it should be noted that the lower-level problem in Section 4.2 is constrained in Euclidean space.

**Questions:**

1. Why are the performances of “hinv-20” and “hin-50” similar in Figures 1(a) and 2(a)?

2. Why is “hinv” more efficient, as shown in Figure 1(b)?

**Limitations:**

Yes, the limitations are acknowledged.

---

> ### Author Rebuttal · Authors · 2024-08-06
>
> We thank the reviewer for positive comments on recognizing our solid theoretical analysis and extensive numerical results. We also greatly appreciate  the constructive feedback and comments.
>
>
> **1. (W1) Clarification regarding [42]: (1) Do other estimators show better performance? (2) Provide numerical comparisons with [42]. (3) Include complexities of methods in [42].**
>
> Thank you for raising these points. Below, we address each one individually.
>
> (1) There exists a accuracy-runtime trade-off for these estimators. The trade-off can be observed from Figure 1 where we compared various estimators in terms of convergence, hypergradient error and runtime. We have also additionally compared per-hypergradient runtime for these estimators in Table 3 (Appendix H.4). From the figures and table, we see hypergradient based on automatic differentiation requires the least runtime while suffers from a poor hypergradient approximation accuracy. Neumann series based estimators balance the trade-off while conjugate gradient requires the most runtime but has the lowest approximation error. Choosing a suitable estimator depends on specific applications.
>
> (2-3) We highlight that the methods presented in [42] are covered by the framework that we propose, i.e., Algorithm 1 in [42] coincides with RHGD-CG and Algorithm 2 in [42] is RSHGD-NS that we propose, except that they use exponential map for the update, while our framework works for general retraction. Thus, the methods in [42] can be considered as special cases in our framework where we have already included the complexities.
>
> *Please find  numerical comparisons with [42] in the one-page supplementary material.* In particular, we compare on the synthetic problem (Figure 1) and shallow and deep hyper-representation (Figure 2). We observe the use of retraction in our case results in more efficient update and thus reduces the runtime for convergence.
>
>
>
>
>
> **2. (W2) The estimations of the inverse Hessian are also widely used in Euclidean space.**
>
> While it is true that the estimation strategies for inverse Hessian have been explore in the Euclidean space, in more general manifold setting, certain more care is needed when using second-order derivatives. For example, $\mathcal{G}^2_{xy}$ (used in Algorithm 1) is a second-order cross derivative relating two different manifolds (one for x and the other for y) and defining this is tricky. Additionally, Hess is now an *operator on the tangent space* instead of a matrix in the Euclidean space. Consequently, its inverse needs to be properly characterized and computed.
>
>
>
>
>
>
> **3. (W3) (1) Assumption of geodesic strong convexity is strong. (2) Lower-level problem in Section 4.2 is in Euclidean space.**
>
> (1) Analyzing bi-level problem with non-(geodesic)-strongly-convex lower-level problem is tricky (even in the Euclidean setting) because of non-unique solution of the lower-level problem.
> Non-unique solution would give rise to several challenging cases such hypergradient becoming ill-defined, invertibility of the Hessian, etc. However, it is possible to relax this assumption to geodesic convex lower objective, by adding a strongly convex regularizer term to the objective function.
>
> (2) Since Euclidean space is a special case of Riemannian manifold, it is not unreasonable to consider problems where lower-level problem is in the Euclidean space. However, we would like to emphasize that Sections 4.1 and 4.4 explore lower-level problems constrained to the SPD manifold.
>
>
> **4. (Q1) Why performances of hinv-20 and hinv-50 similar?**
>
> It should be noted that the numbers 20 and 50 refers to the number of inner iterations for solving the lower-level problem. The similar performance of hinv-20 and hinv-50 is because the lower-level problem for such synthetic problem is easy to solve and 20 iterations already provides a good close-to-optimal solutions.
>
>
> **5. (Q2) Why hinv more efficient as in Figure 1(b)?**
>
> In this synthetic example, we can derive an analytic form of the Hessian inverse. In addition, the problem dimension is not large (d = 50) and thus computing the Hessian inverse directly is more efficient in this setup.

---

> > ### Comment · Reviewer_CDbu · 2024-08-13
> >
> > Thank the authors for clarification. I have increased my score to 6.

---

### Author Rebuttal · Authors · 2024-08-06

We sincerely thank the reviewers for the reviews. We are particularly encouraged by the numerous positive comments on our work, including, **well-written** (Reviewer Bnza, Reviewer 4rfq),  **comprehensive analysis** (Reviewer CDbu, Reviewer kqCt, Reviewer JBnb),  **solid theoretical results** (Reviewer CDbu, Reviewer 4rfq), **interesting applications** (Reviewer CDbu), **extensive numerical results** (Reviewer CDbu, Reviewer kqCt, Reviewer JBnb).

Additionally, we greatly appreciate your constructive feedback, which has significantly helped us improve the paper. In response to your comments, we have

- addressed each point individually, and

- have attached a supplementary one-page PDF containing numerical comparisons with [42].


We hope we have addressed all your concerns and questions through our responses as well as the additional experimental results. We look forward to more interactive discussions during the discussion phase.

---

### Author Response · Authors · 2024-08-14
**Thank you!**

Dear ACs and Reviewers

We are greatly encouraged that our rebuttals have effectively addressed your concerns, resulting in positive adjustment to the scores (Reviewer CDbu, Reviewer Bnza, Reviewer 4rfq) and to the confidence (Reviewer kqCt). We sincerely appreciate your recognition of our contributions as well as the constructive feedback that has improved our paper. Thank you for your invaluable time and insightful comments throughout this process.

Regards

Authors

---

### Decision · Program_Chairs · 2024-09-25

**Decision:**

Accept (poster)

**Comment:**

This paper proposes a framework for solving bilevel optimization problems with both lower and upper level variables constrained on Riemannian manifolds. It introduces hypergradient estimation strategies on manifolds, analyzes their estimation error, and provides convergence and complexity analysis for the proposed hypergradient descent algorithm. The framework is extended to stochastic bilevel optimization and general retraction. Additionally, the paper demonstrates the utility of the framework in several applications.

The paper makes notable contributions by developing techniques for hypergradient estimation on manifolds, providing theoretical analysis for the hypergradient descent algorithm and its stochastic variant, and demonstrating their efficiency through concrete applications.  The reviewers agreed that the paper was strong enough to merit publication, especially after considering the rebuttal. I recommend acceptance.